# COBRA improves the completeness and contiguity of viral genomes assembled from metagenomes

**LinXing Chen** [1,2] ✉ **& Jillian F. Banfield** [1,2,3,4,5] ✉

Viruses are often studied using metagenome-assembled sequences, but genome incompleteness hampers comprehensive and accurate analyses. Contig Overlap Based Re-Assembly (COBRA) resolves assembly breakpoints based on the de Bruijn graph and joins contigs. Here we benchmarked COBRA using ocean and soil viral datasets. COBRA accurately joined the assembled sequences and achieved notably higher genome accuracy than binning tools. From 231 published freshwater metagenomes, we obtained 7,334 bacteriophage clusters, ~83% of which represent new phage species. Notably, ~70% of these were circular, compared with 34% before COBRA analyses. We expanded sampling of huge phages (≥200 kbp), the largest of which was curated to completion (717 kbp). Improved phage genomes from Rotsee Lake provided context for metatranscriptomic data and indicated the in situ activity of huge phages, *whiB*-encoding phages and *cysC*- and *cysH*-encoding phages. COBRA improves viral genome assembly contiguity and completeness, thus the accuracy and reliability of analyses of gene content, diversity and evolution.

Viruses infect and kill their hosts, alter host metabolisms via auxiliary metabolic genes (AMGs) and mediate horizontal gene transfer[1–4]. In the past decade, numerous efforts have made the study of viruses more practical, including but not limited to tools for virus identification[5–7], viral binning[8–10], taxonomic classification[11–13], automating AMG identification[7] and viral genome completeness estimation[14].

Many viral studies rely on metagenome-assembled sequences[4], most of which are partial[14]. The diversity of viruses is extremely high[4], yet a relatively small fraction is represented by complete genomes[15–17], and only a small subset of these are huge phage genomes (≥200 kbp, or jumbo phages)[18–24]. The lack of complete genomes often precludes the classification of extrachromosomal elements and confounds diversity analyses[15]. When complete genomes are available, it is possible to evaluate phage species richness, AMG contents[20], genome structure[25] and genome sizes[19].

A subset of de novo assembled metagenomic contigs can be joined via end overlaps[26]. This is because the assemblers based on the de Bruijn graph generally break at positions with multiple paths. The fragments from a single population can sometimes be joined, potentially to obtain genomes that can be further curated to completion[18–20,26]. Manual curation is used to extend contig ends before joining[26], evaluate the validity of joins and eliminate chimeric joins introduced during assembly. However, manual curation is labour intensive and thus rarely included in metagenomic analysis pipelines. Nonetheless, some tools have been developed to improve the quality of viral and bacterial genomes, including ContigExtender[27], Phables[28] and Jorg[29]. Binning is another strategy to better sample viral genomes from metagenomes[30,31], with available tools including vRhyme[10], CoCoNet[8] and PHAMB[9]. However, binning algorithms are approximate and they do not improve the contiguity of individual sequences. Accordingly, we developed Contig Overlap Based

[1]Department of Earth and Planetary Sciences, University of California, Berkeley, Berkeley, CA, USA. [2]Innovative Genomics Institute, University of California, Berkeley, Berkeley, CA, USA. [3]Department of Plant and Microbial Biology, University of California, Berkeley, Berkeley, CA, USA. [4]Department of Environmental Science Policy, and Management, University of California, Berkeley, Berkeley, CA, USA. [5]Earth and Environmental Sciences, Lawrence Berkeley National Laboratory, Berkeley, CA, USA. ✉e-mail: linxingchen@berkeley.edu; jbanfield@berkeley.edu

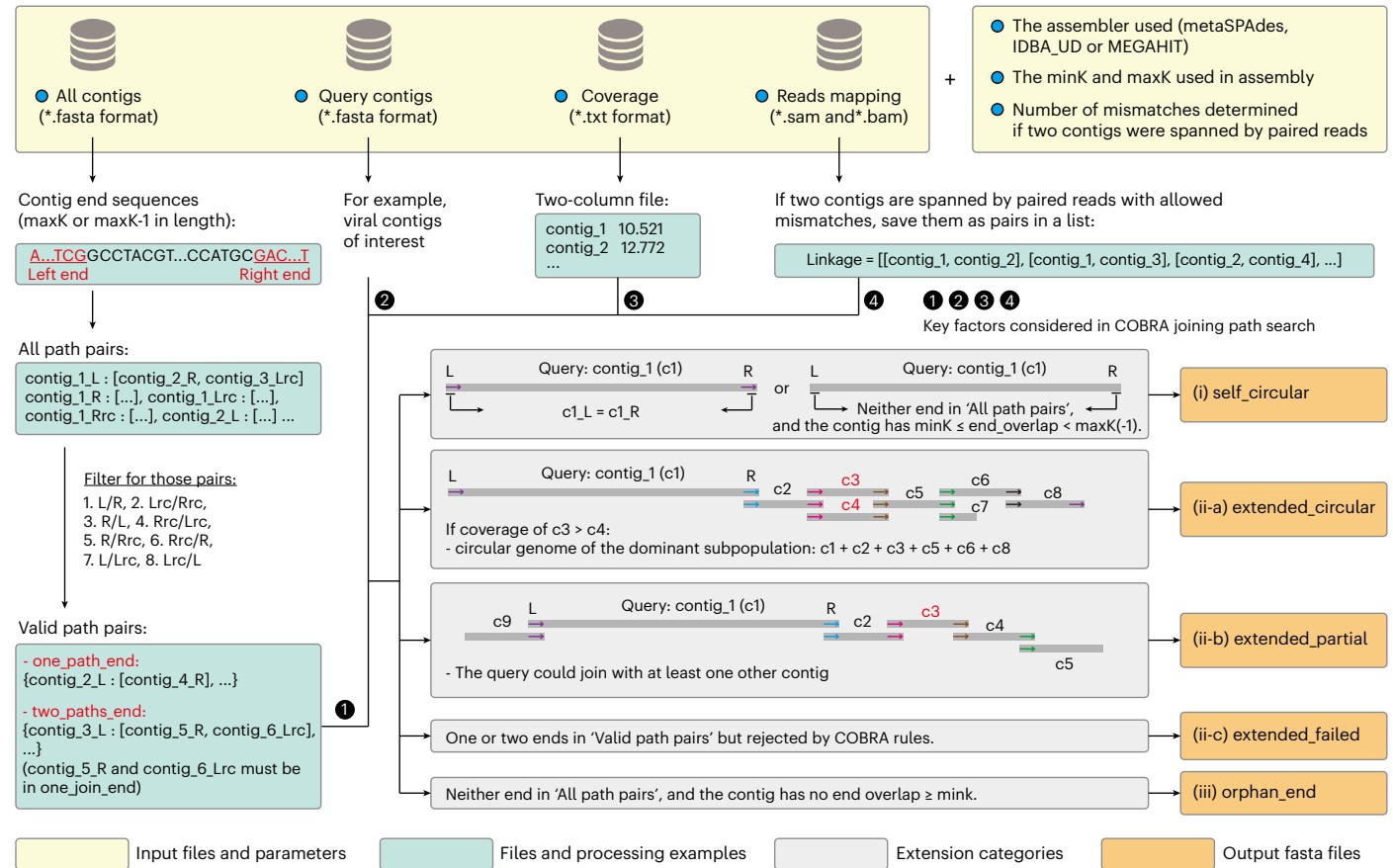

**Fig. 1 | The input files and parameters, processing steps and output files of COBRA.** COBRA requires four input files: a fasta file containing all contigs from the assembly, another fasta file containing the query contigs, a two-column file with the sequencing coverage of each contig and a read mapping file of all contigs. The parameters 'assembler' and 'maxK' determine the length of contig end sequences, and 'minK' evaluates the end overlap sequence length of a given query to determine if it is a 'self_circular' contig. See Extended Data Fig. 1 for more detailed information. The different extension categories are in grey boxes, indicating their corresponding output categories (i, ii or iii) and associated files. COBRA generates five fasta files for each analysis, accompanied by summary files.

Re-Assembly (COBRA) to detect, analyse and join contigs from a single metagenomic assembly. COBRA evaluates coverage and paired read linkages before joining contigs, following manual curation methods.

We tested the ability of COBRA by analysing an ocean virome dataset[32] and a soil viral dataset. COBRA accurately joins contigs assembled from short Illumina reads alone, to generate large genome fragments and sometimes circular genomes. Compared with the performance of the evaluated binning tools, almost all of the COBRA genomes were accurate and not confounded by the contamination introduced by binning. We subsequently used COBRA to recover high-quality phage genomes from 231 freshwater metagenomes, expanding the genomic diversity of huge phages, *whiB*-encoding actinophages and *cysC*- and *cysH*-encoding phages. Thus, we show that COBRA can improve and accelerate viral research.

## Results

**Simulations show the basis for joining contigs.** We used simulations to investigate why and how fragmentation occurs when short paired-end reads are assembled by metaSPAdes, IDBA_UD and MEGAHIT (Supplementary Table 1). The simulations included (1) repeats within a genome (Supplementary Figs. 1 and 2), (2) regions shared by different genomes (Supplementary Fig. 3) and (3) within-population sequence variation (Supplementary Figs. 4 and 5), taking into account a range of relative abundances for cases in (2) and (3). The simulated datasets were de novo assembled individually (Supplementary Information). In the vast majority of cases in which fragmentation occurred because of repeats, the assemblers introduced end sequences (maxK for metaSPAdes and

MEGAHIT assembly, or maxK-1 for IDBA_UD assembly) that could be used to suggest contig joins. We acknowledge that these initially identified potential joins may not be legitimate, but subsequent steps that make use of additional information (see below) identify and remove inaccurate joins. These findings informed the development of COBRA. We suggest using the contigs not scaffolds for COBRA analyses if assembled using IDBA_UD, to avoid any errors that may be introduced during scaffolding[26,33].

COBRA joins metagenome-assembled sequences. COBRA made joins that the assembler chose not to make so long as there is sufficient support. Ideally, the assembler will not make a join that is non-unique, but some non-unique options could arise owing to a single read (for example, because of sequencing error), or a few reads (for example, from a strain variant) that do not represent a unique part of the genome (that is, the coverage is much lower). Thus, the first criterion that COBRA uses to evaluate potential joins is coverage. COBRA detects contigs with shared end overlaps of the expected length (maxK for metaSPAdes and MEGAHIT, maxK-1 for IDBA_UD, same below), and checks whether the contigs have similar sequencing coverage and are spanned by paired reads (Fig. 1 and Extended Data Fig. 1). These joins are considered legitimate and the contigs are joined.

In the first step, COBRA processes all contigs from a given assembly and retrieves the end sequences (maxK or maxK-1) for each contig. COBRA first identifies all contig end pairs with the same end sequences, considering both the end sequence and its reverse complement (rc). These identified pairs are then filtered to retain only those that could potentially be joined (Fig. 1). The filtered pairs are then examined to

identify valid path pairs. COBRA labels an end for which there is only one possible join as 'one_path_end' (Extended Data Fig. 1a) and labels an end A as 'two_paths_end' if it shares its sequence with two other ends (ends B and C), and ends B and C share the sequence exclusively with end A (Extended Data Fig. 1b). Our analysis reveals that in a single assembly, the ends in categories 'one_path_end' and 'two_paths_end' usually account for over 99% of all ends sharing sequences with other ends.

In the second step, COBRA considers each of the provided queries and extends each end sequentially. COBRA first identifies 'self_circular' (category i) contigs with two possible cases, (1) the two ends of a given contig is a valid path pair or (2) neither end of a given contig has an end pair, but has a shorter end overlap length that is ≥minK. Next, COBRA searches for potential joining paths for each end based on valid path pairs. It considers the sequencing coverage ratio between the query contig and a given candidate contig to be included in the joining path (Extended Data Fig. 1c), and requires that the joins are spanned by paired reads. The path search stops when (1) the end does not share its sequence with any other end, (2) the end has three or more paths and (3) the end is 'one_path_end' or 'two_paths_end', but the coverage ratio requirement is not met and/or there is no read pair spanning the join. When a query contig is extended from one end and loops back to the other end, it is classified as 'extended_circular' (category ii-a). For other queries that are extended but do not result in circularization, their status is designated as 'extended_partial' (category ii-b). If at least one end of a query contig matches other ends but the join is not considered valid owing to the coverage ratio and/or lack of spanning paired reads, the query is labelled as 'extended_failed' (category ii-c). In cases in which a query contig does not share any end sequence with others, it is assigned as 'orphan_end' (category iii).

In the third step, COBRA assesses all potential joining paths identified in the second step and ensures that the paths are unique before finalizing joins. An important, but rare, case involves a query that can be extended along two (or more) seemingly unique paths (Extended Data Fig. 2a). In such cases, all queries will be assigned as 'extended_failed'. In addition, COBRA searches for cases in which both ends of a query contig extend into sequences that are closely related to each other, and assign the query to 'extended_failed' once confirmed (Extended Data Fig. 2b).

In the last step, the classifications of the query contigs are compiled. Sequences in the 'self_circular' category are saved, and those in the 'extended_circular' and 'extended_partial' categories are joined and saved.

COBRA accurately joins sequences from benchmarking datasets. For benchmarking, we reanalysed an ocean virome sequenced with both Illumina and Nanopore and for which complete Nanopore-based genomes were obtained[32]. The short Illumina reads of the ocean virome 250 m sample[32] were assembled using metaSPAdes, IDBA_UD and MEGAHIT. For each assembly, we recovered 2,377, 2,304 and 2,321 contigs, respectively (Extended Data Fig. 3 and Supplementary Table 2). These were used as the queries for the following COBRA analyses.

COBRA categorized as circular (that is, 'self_circular') or extended 42–56% of the queries, and 7–14% and 30–50% of the remaining queries were 'extended_failed' and 'orphan_end' (Supplementary Table 2). In all but one case, the queries in the 'orphan_end' category had significantly lower sequencing coverage than the queries of other categories (unpaired $t$-test; Fig. 2a), probably suggesting that these contigs broke during assembly owing to insufficient reads for further extension.

We evaluated the accuracy of COBRA sequences in categories i, ii-a and ii-b by alignment fraction (AF) analyses (Fig. 2b and Methods). Generally, the higher the AF_COBRA value is, the more accurate the COBRA join is considered to be. The higher the AF_polished value is, the more complete the COBRA sequence is. AF_COBRA values averaged 97.0–98.4% (Supplementary Fig. 6), indicating that COBRA accurately joined the Illumina-based contigs. Lower AF_COBRA values generally occurred because (1) COBRA selected and joined strain variant contigs that represented higher-abundance subpopulations, yet the

corresponding polished genome represented lower-abundance subpopulations (Supplementary Fig. 7) or (2) COBRA sequences were very similar, but not identical, to the corresponding polished genomes (Supplementary Fig. 8). In addition, two MEGAHIT COBRA 'extended_partial' sequences had high AF_polished (98.6% and 99.3%, respectively) but relatively low AF_COBRA (72.8% and 76.9%, respectively), as the original queries were longer than the corresponding polished genomes (Fig. 2b).

We assessed the length and quality of the queries and their COBRA sequences in the 'extended_circular' and 'extended_partial' categories. The average length increased from 18.5–20.0 kb to 31.0–32.5 kb (Fig. 2c), and the total number of complete and circular, and high-quality, genomes rose from 28–46 (3–4%) to 215–241 (23–28%) (Fig. 2d). Notably, this was achieved by joining up to 38 contigs into a single sequence (Fig. 2e,f), and 36–45 of the putative complete genomes generated by COBRA were not reported in the original study.

The Nanopore-based analyses obtained more nominally complete genomes (1,864)[32] than COBRA did (100–166), yet COBRA has several advantages. First, it is more cost-effective owing to the lack of requirement for both long and short reads (essential for validation and error correction; for example, ref. [34]). It is applicable on samples with insufficient quantities of high-quality DNA for long-read sequencing. COBRA can be applied to the enormous number of samples that have already been sequenced with only short paired-end reads.

COBRA outperforms prevalent binning tools. We compared the performance of COBRA to that of the binning tool of MetaBAT 2 (ref. [35]), vRhyme[10] and CoCoNet[8]. We filtered the IDBA_UD assembly of the 250 m sample[32] (see above) and obtained 2,632 contigs (Methods) for binning by MetaBAT 2, vRhyme, CoCoNet and contig extension via COBRA (Fig. 3a).

We compared the bins and the COBRA sequences to the published polished complete genomes to evaluate the accuracy of all approaches. We defined 'good bin' and 'good join', 'problematic bin' and 'problematic join', and 'contaminated bin' and 'contaminated join' for binning tools and COBRA, respectively (Fig. 3a and Methods). COBRA far outperformed all binning tools in its ability to recover high-quality viral genomes (Fig. 3b,c). COBRA made 386 'good joins', which are 1.7–5.8 times more than the contig assignments to 'good bins' (66–233; Fig. 3e). The cumulative length of accurate viral sequences generated via 'good joins' is 9.13 Mb, with an average length of 23.6 kb. The cumulative length of 'good bins' is 1.08–4.32 Mb, with an average length of 16.4–19.8 kb per bin (Fig. 3d,e).

We investigated the problematic and contaminated bins or joins. Notably, only 1 out of 400 COBRA sequences was contaminated, while the binning tools generated 111–261 contaminated bins (40–80% of all bins; Fig. 3f). In total, 13 COBRA joins and 5–47 bins were problematic (Fig. 3f), and they were all involved closely related virus genomes with high average nucleotide identity (ANI; 85–92%; Fig. 3g). Compared with binning tools, COBRA had the best metrics, including precision (0.98 versus 0.06–0.47), recall (0.83 versus 0.61–0.71), F1 score (0.90 versus 0.11–0.56), specificity (0.98 versus 0.07–0.61) and accuracy (0.91 versus 0.12–0.64; Supplementary Table 3). COBRA achieved similar performance in joining viral sequences from a soil metagenomic dataset (Extended Data Figs. 4–6).

## Application of COBRA to freshwater metagenomes

Freshwater ecosystems contain phages that infect functionally important populations[16,20,36], yet their diversity is poorly understood. Here we assembled 231 published freshwater metagenomes (Supplementary Table 4) and used COBRA to generate high-quality phage genomes from assembled contigs ≥10 kb (122,107 in total; Extended Data Fig. 7). We filtered the COBRA output for essentially complete genomes as assessed by CheckV14 and obtained 8,527 circular and 3,591 high-quality genomes (Fig. 4a). COBRA substantially improved the quality of the sequences (Fig. 4b), and the product genomes were, on average, 30 kb and 25 kb longer than the query contigs (Fig. 4c).

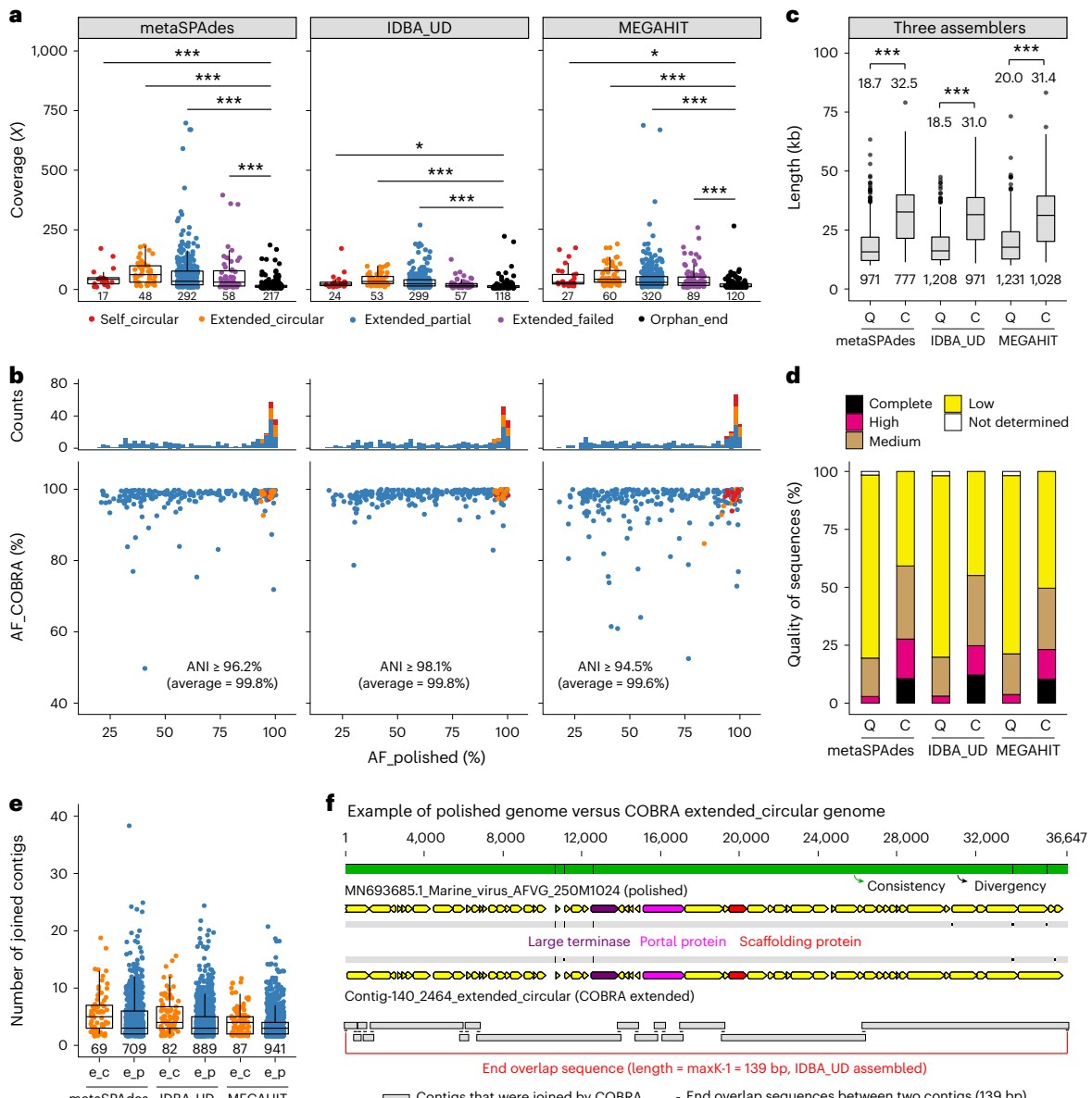

**Fig. 2 | Benchmarking of COBRA using an ocean virome dataset of polished and complete viruses. a**, The coverage of query contigs in the 'orphan_end' category and others. The average coverage was compared using two-sided unpaired *t*-test (\*$P < 0.05$; \*\*\*$P < 0.001$). **b**, The pairwise genome AF of 'self_circular', 'extended_circular' and 'extended_partial' COBRA sequences against the corresponding polished genomes. The number of COBRA sequences in each category is plotted at the top. The aligned region's minimum ANI is shown, with the average ANI in brackets. See **a** for the figure legend. **c,d**, Comparison of the 'extended_circular' and 'extended_partial' sequences and the corresponding contigs joined by COBRA regarding length (**c**) and quality (**d**). If several raw contigs were joined into one COBRA sequence, the length and quality of the COBRA sequence were counted only once. **c**, The length distribution of query contigs

('Q') and COBRA sequences ('C'). The average length of raw contigs and COBRA sequences are shown and compared using two-sided unpaired *t*-test (\*\*\*$P < 0.001$). **d**, The quality of query contigs ('Q') and COBRA sequences ('C') evaluated by CheckV. **e**, The number of contigs joined to generate 'extended_circular' ('e_c') and 'extended_partial' ('e_p') sequences. **f**, An example of an 'extended_circular' sequence compared with the corresponding polished genome. The contigs used are aligned at the bottom with their overlap shown. For box plots in **a**, **c** and **e**, centre lines, upper and lower bounds, and upper and lower whiskers show median values, 25th and 75th quantiles, and the largest and smallest non-outlier values, respectively. Outliers are defined as having a value >1.5 × interquartile range (IQR) away from the upper or lower bounds. No *P* values were corrected. For panels **a**, **c** and **e**, the numbers under that boxes indicate the number of sequences evaluated.

The 12,118 genomes were clustered into 7,432 species-level genomes (95% sequence similarity; Methods and Fig. 4d). Of these, 69 were virophages that replicate along with giant viruses and coinfect eukaryotic cells[37] and 29 were eukaryote viruses or undetermined, which were excluded from further analyses. The remaining 7,334 phage species genomes included 5,169 circular and 2,165 high-quality genomes, with most having a genome size of <50 kb (Supplementary Table 5). Around 70% and 17% of the phage species were represented by only one and two genomes, respectively. More than 99% of the phage species were detected at only one sampling site.

Taxonomic analysis indicated that the phage species were mostly members of class Caudoviricetes, including hundreds that infect Actinobacteria (Supplementary Figs. 9 and 10). Co-analyses of the 7,334 species genomes with the previously published genomes showed that 82% of them (6,047) were novel at the species level (Fig. 4e).

**Diversity expansion and RNA expression of huge phages**
Another motivation for developing COBRA was to obtain genomes of huge phages (≥200 kb in length)[19,38–40]. Of the phage species genomes, 167 were classified as huge phages (Fig. 4d) and their genomes

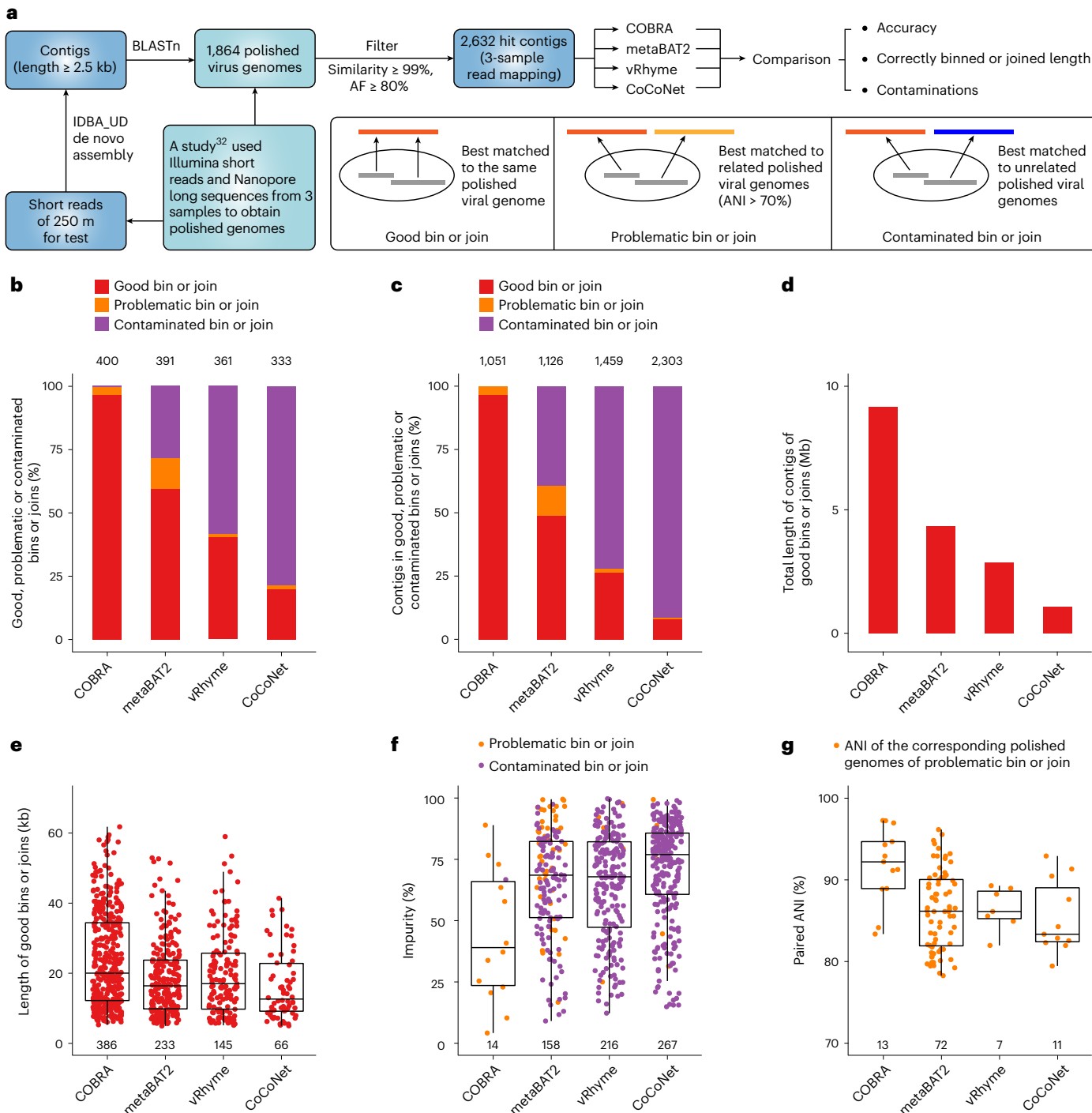

**Fig. 3 | Performance comparison of COBRA and widely used binning tools.**
**a**, The flowchart shows the comparison pipelines. The definitions of 'good', 'problematic' and 'contaminated' bin or join are provided in the accompanying box. Note that only one mapping file is needed for COBRA as input, whereas the coverage profiles were obtained from all three mapping files for the binning tools. **b**, The percentage of 'good', 'problematic' and 'contaminated' bins or joins. **c**, The percentage of contigs in 'good', 'problematic' and 'contaminated' bins or joins. In **b** and **c**, the total absolute numbers are shown at the top. For bins and joins, only those with at least two contigs binned or joined were considered and compared. **d**, The total length of good bins and good joins. **e**, The individual lengths of good bins and good joins; their total lengths are shown at the top. **f**, The impurity rates of 'problematic' and 'contaminated' bins and joins. **g**, The paired ANI of genomes that the contigs of 'problematic' bins or joins were matched to. For box plots in **e**, **f** and **g**, centre lines, upper and lower bounds, and upper and lower whiskers show median values, 25th and 75th quantiles, and the largest and smallest non-outlier values, respectively. In panels **e**–**g**, the numbers under the boxes indicate the number of bins or joins evaluated.

underwent manual curation. In addition, 100 low- or medium-quality huge-phage genomes (coverage ≥20×) were also chosen for manual curation. From the total of 267 huge-phage species genomes, 81 were completed (error-free, gap-free, circular genomes). The largest was initially 712 kb and reached 717 kb after curation to completion. To our knowledge, this is the second-largest complete phage genome (the biggest is 735 kb)[19]. Two phage genomes >800 kb in length were reported recently[41], but they are largely bacterial (Supplementary Fig. 11).

The average genome size of the 267 huge phages generated was 285 kb (Fig. 5a). In comparison, the original query contigs

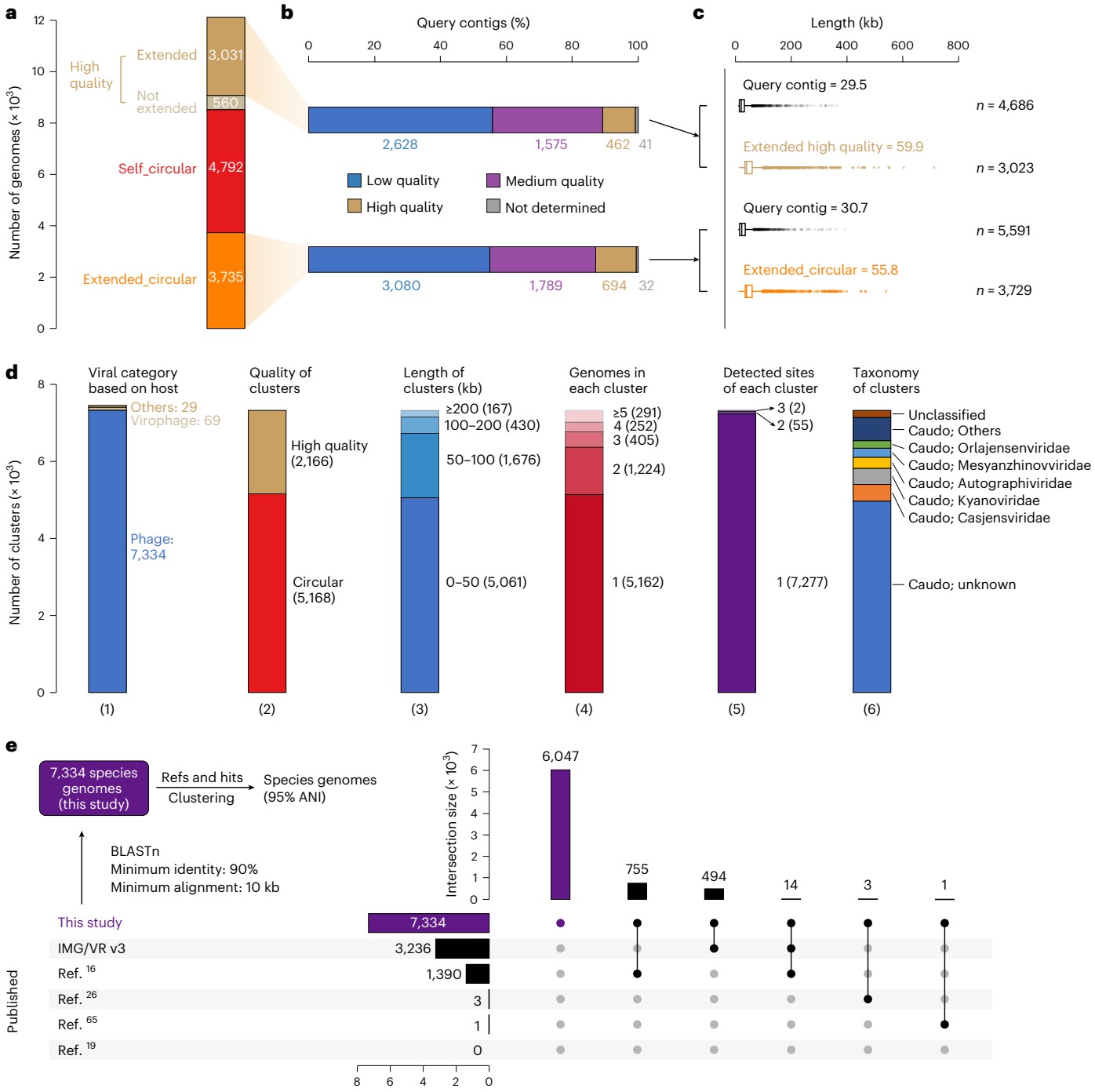

**Fig. 4 | Overview of circular and high-quality phage genomes from freshwater ecosystems. a**, The number of high-quality, 'self_circular' and 'extended_circular' genomes. **b**, The quality of query contigs that COBRA used to generate the extended high-quality and circular genomes. The quality of the genomes was evaluated by CheckV. **c**, The length of COBRA sequences and corresponding query contigs of 'extended_partial' high-quality genomes and 'extended_circular' genomes. In the box plot, the centre lines, upper and lower bounds, and upper and lower whiskers show median values, 25th and 75th quantiles, and the largest and smallest non-outlier values, respectively. Outliers are defined as having a value >1.5 × IQR away from the upper or lower bounds. **d**, The clustering of viral genomes. Bar plots show (1) the number of clusters identified as phages, virophages, eukaryotic viruses and undetermined ('others'). The plots also show

details for the 7,334 phage clusters, including (2) the number of circular and high-quality representative genomes, (3) their length distribution, (4) the number of genomes in each cluster, (5) the number of sites detected with each cluster and (6) the taxonomic assignment of each cluster. Caudo, Caudoviricetes. 'Caudo; others' means the other families excluding the listed ones. 'Caudo; unknown' means all those could be assigned only at the level of Caudoviricetes. **e**, The novelty of phage species genomes identified in this study via comparison with published genomes. Of the 6,046 newly reported phage species genomes, 4,109 are circular and 1,937 are high quality. Please note that, before clustering, the genomes from published databases were prefiltered to retain only those with a minimum alignment length of 10,000 bp (minimum sequence similarity of 90%) with phage genomes obtained in this study.

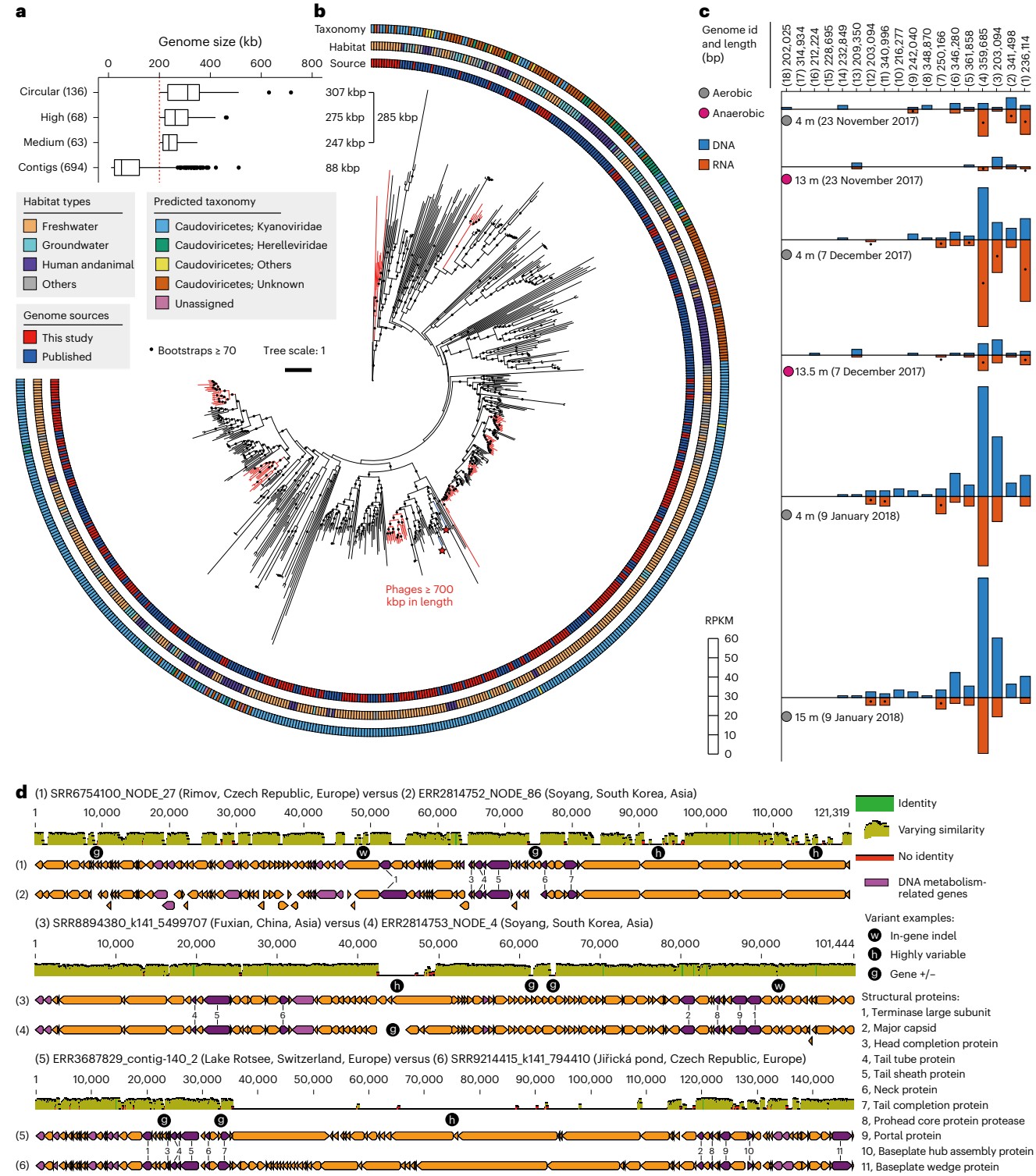

**Fig. 5 | Genomes from freshwater ecosystems expand huge-phage diversity.**
**a**, The number and length of huge phages newly reported in this study from freshwater metagenomes and the corresponding query contigs (≥10 kb in length) joined by COBRA. In the box plot, the centre lines, upper and lower bounds, and upper and lower whiskers show median values, 25th and 75th quantiles, and the largest and smallest non-outlier values, respectively. Outliers are defined as having a value >1.5 × IQR away from the upper or lower bound. **b**, The phylogeny of huge phages based on the concatenated sequences of core structural proteins. The coloured stripes in the inner ring indicate the source of genomes (published or in this study). The coloured stripes in the middle ring indicate the habitats where the phage genomes were reconstructed. The coloured stripes in the outside ring indicate the predicted taxonomy of the genomes. The subclades

with the majority (>80%) of their genomes reconstructed in this study are highlighted in red. The two phages with genome size >700 kb (one published, one from this study) are indicated by red stars. **c**, The detection and transcription profiles of the Rotsee Lake huge phages in the six samples with combined DNA and RNA analyses. The RPKM was calculated for each huge phage in each sample. A black dot indicates that the RNA RPKM is larger than the DNA RPKM of the huge phage in the corresponding sample. **d**, Genomic comparison of similar huge phages from distant collecting sites. Three pairs are shown as examples (see Extended Data Fig. 9 for Mauve alignment). Structural protein genes are shown in purple, their corresponding annotations are included and DNA metabolism-related genes are shown in pink.

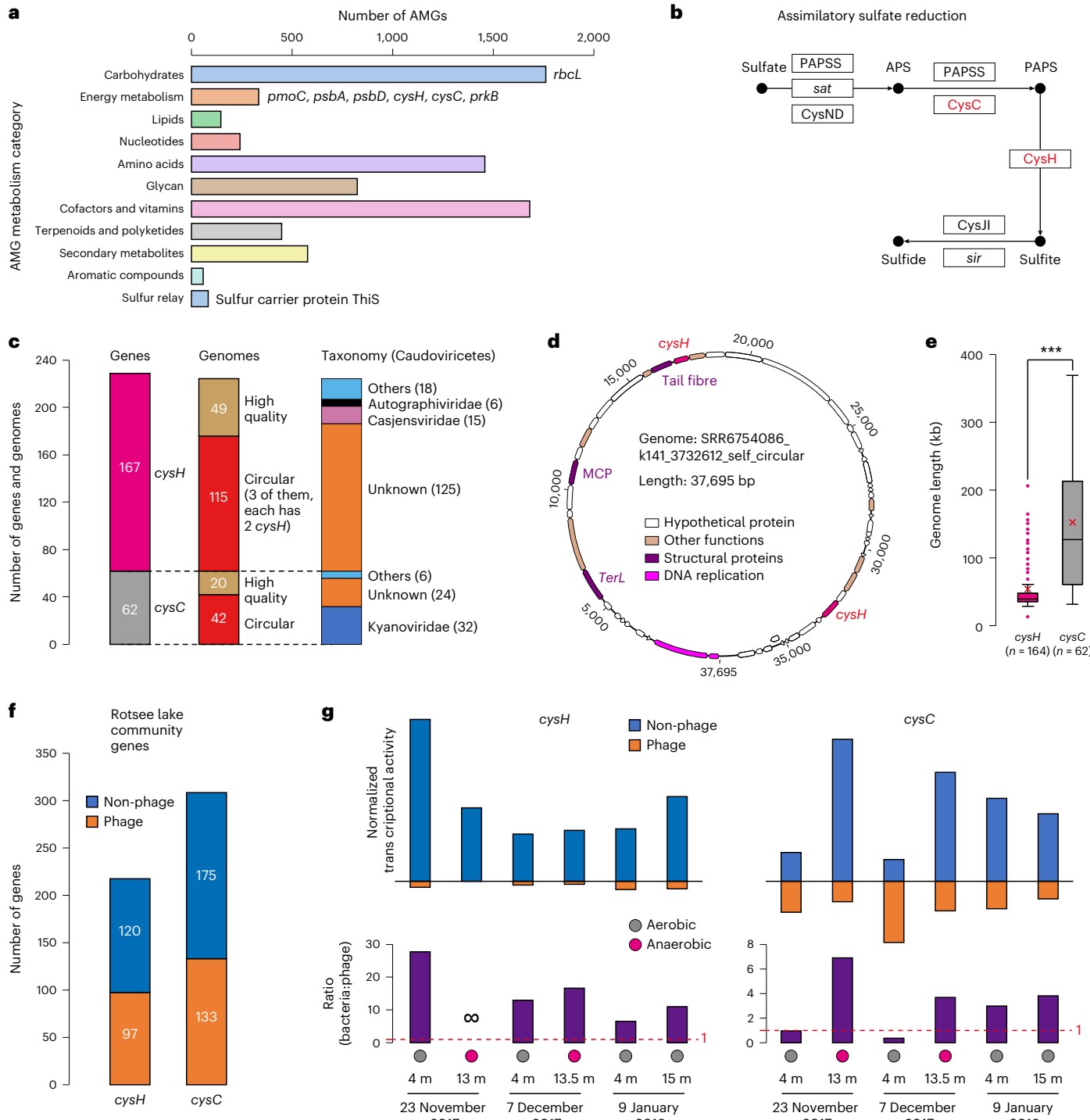

**Fig. 6 | Genomic and transcriptomic analyses of phage species encoding *cysC* and *cysH* genes. a**, Summary of AMGs identified in the phage genomes. **b**, The *cysH* and *cysC* genes are involved in assimilatory sulfate reduction. PAPSS, 3′-phosphoadenosine 5′-phosphosulfate synthase; APS, adenylyl sulfate; PAPS, 3′-phosphoadenylyl sulfate. **c**, The quality and taxonomy of genomes encoding *cysH* and/or *cysC*. **d**, One of three genomes that encode two *cysH* genes each. **e**, The length distribution of genomes encoding *cysH* and *cysC*. The average length of each category is indicated by a red '×'. The average length of *cysH*- and *cysC*-encoding genomes was compared using two-sided unpaired *t*-test (\*\*\**P* < 0.001).

In the box plot, the centre lines, upper and lower bounds, and upper and lower whiskers show median values, 25th and 75th quantiles, and the largest and smallest non-outlier values, respectively. Outliers are defined as having a value >1.5 × IQR away from the upper or lower bounds. **f**, The number of bacteria- and phage-encoded *cysH* and *cysC* identified in Rotsee Lake metagenomes. **g**, The total normalized transcriptional activity and ratio of *cysH* and *cysC* genes in Rotsee Lake metatranscriptomes. Note that the *cysC* phage transcripts are more abundant than the non-phage transcripts in the 4 m sample from 23 November 2017 and 7 December 2017.

had an average size of 88 kb, and only 102 were ≥200 kb. Thus, COBRA is highly effective in generating undersampled huge-phage genomes[19,24].

Phylogenetic analyses based on concatenated sequences (Methods) revealed that the majority of the newly reconstructed huge phages are typically most similar to those identified from

freshwater or groundwater (Fig. 5b). Overall, our analyses broadened the known diversity of huge phages.

Genomic and transcriptomic data from three time points from Rotsee Lake showed the persistence and activity of huge phages, primarily in the aerobic water layers (Fig. 5c). Genes for structural proteins were highly transcribed (Extended Data Fig. 8). Thus, huge phages actively shape microbial community structure and thus biogeochemical cycles within the aerobic parts of the lake.

A comparison of huge-phage genomes from different countries revealed that genes for structural proteins and DNA metabolism retain high nucleotide similarity, and that loss or gain of other genes is primarily driving their divergence (Fig. 5d and Extended Data Fig. 9).

### Detection of AMGs in phage genomes

We explored the inventory of AMGs in the 7,334 high-quality genomes. The majority of AMGs are involved in the metabolism of carbohydrates, amino acids, glycans, and cofactors and vitamins (Fig. 6a), and some with photosynthesis[42] and methane oxidation[20]. We identified 62 *cysC* and 167 *cysH* genes implicated in assimilatory sulfate reduction (Fig. 6b and Supplementary Fig. 12). These genes were generally detected in circular genomes and are from phages from multiple taxa (Fig. 6c). Three circular genomes each contained two *cysH* genes (see Fig. 6d for an example). Notably, phages encoding *cysC* genes exhibited significantly larger genome sizes (Fig. 6e). Of the 231 freshwater samples, 157 contained at least one phage with *cysC* and/or *cysH*, although the majority were present at only one sampling time point (Supplementary Tables 6 and 7).

Using Rotsee Lake metatranscriptomic data, we show that under both aerobic and anaerobic conditions, the transcriptional activity of non-phage-encoded *cysH* genes generally exceeded that of phage-encoded *cysH*. However, phage-encoded *cysC* genes exhibited a greater level of transcriptional activity than their non-phage counterparts under certain aerobic conditions (Fig. 6f,g). These findings show that phages present in freshwater ecosystems can impact sulfur cycling via the assimilatory sulfate reduction process. The infinity sign indicates that there is no detected activity for the phage-encoded *cysH* gene.

### Discussion

Metagenomics is an important approach for studying viruses. However, fragmented genomes hinder the understanding of their diversity and ecological significance[14]. COBRA seeks to complete or nearly complete viral genomes using methods analogous to those used in manual genome curation[26]. COBRA can extend contigs of any length, unlike binning tools that typically require contigs of a length that is sufficient to establish reliable sequence features like tetranucleotide frequency. COBRA can generate single contigs (sometimes, circular genomes; Figs. 1 and 2), whereas binning tools usually obtain MAGs with two or more contigs (Fig. 3). Thus, the resulting COBRA sequences are more readily evaluated for their quality using tools like CheckV[14], which does not work on bins with multiple sequences. MetaBAT 2, vRhyme and CoCoNet need multiple related samples for coverage profile calculation, and PHAMB needs paired metagenome and metavirome datasets for better performance. By contrast, COBRA works efficiently on a single metagenome sample (Fig. 3). Most importantly, COBRA is much more accurate than the evaluated binning tools (Fig. 3). When compared against other genome improvement tools, COBRA is much faster than ContigExtender (Supplementary Table 8)[27] and does not require assembly graphs (for example, Phables[28]) or resource-intensive reassemblies (for example, Jorg[29]). Thus, COBRA will serve as a powerful tool in viromics research.

In showing the utility of COBRA, we added >6,000 new phage species genomes (Fig. 5) from freshwater ecosystems[15,36]. There is minimal overlap between the viral genomes reconstructed in this study and published viral datasets. As our study included only 231 freshwater metagenomes, it is likely that viruses in the freshwater ecosystems remain underexplored. The reconstruction of huge-phage genomes from distinct sampling sites allowed us to directly compare their genomes and revealed the importance of gene gain and loss in their evolution (Fig. 5). The expanded diversity of *whiB*-encoding actinophages suggested that the acquisition of multiple *whiB* genes is probably a persistent feature of several unrelated subclades (Supplementary Fig. 9).

The *cysC* and *cysH* genes are typically responsible for the assimilation of inorganic sulfate into organic compounds (for example, cysteine). Several *cysC*- and *cysH*-encoding viruses have been reported[43–45], yet their overall diversity and activity have yet to be fully understood. Here we expand evidence for virus-driven sulfur cycling[43,46,47] by showing a wide distribution of phages encoding *cysC* and *cysH* (Supplementary Table 7). These genes may play a role in bacterial sulfur metabolism during phage replication. Importantly, we show higher transcription of phage-encoded *cysC* and *cysH* compared with bacterial *cysC* and *cysH* genes in some samples (Fig. 6).

We note three limitations of the current version of COBRA. First, COBRA generally could not extend query contigs with very low sequencing coverage (Fig. 2a). A tool that automatically extends contigs using sequences from multiple related samples[26] might be developed to overcome this limitation. Second, COBRA runs relatively slowly if the corresponding community is complicated (indicated by 'total sharing ends'; Supplementary Table 9), for example, soil and underground water; all but one of our tested processings were finished within half an hour to 4 h. Third, unlike ContigExtender[27], COBRA directly uses the contigs generated by assemblers; thus, it cannot fix (or detect) chimeras that are introduced during assembly.

## Methods

### Simulated genomes for evaluation of contig breaking rules in de novo assembly

To evaluate how the assemblers of IDBA_UD, metaSPAdes and MEGAHIT will fragment the contigs during assembly in dealing with intra-genome repeats, inter-genome shared region and within-population variation (that is, local variation), we simulated the artificial genomes using Geneious Prime[48] for different cases that are described in detail in Supplementary Information. In each case, the artificial genomes were simulated for Illumina paired-end reads using InSilicoSeq with the 'HiSeq' error model, which generated paired-end reads in the length of 126 bp. The simulated reads were then assembled using IDBA_UD[49] ('mink = 20, maxk = 100, -step = 20, -pre_correction'), metaSPAdes version 3.15.149 ('-k 21,33,55,77,99') and MEGAHIT version 1.2.950 ('-k-list 21,29,39,59,79,99'). The obtained contigs from each assembly of each case were manually checked for breaking points and the possibilities of joining via their end sequences with a determined length (that is, 99 bp), which are shown in detail in Supplementary Information.

### Evaluation of contig breaking rules in de novo assembly using simulated genomes

To assess how IDBA_UD, metaSPAdes and MEGAHIT fragment contigs during assembly when confronted with intra-genome repeats, inter-genome shared regions and within-population variation (local variation), we generated artificial genomes using Geneious Prime[48]. Detailed descriptions of each case can be found in the Supplementary Information. For each case, artificial genomes were simulated for Illumina paired-end reads of 126 bp in length using InSilicoSeq[50] with the 'HiSeq' error model. Subsequently, the simulated reads were assembled using the following parameters: IDBA_UD[49]: 'mink = 20, maxk = 100, -step = 20, -pre_correction', metaSPAdes[51] version 3.15.1 18: '-k 21,33,55,77,99' and MEGAHIT[52] version 1.2.9 19: '-k-list 21,29,39,59,79,99'. The resulting contigs from each assembly in each case were manually inspected for breaking points and the potential for joining via their end sequences, with a specified length of 99 bp. Further details can be found in Supplementary Information.

## Benchmark COBRA using a previously published ocean virome dataset

Three virome datasets with samples collected from different depths (that is, 25 m, 117 m and 250 m) were reported previously. The authors sequenced the extracted DNA using both Illumina paired-end reads (150 bp in length) and also Nanopore single-molecule reads[32]. With these reads, the authors detected and polished complete viral genomes, mostly from the sample collected at 250 m. This dataset was used to benchmark the performance of COBRA. To evaluate the performance of COBRA, the raw reads of the 250 m sample were downloaded from NCBI and trimmed using https://github.com/najoshi/sickle using default parameters to remove low-quality bases. The adaptor sequence and other contaminants were detected and excluded using bbmap (https://sourceforge.net/projects/bbmap/). The trimmed reads were assembled using IDBA_UD[49] ('mink = 20, maxk = 140, -step = 20, -pre_correction'), metaSPAdes version 3.15.149 ('-k 21,33,55,77,99, 127') and MEGAHIT version 1.2.9 (ref. [52]) ('-k-list 21,29,39,59,79,99,119,141'). For each assembly, the contigs with a minimum length of 10 kb were compared against the polished viral genomes reported in a previous study[32] using BLASTn; the hits with a minimum nucleotide similarity of 97% and minimum alignment length of 10 kb were retained as queries for COBRA analyses. The quality reads of each sample were respectively mapped to all the contigs of the corresponding sample using Bowtie2 version 2.3.5.1 with default parameters[53]. The sequencing coverage of the contigs was determined using the 'jgi_summarize_bam_contig_depths' function from MetaBAT version 2.12.135 and transferred to a two-column file using in-house Perl script. COBRA analyses were performed for BLASTn hits contigs from each assembler, with a mismatch of 2 for linkage of contigs spanned by paired-end reads; the maxK and assembler were flagged according to that used in assembly. The ANI analyses between COBRA sequences and polished genomes were performed by fastANI version 1.3 (ref. [54]), and the alignment fraction was calculated accordingly. The quality of viral genomes was evaluated by CheckV[14].

## Comparison of the performance of COBRA and binning tools

We compared the quality of sequences joined by COBRA to bins generated by various binning tools, namely MetaBAT 2 (ref. [35]), vRhyme[10] and CoCoNet[8]. Using the IDBA_UD-assembled contigs (≥2,500 bp) from the 250 m ocean virome sample, we searched for contigs that exhibited ≥99% nucleotide similarity and ≥80% alignment coverage with polished genomes, resulting in a set of 2,632 contigs termed 'query contigs'. These query contigs were extended using COBRA and also binned using the aforementioned binning tools. For coverage calculation, the quality reads from the virome samples (25 m, 117 m and 250 m) were mapped to the query contigs individually using Bowtie2 version 2.3.5.1 with default parameters[53]. The coverage profiles derived from all three mapping files were used as input for the three binning tools. However, COBRA used only the mapping file and coverage profile of the 250 m sample. Each bin contained a minimum of two contigs, and if a bin contained only one contig, it was assigned as 'unbinned'.

To evaluate the accuracy of COBRA joins and sequences represented by bins, we matched the joined contigs and bins back to the polished genomes. For the binning tools, if all the contigs from a given bin were best matched to the same polished genome, the bin was termed a 'good bin'. For COBRA, if all the contigs joined into a COBRA sequence were best matched to the same polished genome, the join was termed a 'good join'. If some of the contigs matched to one polished genome (genome a), and some others best matched to another one (genome b), when genome a and genome b shared ≥70% ANI (determined by fastANI version 1.3 (ref. [54])), the bin was termed 'problematic bin' (for those from binning tools), and the join as 'problematic join' (for those from COBRA). To determine the extent to which the 'problematic bin' or 'problematic join' was affected by contigs from related (sub)populations, we also compared the ANI of the corresponding matched polished genomes.

For a given bin or join, if some contigs matched to one polished genome (genome a), and some others best matched to another one (genome b), when genome a and genome b shared <70% ANI (determined by fastANI version 1.3 (ref. [54])), it was termed 'contaminated bin' or 'contaminated join', respectively. To determine the contamination rate of the 'contaminated bin' or 'contaminated join', we calculate the total length of the bin or join (Total_len), and also the total length of contigs best matched to each of the polished genomes, then picked up the polished genome match with the maximum total length of contigs (that is, Max_len). We calculated the contamination rate of the contaminated bin or join as below, in which 'Num_polished' is the total number of matched polished genomes. By doing so, the theoretical maximum contamination rate of a contaminated bin or join is normalized (that is, 100%).

$$((\text{Total\_len} - \text{Max\_len})/\text{Total\_len})/((\text{Num\_polished} - 1)/\text{Num\_polished})$$

For example, where the total length of the bin or join is 100 kb and the contigs are matched to two polished genomes, with one polished genome best matching contigs with a total length of 60 kb and the other polished genome matching contigs having a total length of 40 kb, Total_len = 100 kb, Max_len = 60 kb and Num_polished = 2; thus, the contamination rate = 80%.

## Benchmark COBRA using a composite soil metagenome dataset

To test whether COBRA could work on metagenomic datasets with higher complexity, we extracted soil viral genomes ≥10,000 bp in length and without any 'N' from IMG/VR v3 (ref. [55]). These genomes (53,381 in total) were clustered with ≥95% similarity, resulting in a total of 34,303 clusters, each of which was represented by a representative genome. We randomly picked (1) 500 representative genomes and added a direct terminal repeat (100–200 bp) to each of them (category 'with_DTRs'), (2) 500 representative genomes and randomly mutated 1% bases of each genome (category 'two_subpopulations_1p'), (3) 500 representative genomes and randomly mutated 3% bases of each genome (category 'two_subpopulations_3p'), (4) 500 representative genomes and randomly mutated 5% bases of each genome (category 'two_subpopulations_5p') and (5) 300 representative genomes, each genome was mutated with 3% bases, and also each genome mutated with 5% bases (category 'three_subpopulation'). For (2)–(5), the initial representative genome will be kept; thus, we have a total of 500 + 500 × 2 × 3 + 300 × 3 = 4,400 simulated genomes. Note that each representative genome was used only once.

The 4,400 genomes were in silico sequenced using paired-end (126 bp in length) Illumina HiSeq using InSilicoSeq[50], with a random sequencing coverage of 10–100 for each genome, which generated ~20 Gb reads. To raise the complexity, the simulated reads were combined with ~12 Gb paired-end reads from a natural soil sample from California, USA[56]. This composite dataset was assembled using metaSPAdes with the kmer set of '21,33,55,77,99' with 48 threads.

The scaffolds ≥2,500 bp were searched against the 4,400 simulated genomes using BLASTn, and those hits with ≥99% similarity and ≥97% alignment length were retained as queries for subsequent COBRA analyses. To evaluate the performance of binning tools (MetaBAT 2, vRhyme and CoCoNet) on the same set of query scaffolds, we in silico sequenced another two sets of paired-end reads from the 4,400 simulated genomes, so the binning tools could have a coverage profile from at least three samples for better performance. The mapping of reads to all assembled contigs and the calculation of coverage were performed as described above. The 'extended_circular' and 'extended_partial' COBRA sequences, and the bins (with at least two scaffolds), were compared against their corresponding simulated genomes for accuracy evaluation. A given COBRA sequence or bin was assigned as 'a good join or bin' if all the scaffolds were from a single simulated genome,

as 'a problematic join or bin' if some scaffolds were from a simulated genome while some others were from the mutated genome, and as 'a contaminated join or bin' otherwise.

### Comparison of COBRA and ContigExtender

ContigExtender is a tool to improve the viral sequences assembled from metagenomic datasets using the reads. It uses a novel recursive extending strategy that explores multiple extending paths to extend the contigs. ContigExtender analysis was performed on the same contig set as used in Comparison of the Performance of COBRA and Binning Tools (2,632 in total); however, given the long processing time (~8 days to extend 29 contigs using 16 threads), we included only the results of the first 29 contigs in our comparison against COBRA. We summarized the extending results of the corresponding contigs by COBRA, and compared the performance of both tools, including the extended length and also the extending accuracy (Supplementary Table 4). The extended sequences from COBRA and ContigExtender were aligned against the corresponding polished complete genomes in Geneious[48] and also compared using BLASTn, and manually checked for extension accuracy.

### Collection and analyses of published freshwater metagenomic datasets

The freshwater metagenomic datasets from two previously published studies[16,57] were used. The raw paired reads were downloaded from NCBI using sratoolkit.2.11.1 (https://hpc.nih.gov/apps/sratoolkit.html) and filtered to remove any low-quality reads and bases, adaptors and other contaminants as described above. The de novo metagenomic assembly was first performed using the quality reads by IDBA_UD52 ('mink = 20, maxk = 140, -step = 20, -pre_correction') or metaSPAdes version 3.15.149 ('-k 21,33,55,77,99,127'). If the RAM of our computing server was not sufficient to assemble the reads of a given sample, it was assembled using MEGAHIT version 1.2.9 (ref. 52) ('-k-list 21,29,39,59,79,99,119,141'). If a given sample could not be assembled using any of the three assemblers, it was excluded from the analyses. The assembler details for each dataset are shown in Supplementary Table 5.

The generated contigs with a minimum length of 10 kb from each assembly were predicted for viral sequences using VIBRANT7 using default parameters. The identified lysogenic and lytic virus contigs by VIBRANT were used as queries for COBRA analyses. A max mismatch of 2 in each read was set to identify the linkage of contigs spanned by paired-end reads, and the minK, maxK and assembler were flagged according to what was used in assembly (Supplementary Table 5).

### Filtering of COBRA sequences and evaluation of assembly gaps

The COBRA sequences from all 231 freshwater metagenomic datasets were evaluated by CheckV (version 0.7.0)[14]. The 'self_circular' and 'extended_circular' COBRA genomes and those identified as 'high-quality' by CheckV were retained for further analyses. To evaluate and fix the assembly gaps, we checked the genomes by parsing the reads mapped to them (with Bowtie2 as described above) using a custom script named 'gap.check.py' (available at https://github.com/linxingchen/cobra). The script filtered the mapped reads to allow two mismatches for each read; for a region in a given genome sequence without any base mapped, the region was replaced by 10x Ns. The resulting sequences were used for further analyses.

### Genome completeness evaluation of the query contigs

To determine the extent to which COBRA raised the quality of the viral genomes, we evaluated the original query contigs that were joined into 'extended_partial' or 'extended_circular' genomes, using CheckV (version 0.7.0)[14]. The percentages of original contigs assigned by CheckV to 'Low-quality', 'Medium-quality', 'High-quality' and 'Not-determined' were profiled and shown in Fig. 5b.

### Clustering of quality viral sequences

The quality viral sequences were clustered at the species level using the rapid genome clustering approach provided by CheckV14 (available at https://bitbucket.org/berkeleylab/checkv/src/master/). The clustering parameters were set as follows: -perc_identity = 90 (for BLASTn), -min_ani = 95, -min_qcov = 10 and −min_tcov = 80 (for aniclust.py). The quality viral sequences were clustered into species-level clusters. Among these representative sequences, 6,430 had no assembly gaps, 815 had one gap, 195 had two gaps and 71 had three or more gaps. It is worth mentioning that any identified gaps were filled with 10x Ns during the clustering process.

### Identification of eukaryote viruses, virophages and phages

The protein-coding genes were predicted using Prodigal (-p meta)[58]. The eukaryote viruses were identified by searching the core structural protein sequences via BLASTp against the RefSeq database[59]. The virophage sequences were identified by searching their major capsid proteins (MCPs) against the virophage-specific HMM databases reported previously[37] using hmmsearch[60] version HMMER 3.3 ($-E = 1 \times 10^{-6}$). Those sequences with virophage-specific MCP hits were confirmed by building a tree with the MCPs from reference virophage sequences published previously[37,61].

### Taxonomy assignment of phages

To taxonomically assign the phages with genomes reconstructed in this study using the standardized ICTV taxonomy updated recently[62], we used PhaGCN213 (minimum score, 0.5) and geNomad[63]. The results from these two tools were considered; for a given genome, (1) it was assigned as 'unclassified' if both tools failed to assign it, or it was assigned to different taxa, and (2) it was assigned to the taxonomic level determined by one of the tools if the other failed to assign.

### Identification of new species phage genomes obtained in this study

The viral genomes from several published datasets were included for comparison, including the IMG/VR[64], the huge phages across ecosystems[19], the complete viral genomes from freshwater metagenomes[16], the pmoC phages[20] and the bS21 phages[65]; these genomes were termed 'viral_refs'. The 'viral_refs' genomes were first searched against our cluster representative genomes using BLASTn with a minimum e-value of $1 \times 10^{-50}$ and a minimum similarity of 90%. The BLASTn results were parsed to retain those with at least one hit with a minimum alignment length of 10,000 bp, and the corresponding genomes were extracted for genome clustering. If a given phage genome from the clusters could cluster with any of the 'viral_refs', it was labelled as 'reported' and as 'new species genome' otherwise.

### Huge-phage analyses

The subset of representative phage genomes with a minimum length of 200 kb were classified as huge phages. To include more huge-phage genomes from the freshwater datasets, we checked the low-quality and medium-quality genomes for huge phages and manually curated some of them. Protein-coding genes were predicted from them using Prodigal version 2.6.3 (-m -p meta)[58]. The predicted proteins were searched using BLASTp (e-value threshold = $1 \times 10^{-5}$) against the proteins of large terminase subunit (TerL), MCP, portal protein and prohead protein from the huge phages reported previously[19,23]. The BLASTp hits were confirmed using the online HMM search[66]. The confirmed proteins were individually aligned using MUSCLE (version 5.1.linux64)[67] and filtered to remove the columns accounting for >90% gaps using Trimal[68]. The filtered sequences for each genome were concatenated, and the phylogenetic tree was built using IQ-TREE version 1.6.12 (-bb = 1000, -m = LG + G4)[69].

To evaluate the abundance of each huge phage in each of the samples from Lake Rotsee (Fig. 6c), reads per kilobase per million

reads mapped (RPKM) were calculated as follows: RPKM = Nphage/(Lphage/1,000)/(Nsample/1,000,000), where Nphage is the number of reads to the phage genome, Lphage is the length of the phage genome (bp) and Nsample is the number of reads mapped to the whole metagenome-assembled contig set. The DNA read mapping to genomes or contigs was performed by Bowtie2 (version 2.3.5.1)[53] with default parameters excepting $-X = 2,000$, and filtered using the pysam Python module[70] to allow 0 or 1 mismatch for each mapped read. RPKM calculation of RNA reads to phage genomes was performed in the same way.

### Analyses of actinophages

We searched the phages infecting Actinobacteria (that is, actinophages), which are abundant in freshwater ecosystems, by searching for the *whiB* gene[71] via BLASTp search against NCBI RefSeq whiB protein sequences and by manual validation using the online HMM search tool (www.ebi.ac.uk/Tools/hmmer/search/). We determined the subset of the recovered genomes encoding *whiB* that have been reported previously[16]. A total of 4,288 (519 high-quality species genomes) from IMG/VR and 158 (79 species genomes) from ref. 16 were included in our analyses, along with 4,070 actinophage genomes (1,116 encode *whiB*) from 'The Actinobacteriophage Database' (https://phagesdb.org/). The entire set were clustered to identify distinct species genomes as described above (Clustering of quality viral sequences). The genes encoding TerL, MCP, portal protein and prohead protein were identified from each of the genomes. The sequences were individually aligned using MUSCLE[67] and filtered to remove the columns with >90% gaps using Trimal[68]. The concatenated sequences were used to reconstruct a phylogenetic tree to show the expansion of the *whiB*-encoding actinophage dataset via this study. The tree was built using IQ-TREE version 1.6.1269 with 1,000 bootstraps and the 'LG + G4' model.

### Transcriptional activity analyses

For the analysis of viral metabolic gene expression in situ, RNA reads obtained from Rotsee Lake samples were used. Metagenome-assembled contigs with a minimum length of 5 kb were examined using CheckV14 and VIBRANT7 to identify non-phage- and phage-encoded *cysC* and *cysH* genes. Only contigs with consistent predictions (either non-phage or phage) from both CheckV and VIBRANT were retained. The RNA reads from each sample were mapped to the corresponding contigs harbouring *cysC* and/or *cysH* genes. Subsequently, the transcriptional activity of each gene was normalized and summed separately for those encoded by non-phages and phages. The ratio of total transcriptional activity between non-phage and phage was calculated individually for *cysC* and *cysH* in each sample.

### Reporting summary

Further information on research design is available in the Nature Portfolio Reporting Summary linked to this article.

## Data availability

The high-quality and complete genomes obtained from the 231 freshwater metagenomes are available at figshare via https://figshare.com/articles/dataset/viral_genomes_fasta/23282789. The Actinobacteriophage database used in this study is available at https://phagesdb.org/. Source data are provided with this paper.

## Code availability

COBRA is available as an open-source Python program on GitHub (https://github.com/linxingchen/cobra), which could be installed via both PyPI and Conda.

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

## Acknowledgements

We thank R. Sachdeva for technical support. The study was supported by NSERC Canada and Syncrude Canada (grant number CRDPJ 403361-10); the Chan Zuckerberg Biohub and the Innovative Genomics Institute at the University of California, Berkeley; and the Genomic Science Program (GSP) LLNL 'Microbes Persist' Soil Microbiome Scientific Focus Area SCW1632 from the Office of Biological and Environmental Research, US Department of Energy (DOE).

## Author contributions

L.X.C. and J.F.B. conceived and designed the study. L.X.C. developed the tool; wrote the script; conducted simulated analyses, metagenomic assembly and viral contig identification; compared tools; and performed phylogenetic, genomic and RNA expression analyses. L.X.C. drafted the paper; both authors contributed to paper revisions.

## Competing interests

J.F.B. is a co-founder of Metagenomi. The other author declares no competing interests.

## Additional information

**Extended data** is available for this paper at https://doi.org/10.1038/s41564-023-01598-2.

**Correspondence and requests for materials** should be addressed to LinXing Chen or Jillian F. Banfield.

**a** one_path_end

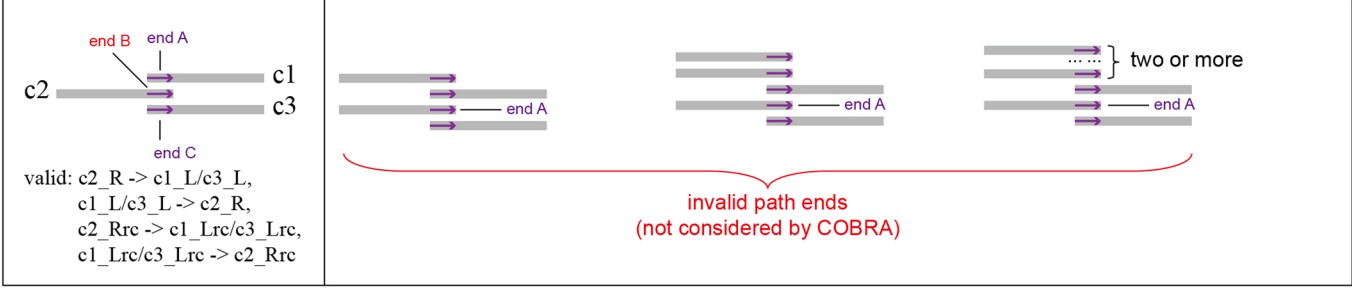

**b** two_paths_end

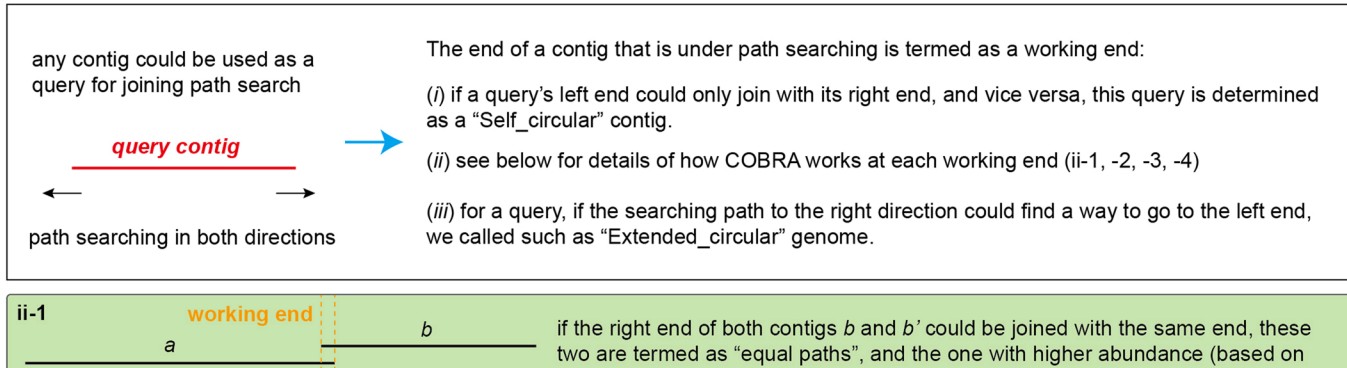

**c** COBRA processing mechanisms at each working end

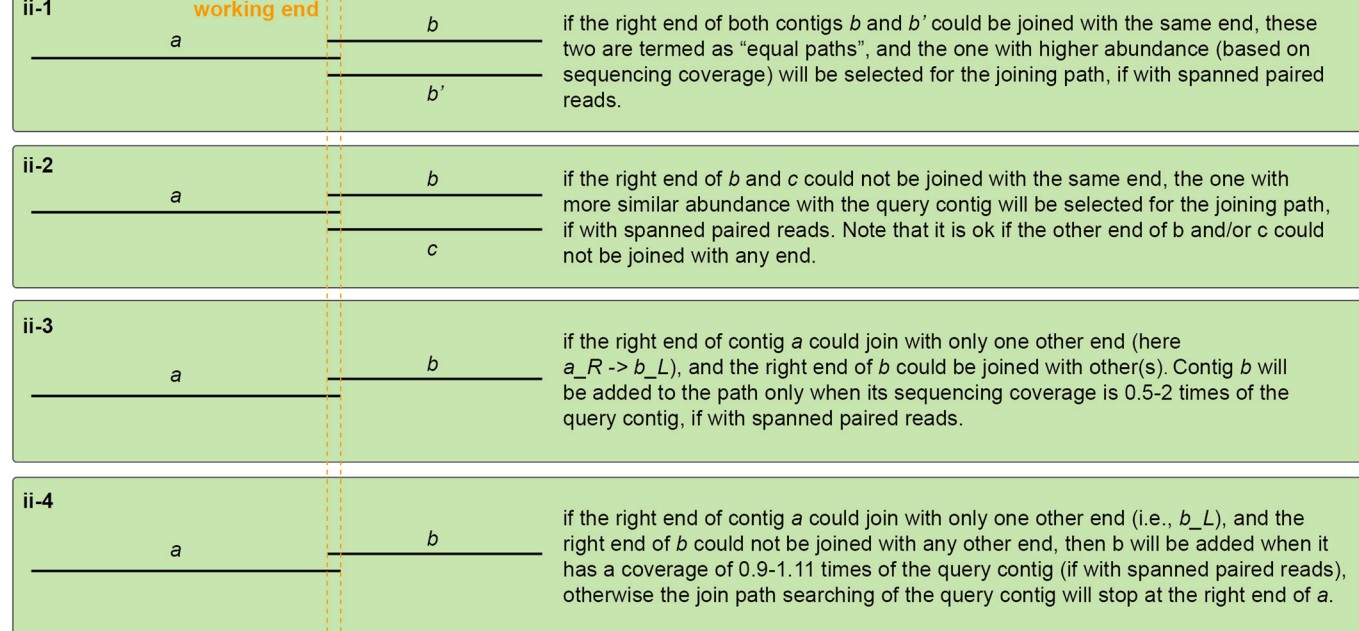

any contig could be used as a query for joining path search

*query contig*

path searching in both directions

The end of a contig that is under path searching is termed as a working end:

(*i*) if a query's left end could only join with its right end, and vice versa, this query is determined as a "Self_circular" contig.

(*ii*) see below for details of how COBRA works at each working end (ii-1, -2, -3, -4)

(*iii*) for a query, if the searching path to the right direction could find a way to go to the left end, we called such as "Extended_circular" genome.

**ii-1** if the right end of both contigs *b* and *b'* could be joined with the same end, these two are termed as "equal paths", and the one with higher abundance (based on sequencing coverage) will be selected for the joining path, if with spanned paired reads.

**ii-2** if the right end of *b* and *c* could not be joined with the same end, the one with more similar abundance with the query contig will be selected for the joining path, if with spanned paired reads. Note that it is ok if the other end of b and/or c could not be joined with any end.

**ii-3** if the right end of contig *a* could join with only one other end (here *a_R -> b_L*), and the right end of *b* could be joined with other(s). Contig *b* will be added to the path only when its sequencing coverage is 0.5-2 times of the query contig, if with spanned paired reads.

**ii-4** if the right end of contig *a* could join with only one other end (i.e., *b_L*), and the right end of *b* could not be joined with any other end, then b will be added when it has a coverage of 0.9-1.11 times of the query contig (if with spanned paired reads), otherwise the join path searching of the query contig will stop at the right end of *a*.

**Extended Data Fig. 1 | See next page for caption.**

**Extended Data Fig. 1 | Detailed working principles of COBRA. (a)** and **(b)** illustrate the valid joins considered by COBRA during the joining path search for each query contig, specifically in the scenarios of "one_path_end" and "two_paths_end" respectively. COBRA first identifies all contig end pairs with the same end sequences, considering both the end sequence and its reverse complement (rc). For example, the left end of contig 1 is referred to as contig_1_L, and the reverse complement as contig_1_Lrc. These identified pairs are then filtered to retain only those that could potentially be joined. For instance, joining the left end of one contig with the right end of another is possible if contig_1_L equals contig_2_R. In this case, the right end of contig 2 can be connected with the left end of contig 1, resulting in the combined contig 2 + contig 1. Joining the left end of one contig with the left end of another (for example, when contig_1_L equals contig_2_L) is not possible. However, if contig_1_L equals contig_2_Lrc, the contigs can be joined, resulting in the combination of rc(contig 2) + contig 1. In the diagrams, each gray bar represents a contig (for example, c1 = contig_1), with its right sequence end labeled as "c1_R". The term "rc" denotes reverse complementary, and the arrow represents the contig end sequence, which is of length maxK or maxK-1. In each case, the name of the subject end under extension is in red, while all valid joins (direction sensitive) are displayed below the corresponding diagram. The filtered pairs are then examined to identify valid path pairs. COBRA labels an end for which there is only one possible join as "one_path_end". For example, end A of contig 1 shares a sequence with only one other contig end, that is, end B of contig 2 (subfigure (a), case 1). End B might share its sequence with just one other end, that is, end A. Alternatively, end B could share its sequence with two or more ends including end A (subfigure (a), cases 2–4). This could indicate that the region of end B occurs multiple times in the genome or is present in two or more different genomes. COBRA labels an end as "two_paths_end" if end B shares its sequence with two other ends (end A and end C), and ends A and C share the sequence exclusively with end B (subfigure (b)). Although this is equivalent to the reverse path for case 2 of 'one_path_end', it is considered separately because the end under consideration for extension is different. **(c)** A comprehensive overview of COBRA's working principles at each stage. In the case of "two_paths_end", the path with a closer coverage match to that of the "query contig" will be selected.

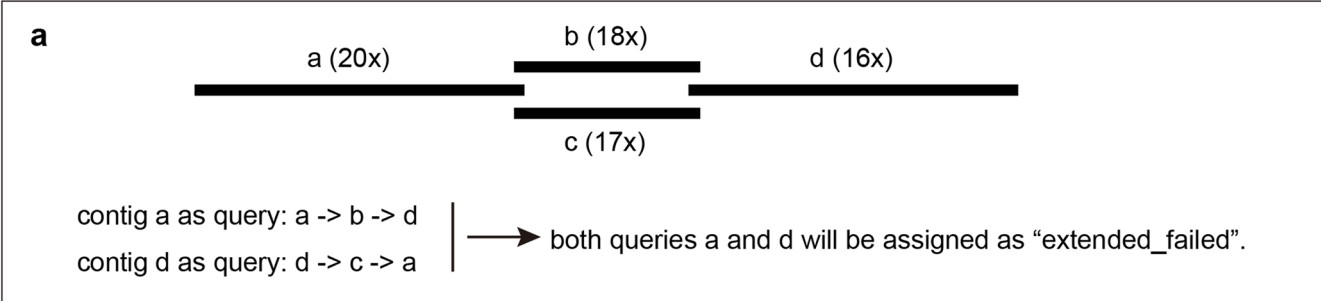

**a**

contig a as query: a -> b -> d

contig d as query: d -> c -> a

both queries a and d will be assigned as "extended_failed".

**b** query contig: "contig-140_18"

contig-140_18_extended_partial

These two parts are very similar, probably from related (sub)populations

to avoid such cases, COBRA divides the joined sequence into two parts

BLASTn comparison was performed

**BLASTn results:**

| qseqid | sseqid | sequence identity | length | mismatch | gapopen | qstart | qend | sstart | send | evalue | bitscore |
|---|---|---|---|---|---|---|---|---|---|---|---|
| contig-140_18_extended_partial_2 | contig-140_18_extended_partial_1 | 87.092 | 5609 | 703 | 10 | 43640 | 49236 | 6881 | 12480 | 0 | 6826 |

**Mauve alignment:**

If there is a region of ≥ 1000 bp that is shared between the two halves with ≥ 70% nucleotide identity, then the query contig(s) will be assigned to the "extended_failed" category (ii-c).

**Extended Data Fig. 2 | See next page for caption.**

**Extended Data Fig. 2 | The identification of non-unique joining paths. (a)** An important, but rare, case involves a query that can be extended along two (or more) seemingly unique paths. Both contig a and d are queries, however, different joining paths will be generated for them. The sequencing coverage of the contigs are shown in the brackets. (**b**)Screening of contigs joined from closely related genomes using BLASTn comparison. To prevent the joining of fragmented contigs from closely related (sub)populations, a BLASTn comparison is conducted between the first half and second half of each joined COBRA sequence. If the two parts share a region with a minimum length of 1000 bp and a minimum nucleotide similarity of 70%, all the query contigs involved in the join are labeled as "extended_failed" to indicate the failed extension.

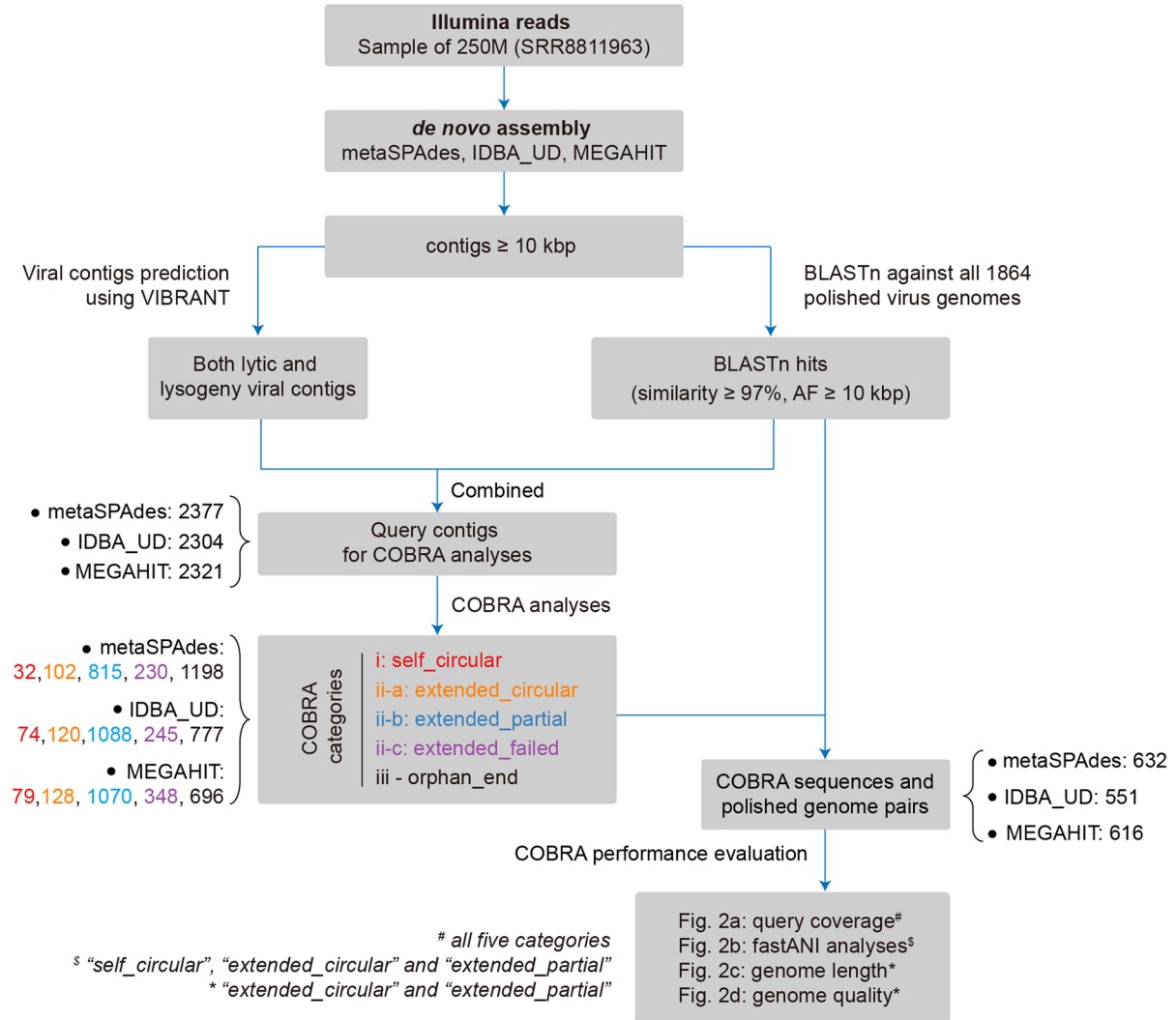

**Extended Data Fig. 3 | The pipeline to evaluate the performance of COBRA using a published virome sample.** The dataset was previously published by Beaulaurier et al. (2019). The raw Illumina reads were downloaded from NCBI, and quality control was conducted (see Methods in the main text). The quality paired-end reads were respectively assembled using metaSPAdes, IDBA_UD, and MEGAHIT. The assembled contigs with a minimum length of 10 kbp were extracted for viral prediction (with VIBRANT) and searched against the polished viral genomes (see Beaulaurier et al. 2019 for details) for viral contigs (with BLASTn, nucleotide similarity ≥ 97%, aligned fraction ≥ 10 kbp). The obtained viral contigs were combined as queries for COBRA analyses. The 'extended_circular' or 'extended_partial' COBRA sequences were compared with the corresponding polished genomes (determined by the abovementioned BLASTn search) to evaluate the performance of COBRA (see results in Fig. 2 of the main text).

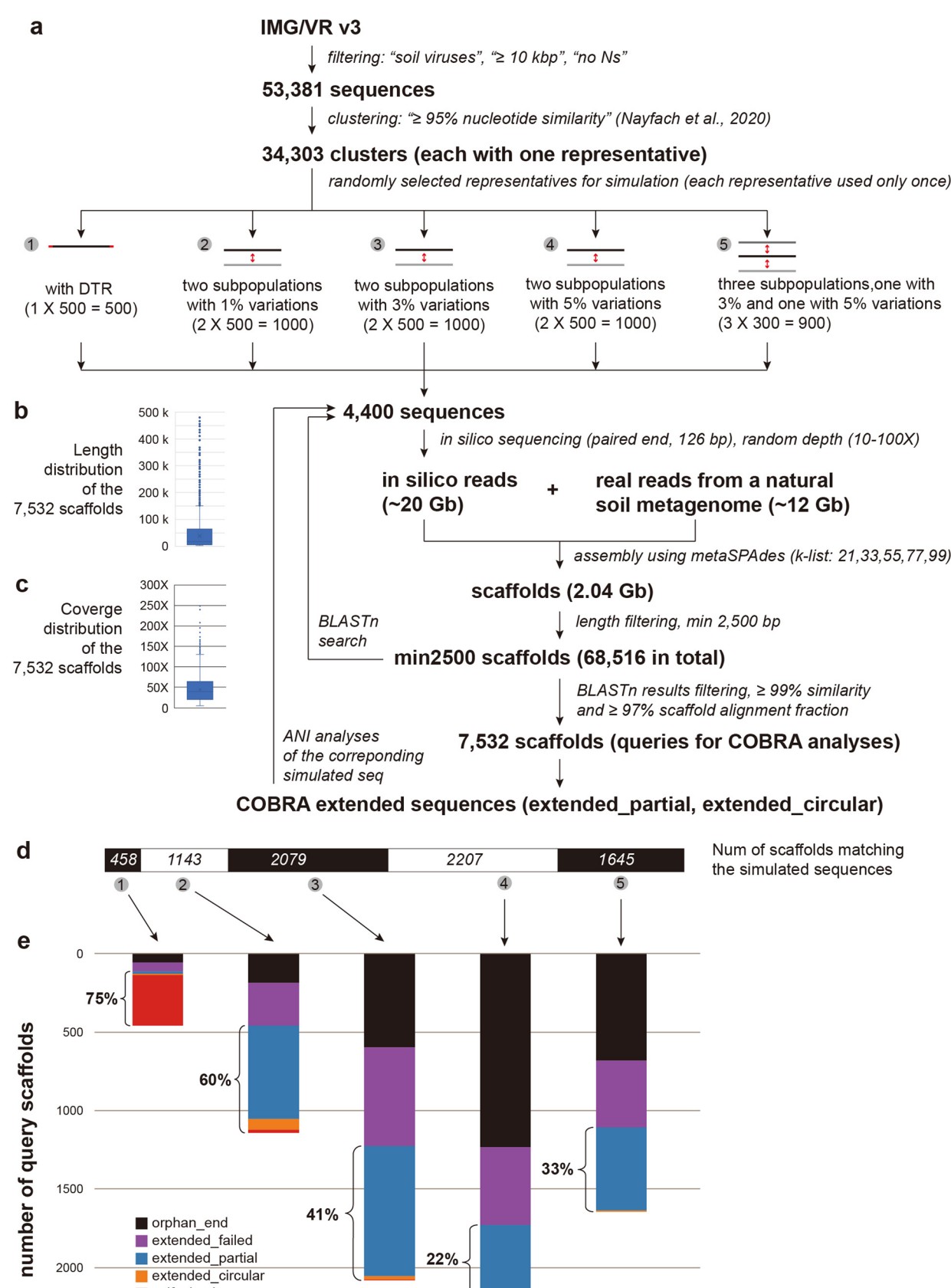

**Extended Data Fig. 4 | See next page for caption.**

**Extended Data Fig. 4 | Evaluation of COBRA with a composite soil metagenome dataset.** (**a**) The pipelines to obtain the soil viral simulated genomes from IMG/VR v3 for benchmarking. (**b**) The length distribution of the 7,532 query scaffolds for COBRA analyses. (**c**) The coverage distribution of the 7,532 query scaffolds for COBRA analyses. (**d**) The number of query scaffolds matching each category of the simulated genomes. (**e**) Bar plots showing the percentage of COBRA sequences for the query scaffolds matched each category of the simulated genomes. In the box plots, the centre lines, upper and lower bounds, and upper and lower whiskers show median values, 25th and 75th quantiles, and the largest and smallest non-outlier values, respectively. Outliers are defined as having a value >1.5 × interquartile range (IQR) away from the upper or lower bounds.

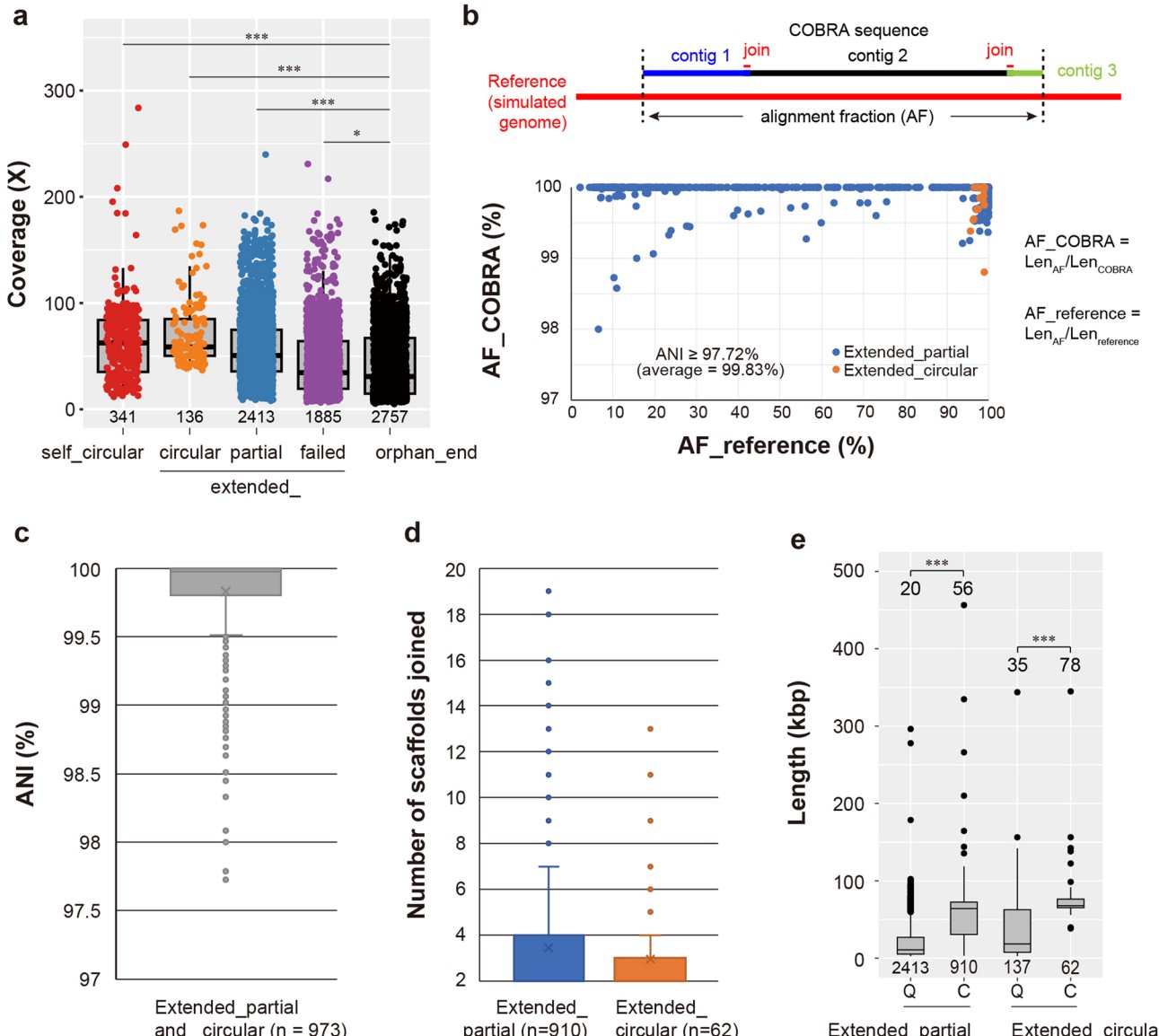

**Extended Data Fig. 5 | The performance of COBRA on a composite soil metagenome dataset.** (**a**) The sequencing coverage of query scaffolds in each COBRA output category. The average coverage between the 'orphan_end' category and others was compared using two-sided unpaired t-test (* p < 0.05, *** p < 0.001). (**b**) The profiles of the alignment fraction of COBRA sequences and the corresponding reference genomes. The definition and calculation of AF_reference and AF_COBRA are shown. (**c**) The ANI distribution of COBRA sequences and the corresponding reference genomes. (**d**) The number of scaffolds joined by COBRA to obtain 'extended_partial' (left) and 'extended_ circular' (right) sequences. (**e**) The length of the query scaffolds and the COBRA sequences of 'extended_partial' and 'extended_circular'. The average length of raw contigs and COBRA sequences are shown and compared using two-sided unpaired t-test (*** p < 0.001). In the box plots, the centre lines, upper and lower bounds, and upper and lower whiskers show median values, 25th and 75th quantiles, and the largest and smallest non-outlier values, respectively. Outliers are defined as having a value >1.5 × interquartile range (IQR) away from the upper or lower bounds.

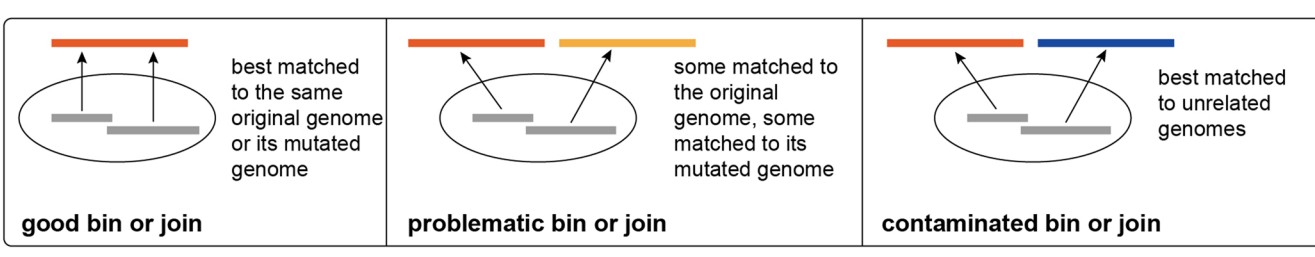

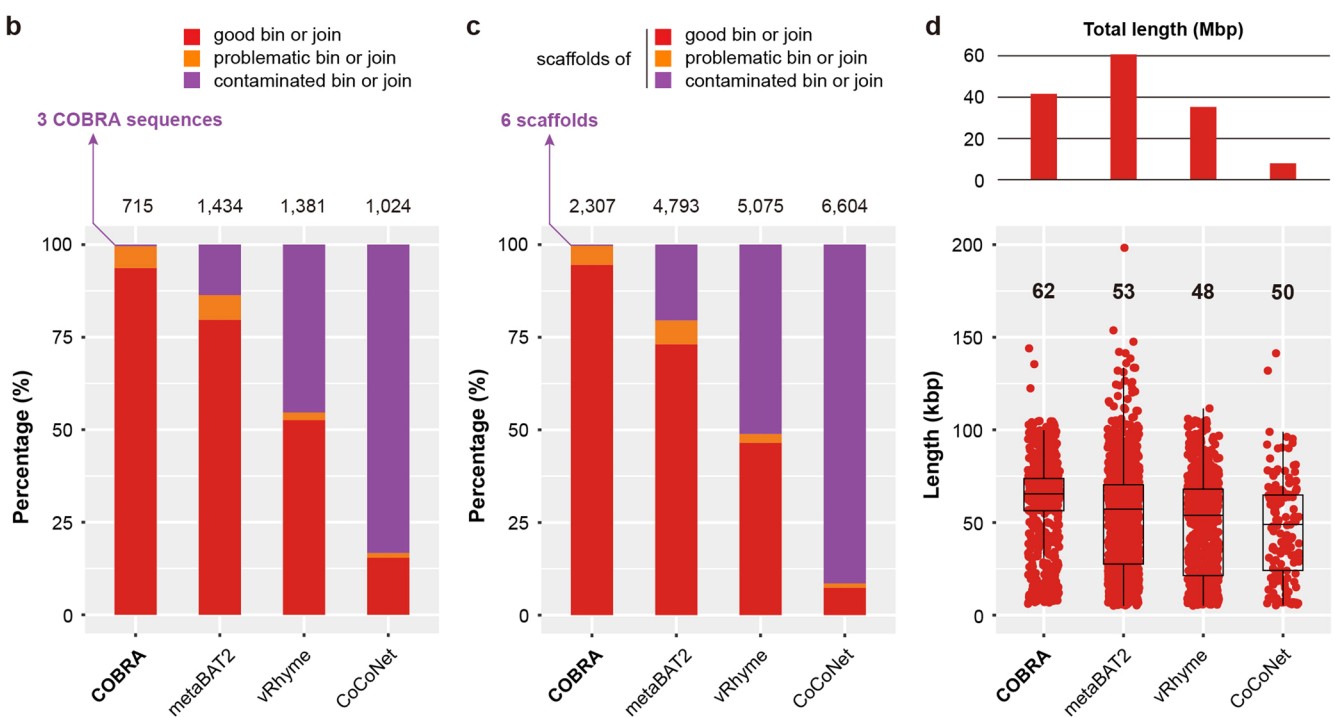

**Extended Data Fig. 6 | Performance comparison of COBRA and widely used binning tools on a composite soil metagenomic dataset.** (**a**) The flowchart shows the definitions of 'good', 'problematic', and 'contaminated' bin or join. (**b**) The percentage of 'good', 'problematic', and 'contaminated' bins or joins. (**c**) The percentage of scaffolds in 'good', 'problematic', and 'contaminated' bins or joins. In (b) and (c), the total absolute numbers are shown at the top. For bins and joins, only those with at least two scaffolds binned or joined were considered and compared. (**d**) The individual length of good bins and good joins, and their total length is shown at the top. In the box plots, the centre lines, upper and lower bounds, and upper and lower whiskers show median values, 25th and 75th quantiles, and the largest and smallest non-outlier values, respectively.

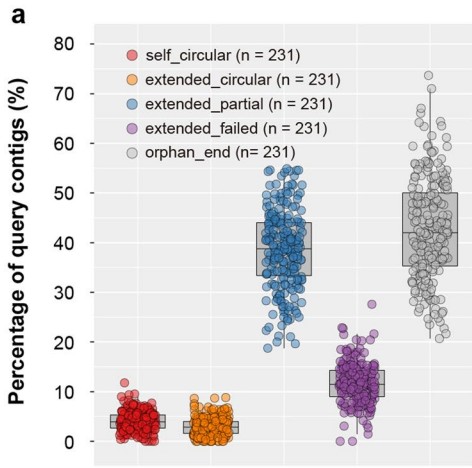

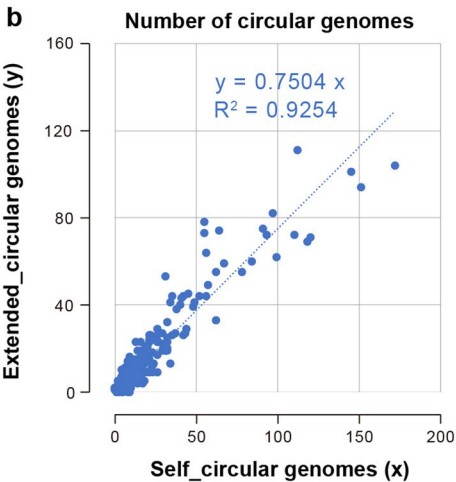

**Extended Data Fig. 7 | COBRA analyses of 231 freshwater metagenomes. (a)** The percentage of queries in each COBRA category. In the box plots, the centre lines, upper and lower bounds, and upper and lower whiskers show median values, 25th and 75th quantiles, and the largest and smallest non-outlier values, respectively. **(b)** The comparison of the number of 'self_circular' and unique 'extended_circular' genomes from freshwater metagenomes. Interestingly, the number of 'self_circular' genomes is highly related to that of unique 'extended_circular' genomes.

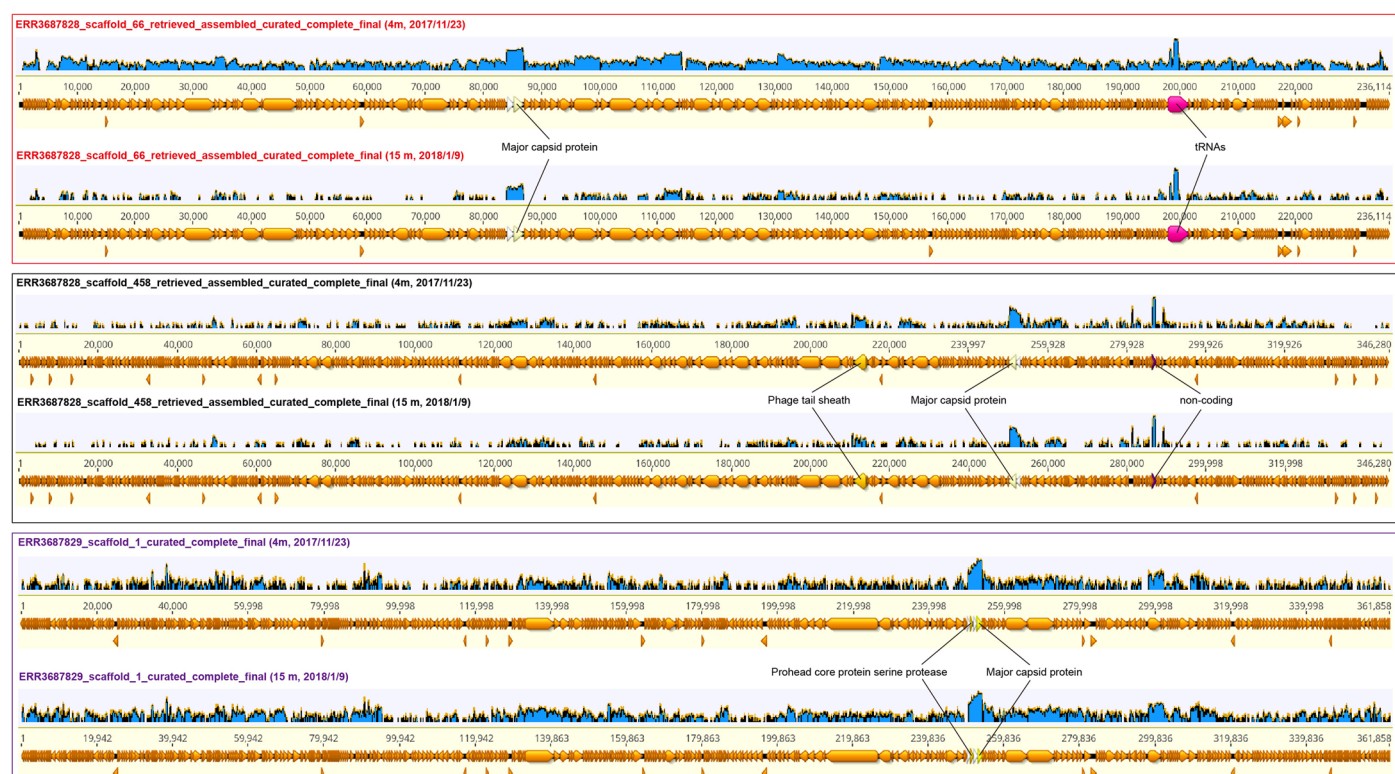

**Extended Data Fig. 8 | Examples of the in situ gene expression of huge phages with genomes reconstructed from Lake Rotsee.** The reads from different metatranscriptomic samples were mapped to each of the three genomes using Bowtie2 with default parameters. Then the bam files were imported to Geneious and remapped allowing no mismatch for each read. The huge phage genome names, sampling depths, and sampling time points are shown. The most highly transcribed ones are highlighted with their annotations.

(1) SRR6754100_NODE_27 (Rimov, Czech, Europe) v.s. (2) ERR2814752_NODE_86 (Soyang, South Korea, Asia)

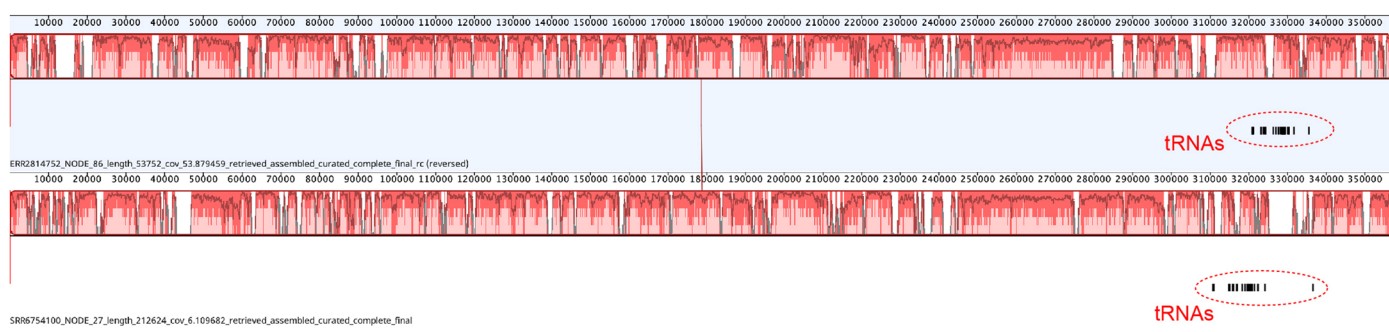

(3) SRR8894380_k141_5499707 (Fuxian, China, Asia) v.s. (4) ERR2814753_NODE_4 (Soyang, South Korea, Asia)

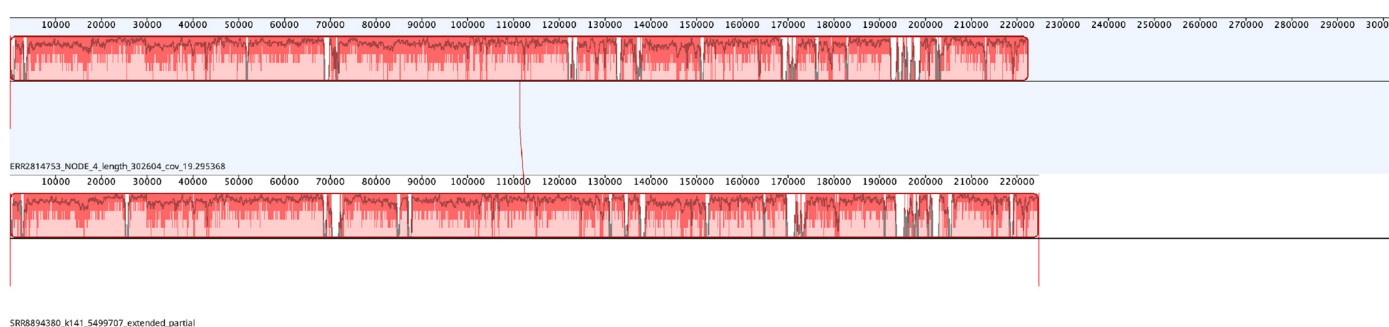

(5) ERR3687829_contig-140_2 (Lake Rotsee, Switzerland, Europe) v.s. (6) SRR9214415_k141_794410 (Jiřická pond, Czech, Europe)

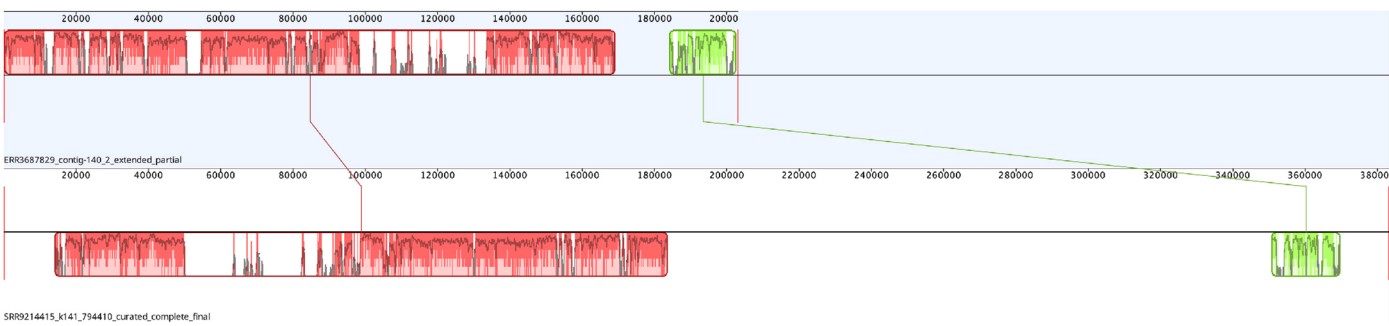

**Extended Data Fig. 9 | Mauve alignment of similar huge phage genomes from distant sampling sites.** The alignments of huge phage genomes were reconstructed from freshwater lakes of different countries. The tRNAs in the first pair, that is, genome (1) and (2) were highlighted with dashed circles.

# Reporting Summary

## Statistics

For all statistical analyses, confirm that the following items are present in the figure legend, table legend, main text, or Methods section.

| n/a | Confirmed | |
|---|---|---|
| ☐ | ☒ | The exact sample size ($n$) for each experimental group/condition, given as a discrete number and unit of measurement |
| ☒ | ☐ | A statement on whether measurements were taken from distinct samples or whether the same sample was measured repeatedly |
| ☐ | ☒ | The statistical test(s) used AND whether they are one- or two-sided<br>*Only common tests should be described solely by name; describe more complex techniques in the Methods section.* |
| ☒ | ☐ | A description of all covariates tested |
| ☒ | ☐ | A description of any assumptions or corrections, such as tests of normality and adjustment for multiple comparisons |
| ☐ | ☒ | A full description of the statistical parameters including central tendency (e.g. means) or other basic estimates (e.g. regression coefficient) AND variation (e.g. standard deviation) or associated estimates of uncertainty (e.g. confidence intervals) |
| ☐ | ☒ | For null hypothesis testing, the test statistic (e.g. $F$, $t$, $r$) with confidence intervals, effect sizes, degrees of freedom and $P$ value noted<br>*Give P values as exact values whenever suitable.* |
| ☒ | ☐ | For Bayesian analysis, information on the choice of priors and Markov chain Monte Carlo settings |
| ☒ | ☐ | For hierarchical and complex designs, identification of the appropriate level for tests and full reporting of outcomes |
| ☒ | ☐ | Estimates of effect sizes (e.g. Cohen's $d$, Pearson's $r$), indicating how they were calculated |

*Our web collection on statistics for biologists contains articles on many of the points above.*

## Software and code

Policy information about availability of computer code

| Data collection | sratoolkit.2.11.1 is used to download the SRA reads from NCBI. |
|---|---|
| Data analysis | metaSPAdes v3.15.5, IDBA_UD, MEGAHIT v1.2.9, Geneious Prime, InSilicoSeq, sickle version 1.33, bbmap Version 39.01, blastn: 2.14.0+, Bowtie2 version 2.3.5.1, MetaBAT version 2.12.1, fastANI version 1.3, checkV version 0.7.0, vRhyme Version 1.1.0, CoCoNet, VIBRANT v1.2.1, Prodigal V2.6.3, hmmsearch version HMMER 3.3, IQ-TREE version 1.6.12, MAFFT v7.453, trimAL v1.4.rev15, ContigExtender, genomad, MUSCLE version 5.1.linux64, COBRA version1.2.2 |

For manuscripts utilizing custom algorithms or software that are central to the research but not yet described in published literature, software must be made available to editors and reviewers. We strongly encourage code deposition in a community repository (e.g. GitHub). See the Nature Portfolio guidelines for submitting code & software for further information.

## Data

Policy information about availability of data

All manuscripts must include a data availability statement. This statement should provide the following information, where applicable:

- Accession codes, unique identifiers, or web links for publicly available datasets
- A description of any restrictions on data availability
- For clinical datasets or third party data, please ensure that the statement adheres to our policy

The high-quality and complete genomes obtained from the 231 freshwater metagenomes are available at figshare via https://figshare.com/articles/dataset/viral_genomes_fasta/23282789. The Actinobacteriophage database used in this study is available at https://phagesdb.org/.

## Research involving human participants, their data, or biological material

Policy information about studies with human participants or human data. See also policy information about sex, gender (identity/presentation), and sexual orientation and race, ethnicity and racism.

| | |
|---|---|
| Reporting on sex and gender | Not applicable. |
| Reporting on race, ethnicity, or other socially relevant groupings | Not applicable. |
| Population characteristics | Not applicable. |
| Recruitment | Not applicable. |
| Ethics oversight | Not applicable. |

Note that full information on the approval of the study protocol must also be provided in the manuscript.

# Field-specific reporting

Please select the one below that is the best fit for your research. If you are not sure, read the appropriate sections before making your selection.

☐ Life sciences  ☐ Behavioural & social sciences  ☒ Ecological, evolutionary & environmental sciences

For a reference copy of the document with all sections, see nature.com/documents/nr-reporting-summary-flat.pdf

# Ecological, evolutionary & environmental sciences study design

All studies must disclose on these points even when the disclosure is negative.

| | |
|---|---|
| Study description | This study develops a tool that identify the fragmentation points of de bruijn graph based short reads assemblers and join the contigs if some specific conditions are met. |
| Research sample | A total of 231 published freshwater metagenomes were included in analyses, for each of them, their paired reads were downloaded from NCBI, followed by quality control, de novo assembly, viral contig prediction and analyses by the tool developed in this study. |
| Sampling strategy | We analyzed 231 published freshwater metagenomes from different countries around the world, which is sufficient to test the performance of the tool developed in this study. |
| Data collection | na |
| Timing and spatial scale | na |
| Data exclusions | na |
| Reproducibility | na |
| Randomization | na |
| Blinding | na |

Did the study involve field work?  ☐ Yes  ☒ No

# Reporting for specific materials, systems and methods

We require information from authors about some types of materials, experimental systems and methods used in many studies. Here, indicate whether each material, system or method listed is relevant to your study. If you are not sure if a list item applies to your research, read the appropriate section before selecting a response.

## Materials & experimental systems

| n/a | Involved in the study |
|-----|-----------------------|
| ☒ | ☐ Antibodies |
| ☒ | ☐ Eukaryotic cell lines |
| ☒ | ☐ Palaeontology and archaeology |
| ☒ | ☐ Animals and other organisms |
| ☒ | ☐ Clinical data |
| ☒ | ☐ Dual use research of concern |
| ☒ | ☐ Plants |

## Methods

| n/a | Involved in the study |
|-----|-----------------------|
| ☒ | ☐ ChIP-seq |
| ☒ | ☐ Flow cytometry |
| ☒ | ☐ MRI-based neuroimaging |

