## [Peer Review File · Nature Microbiology]

Peer Review Information

Journal: Nature Microbiology

Manuscript Title: COBRA improves the quality of viral genomes assembled from metagenomes

Corresponding author name(s): Professor Jillian Banfield

Reviewer Comments & Decisions:

Decision Letter, initial version:

Message 26th July 2023

Dear Professor Banfield,

Thank you for your patience while your manuscript "COBRA improves the quality of viral genomes assembled from metagenomes" was under peer-review at Nature Microbiology. It has now been seen by 3 referees, whose expertise and comments you will find at the end of this email. Although they find your work of some potential interest, they have raised a number of concerns that will need to be addressed before we can consider publication of the work in Nature Microbiology.

In particular, the referees note that the method requires more careful benchmarking, the limitations of the approach should be mentioned more clearly, and the clarity of the text should be improved. Please note that editorially, we would need the referees' concerns to be addressed in full in order to further consider this study.

Should further experimental data allow you to address these criticisms, we would be happy to look at a revised manuscript.

Please include a data availability statement as a separate section after Methods but before references, under the heading "Data Availability". This section should inform readers about the availability of the data used to support the conclusions of your study. This information includes accession codes to public repositories (data banks for protein, DNA or RNA sequences, microarray, proteomics data etc...), references to source data published alongside the paper, unique identifiers such as URLs to data repository entries, or data set DOIs, and any other statement about data availability. At a minimum, you should include the following statement: "The data that support the findings of this study are available from the corresponding author upon request", mentioning any restrictions on availability. If DOIs are provided, we also strongly encourage including these in the Reference list (authors, title, publisher (repository name), identifier, year). For more guidance on how to write this section please see: <http://www.nature.com/authors/policies/data/data-availability-statements-data-citations.pdf>

* If you have not done so already we suggest that you begin to revise your manuscript so that it conforms to our Article format instructions at <http://www.nature.com/nmicrobiol/info/final-submission>. Refer also to any guidelines provided in this letter.

Note: This url links to your confidential homepage and associated information about manuscripts you may have submitted or be reviewing for us. If you wish to forward this e-mail to co-authors, please delete this link to your homepage first.

Nature Microbiology is committed to improving transparency in authorship. As part of our efforts in this direction, we are now requesting that all authors identified as 'corresponding author' on published papers create and link their Open Researcher and Contributor Identifier (ORCID) with their account on the Manuscript Tracking System (MTS), prior to acceptance. This applies to primary research papers only. ORCID helps the scientific community achieve unambiguous attribution of all scholarly contributions. You can create and link your ORCID from the home page of the MTS by clicking on 'Modify my Springer Nature account'. For more information please visit www.springernature.com/orcid.

If you wish to submit a suitably revised manuscript we would hope to receive it within 6 months. If you cannot send it within this time, please let us know. We will be happy to consider your revision, even if a similar study has been accepted for publication at Nature Microbiology or published elsewhere (up to a maximum of 6 months).

Reviewer Expertise:

Referee #1: Computational biology, bioinformatics, viral ecology

Referee #2: Viruses, bioinformatics

Referee #3: Metagenomics, viral ecology

Reviewer Comments:

Reviewer #1 (Remarks to the Author):

In this paper, Chen and Banfield describe COBRA, an approach to extend scaffolds generated from metagenomic assemblies. Using simulated data and benchmarking with some ocean viromes, the authors show that COBRA can generate a greater number and quality of viral scaffolds and genomes from metagenomes. Next, the authors extend this approach to all metagenomes and microbial genomes from a few datasets. Finally, the authors use COBRA on freshwater lake metagenomes to recover new phages and describe their diversity and ecology. Overall, this is an interesting take on the problem of recovering viral genomes and issues with genome binning.

However, this study needs more careful consideration of the implementation and benchmarking before it can be used widely. While the viral ecology part of the manuscript is generally well done, there are several flaws in the COBRA's approach.

In principle, COBRA's logic and workflow make sense but I have several reservations about the manner in which COBRA has been implemented and benchmarked. Additionally, I describe some specific flaws with the approach which could lead to outright wrong extensions. First, I think the authors are cherry-picking results and the benchmarking in my opinion is very poor and not up to the mark. Second, there are several outdated approaches as well as lack of sophistication that lead to several questions about the accuracy of this approach. Third, the description of viral diversity and ecology (especially taxonomy and other approaches) involve non-standardized approaches that are not easy to translate to other studies. I elaborate on these and other issues below.

Major concerns

1. My primary concern with COBRA is that it has not been benchmarked for lacking bias. In the results the authors clearly mention that "We acknowledge that these joins may not be legitimate, but rather serve as hypotheses that can be evaluated using information discussed below." There are two sources of potential bias that must be investigated:

Is this something that will work in ocean viromes which are not very complex but not in other environments? This should be benchmarked in a few diverse environments (soil and human gut come to mind). Soil is especially interesting since there can be a number of microbes and viruses at similar coverage due to its complexity. How will COBRA fare in view of this complexity?

2. I feel like there are some specific flaws in COBRA's workflow that the authors ignore. For starters some of the options presented are impractical and will certainly be used by the community in an inappropriate fashion.

- COBRA begins by processing all contigs/scaffolds from an assembly and looks for joins. This makes sense. But there seems to be an incomplete understanding and description of some of these processes. For example, it is posited that end A may only share a join with end B – but this seems to be impossible. As in why would the assembler break these scaffolds if there only one path forward and there clearly were reads that spanned this region (as presented by the join?). This is also shown in Supplementary Figure 6 and as presented seems to be an oversimplification.
- I would really implore the authors to consider the de bruijn graph itself rather than simply use scaffolds.
- A fundamental flaw in COBRA's workflow IMO is the provision for query contigs in step 2. Frankly I am confused by this. My understanding is that this query file will always be a subset of the entire scaffold file. If there are multiple paths to extend a contig, an easy but wrong solution is to provide a query dataset since an extension can be conducted but should not have been. I think COBRA should only extend all scaffolds for accuracy but then again I realize the caveats of this also being an issue for binning (A user may indeed choose to bin a subset of scaffolds instead of the entire dataset).

3. I agree with the choice of MetaSpades and Megahit but why use IDBA-UD? To my knowledge, this assembler generates erroneous scaffolds with serious issues in making contig joins and has not been widely used in the field for a few years now. I would not recommend using these analyses. Please consider switching to MetaSPADES scaffolds for all comparisons in the manuscript.

4. Contamination and accuracy: This entire benchmarking relies on a single dataset for which I could not find any information about the illumina reads. For starters, I am unable to determine if any amplification was involved that may throw off read coverage and binning (After all this is very common for illumina library prep such as with Nextera or TruSeq). The original manuscript focuses entirely on nanopore reads and only makes the illumina reads available but does not describe them. I would really suggest using additional datasets from other environments to get a sense of how good these numbers are, and specifically that these patterns of COBRA doing much better than binners is consistent.

5. There is no benchmarking on metagenomes whatsoever and the dataset used is for viruses/viromes. Without any benchmarking on microbial genomes (which can be more

complex in structure than viruses), I do not agree with the recommendation to broadly use this on metagenomes as well as microbial genomes. And if anything, asking users to use this on a subset of scaffolds is even more problematic.

6. If the authors want to suggest using COBRA on metagenomes and microbial genomes, I think it would be important to benchmark their analyses against microbial binners: Metabat2, Maxbin2, VAMB, Dastool to name a few.

7. No stats are provided on accuracy, F1 score, precision etc. I think using these metrics are important for evaluating performance.

8. While viral diversity and ecology analyses are generally well conducted, the taxonomy and some approaches used are very outdated. I would suggest using standardized ICTV taxonomy. For example, instead of terL trees, I would suggest using vContact2, gene networks, or atleast a concatenated protein tree.

9. For all taxonomy, I would suggest using ICTV's standardized taxonomy. The jumbo phage families described here are not recognized by ICTV to my knowledge.

Minor concerns

Change quarry -> query

Reviewer #2 (Remarks to the Author):

In "COBRA improves the quality of viral genomes assembled from metagenomes" the authors introduce and evaluate the COBRA tool, which improves metagenomic assemblies and allows better biological interpretation from sequencing data. Notably, the authors exemplify COBRA's potential by using it to retrieve a large amount of complete or near-complete viral genomes that were not originally assembled in a single contig. The algorithm underlying the tool is solid and represents an automatization of the manual genome curation process that the authors described in a previous work.

Undoubtedly, COBRA is a highly useful tool with widespread applicability within the field. The manuscript provides compelling illustrations of COBRA's practical application in the investigation of environmental viruses, a field that is hampered by the fragmentation of assembled genomes. However, some technical details require clearer explanation, and the methodology and results of the novelty analysis need further clarification. Additionally, improving the tool's installation process and usability would greatly benefit users.

Below, I list my main comments and suggestions.

- In my opinion, COBRA represents an important advancement to enable the precise discovery of metagenomic plasmids. At the present time, the research of environmental plasmids is largely precluded by the small number of circular sequences in metagenomes. Since plasmids present a gene content that is very similar to that of integrated elements, it is often difficult to distinguish between plasmid fragments and integrated genomic islands. COBRA enables researchers to increase the retrieval of circular sequences and improve the identification of plasmids.

Although the paper is focused on the application of COBRA for phage discovery, I wonder if it also allowed the retrieval of complete plasmids. I think it could be valuable to report the number of complete plasmids detected before and after processing the freshwater and groundwater metagenomes with COBRA.

- Have the authors performed some sort of analysis to estimate the false positive rate (i.e. the fraction of circularizations that are incorrect). I understand that this analysis might be hindered by the lack of a ground truth, but it could be feasible with simulated data. An estimate, based on the authors' manual curation, would also work. Another option would be to look at clearly chromosomal circular sequences that are way below the expected length for a chromosome. I believe it is important that the readers get an idea whether they should expect false circularizations and how common they are.

- Regarding the novelty analysis presented in Figure 4, I think some clarification regarding the data and methodology are necessary. In panel E, the numbers of phage

species from other studies (horizontal bars) seem strangely low. All these studies have a higher number of species than shown here. From the Methods, it seems that the authors performed a filter prior to the clustering, but this is not immediately clear from the figure or the legend. Also, the figure says that the authors required a 10 kb alignment between their viruses and the references in order to include a given reference in the clustering, however the methods section states that the minimum alignment length was 5 kb. Which cutoff was used?

- How was the taxonomic assignment of viral genomes performed (Figures 4 and 7)? This is not described in the methods. Additionally, the Myoviridae, Podoviridae, and Siphoviridae families are not recognized by ICTV anymore, as these groupings are not supported by phylogenetic analysis. Can these phages be assigned to any families, orders, or classes of the current ICTV taxonomy?

- "For example, paths may differ due to a strain variant, 'contig 1-> contig 2a -> contig 3' and 'contig 1 -> contig 2b -> contig 3', where 2a and 2b are sequence variants. In this case, the same contigs (contig 1 and 3) have more than a single possible placement so no joins are made ('extended_failed', category ii-c)". Isn't this the situation (ii-1) in Supplementary Figure 6C? From that figure, I'd assume that the query would be expanded using either 2a or 2b, depending on their coverage.

- In Figure 1, the representation of the second case as a "self-circular" contig may be misleading since the depicted scaffold does not exhibit DTRs (that is, $c1_L \neq c1_R$). Why is circularity inferred in that case?

- In Figure 2, can the authors change the colors that were used for "extended" partial and "extended_failed"? I'm colorblind and I couldn't distinguish those. A safe bet is to use the Okabe-Ito color scheme. In panel F, I can't see the difference between "consistency" and "divergency".

- "Of the 7,334 phage species genomes identified, 167 were classified as huge phages (Fig. 3f)". Is the correct figure being referenced here? This panel does not show huge phages in any way.

- In Figure 7, non-phage sequences are referred to as "Bacteria". However, no taxonomic assignment was performed and some genes could be encoded by other taxa (e.g. Archaea). It would be more precise to refer to those genes as "non-phage" or something similar.

- Could the authors include a benchmark of speed and memory usage to give readers an expectation of how fast it is?

Minor comments:

- It is important to acknowledge the existence of other tools that aim to improve the circularization of metagenomes (e.g. <https://github.com/lmlui/Jorg>).

- "(1) the length of the overlap is $\max K$ (for metaSPAdes and MEGAHIT) or $\max K-1$ (for IDBA_UD), or (2) the length of the overlap is at least the minimum kmer length used in the assembly. Next, COBRA searches for potential joining paths for each end based on valid path pairs (either "one_path_end" and "two_paths_end")" Case (2) should include all instances of (1), since the minimum required DTR length is lower. Why are they listed separately?

- "Although this is equivalent to the reverse path for the case 2 of 'one_path_end', it is considered separately because the end under consideration for extension depends on the direction that COBRA is progressing through the contig set". Does this mean that you might get different results depending on the order the contigs are processed? For instance:

A_R → B_L

A_R → C_L

B_Rrc → A_Lrc

C_Rrc → A_Lrc

Assuming all joins are valid (two_paths_end), if you start processing from A, you could get (A+B, C). If you start processing from C, you'd get (C+A, B). If that's indeed the

case, please clarify in the text.

- In Figure 1 the acronym "DTR" is used, however nowhere in the text it is mentioned that DTR means direct-terminal repeat, which could confuse readers that are not familiar with the term.

- COBRA is written as "CORBA" multiple times in the manuscript.

- "The short Illumina reads of the ocean virome 250M sample"
"250M" should be replaced by "250 m", since the "m" stands for meters.

- "blastn" and "BLASTn" are being used interchangeably.

- In Figure 2, the acronyms are confusing. For example, "C" means "COBRA" in panel D and "extended_circular" in panel E.

- In Figure 2, I recommend replacing "AF similarity" with "ANI" to keep consistency with the rest of the manuscript.

- In Figure 4 panel E, the "References" text on top of the box "7,334 species genomes" make it seem that the 7,334 species are from the references, when they are actually from this study.

- Which version of IMG/VR was used in the novelty analysis? The citation is for version 2, but it is not clear in the text if that was indeed the version the authors used.

- Is there a reason for some genes being italicized (e.g. *cysC*) and some not (e.g. *whiB*, *pmoC*, *TerL*)?

- The "gap.check.py" script is not publicly available. Make sure to provide it before publication.

Considering that the paper aims to describe a tool that is publicly available, I have conducted a test on COBRA and noted down my primary recommendations below. I firmly believe that COBRA holds the potential to become a widely-used tool. However, its current implementation is not friendly to a broad audience and could be easily improved to enhance adoption by the community. The authors should feel free to not implement those suggestions, as these won't affect the overall quality of the manuscript.

- The current step-by-step assumes a lot from the user and doesn't make several important details clear, for instance: (1) to compute the coverage, the BAM/SAM files need to be sorted beforehand (an information that is not mentioned in the guide), so it could be useful to have an example showing how to produce the BAM/SAM and sort it (using Bowtie2 and samtools, for example); (2) the guide says that the user should process the coverage file to produce a two-column file, but it doesn't explain how this can be achieved.

- Right now, COBRA is not distributed in any package manager (Conda and/or PyPI), which is expected of bioinformatics tools nowadays. The process to install COBRA is very manual (install the correct Python requirements, external bioinformatics tools, clone the repository, etc.) and prevents it from being easily incorporated into pipelines.

- As a consequence of the lack of distribution through a package manager, COBRA is not in the user PATH and needs to be executed as a script ("python cobra.py" instead of just "cobra"), preventing it from being available as a central installation in servers.

- The execution log is not shown on the screen. Instead, the script prints the name of some of the input contigs without any context.

- Is there a reason that the query contigs need to be provided as a FASTA file if their sequences are already present in the whole assembly? It would be more user-friendly if the query could be provided as a .txt with a list of accessions. For big assemblies, storing a redundant FASTA file would be a waste of space.

- "jgi_summarize_bam_contig_depths" could be replaced with pyCoverM or CoverM.

These tools are much faster and eliminate the need for processing the coverage file to generate the required two-column format, as they can directly produce the desired format. By utilizing pyCoverM or CoverM

- Would it be possible to make the tool assembler-agnostic, as long as "mink" and "maxk" are provided? If the user uses an assembler other than MEGAHIT, IDBA_UD, or metaSPAdes, it is not immediately clear that they can use COBRA as long as they input the correct k-mer lengths and "lie" to the tool about the assembler they used.

Reviewer #3 (Remarks to the Author):

Key results

Your overview of the key messages of the study, in your own words, highlighting what you find significant or notable. Usually, this can be summarized in a short paragraph. The manuscript by Chen and Banfield describes COBRA, an informatics tool that extends contigs from metagenomic assemblies by leveraging contig overlap and sequencing coverage to resolve multiple suggested contig joins at de Bruijn graph-based assembler breakpoints. Contig end pairs with unique significant overlap and similar coverage are joined. Extending contigs is a beneficial step in a metagenomics pipeline because it enables identification of longer, more complete sequences, which in turn improves all downstream analyses such as comparative genomics and population genetics.

COBRA was optimized against simulated data (that included within genome repeats, regions shared across genomes, and within population sequence variation) and then benchmarked using an ocean virome dataset that had both short and long (nanopore) sequencing data available against modern assemblers and binning tools. Against the assemblers, COBRA was able to extend ~50% of the contigs and increased the average contig length from 20 to 32.5 kbp and increased the number of complete and high-quality viral genomes from 46 to 241 from the short-read data alone (the polished short- plus long-read datasets achieved 1,864 complete viral genomes). Against the binners, COBRA was much more accurate in combining sequences from the same viral genome, recovering 1.7 to 5.8 times more accurate viral bins, and each bin was 3-6 kbp longer using COBRA than other bins. COBRA also had extremely low bin contamination, with only 1 out of 400 of the bins being contaminated versus much worst numbers (111-261 contaminated bins) across the binning algorithms. Once optimized and benchmarked, COBRA was applied to 122K (>10kb) contigs from 231 published freshwater metagenomic datasets and filtered the results for CheckV-complete genomes, which resulted in 12,118 high-quality or complete viral genomes. These data were mined for some high-level inferences including expanding genomic representation for huge phages, assessing marker genes for transcripts, and uncovering Actinobacteria phages and phage-encoded AMGs of interest. There are tens of papers that could be written about this treasure trove of new sequences, and it will be of great value for the freshwater microbial and virus ecology community to be able to mine such data with domain specific knowledge and starting from such high-quality genomes.

Validity

Your evaluation of the validity and robustness of the data interpretation and conclusions. If you feel there are flaws that prohibit the manuscript's publication, please describe them in detail.

Overall, this study clearly demonstrates COBRA's ability to extend contigs and produce more complete viral genomes from virus-enriched (virome) datasets and whole metagenomes. Comparing COBRA constructed complete viral genomes to "polished" virus genomes (short- plus long-read hybrid assembled virus genomes) made it easy to see COBRA's accuracy. It was impressive to see how well it outperformed standard assembly-only and viral and non-viral binning tools.

However, several suggestions are as follows:

- 1) What is the run-time and computational power needed for COBRA on datasets of different sizes?
- 2) Is COBRA limited to reconstructing double-stranded DNA viruses, or can it be used for RNA viruses? ssDNA viruses?
- 3) How does COBRA compare against contig extension papers already out, such as ContigExtender (<https://bmcbioinformatics.biomedcentral.com/articles/10.1186/s12859-021-04038-2>)? Are there others?
- 4) Can the authors make clear, what are the current technical limitations of COBRA?

How can contig extension be further improved?

Significance

Your view on the potential significance of the conclusions for the field and related fields. If you think that other findings in the published literature compromise the manuscript's significance, please provide relevant references.

COBRA appears extremely useful for recovering more high-quality and complete viral genomes from short-read datasets, and the tool focuses in an area that is under-developed in viromics. However, the authors largely ignore the virus ecogenomics field. Though virus contig extension is a large hole, and one COBRA seems to handle well, there is a lot of the remaining pipeline that is years in the making and solid. Consider an early few sentences or paragraph that speaks to the advances the field has made in areas such as virus identification, taxonomic classification, automating AMG identification, estimating completeness, etc. These each represent large efforts in the field, but all will be made better by the larger genomic context that COBRA provides at the front-end of these advances.

As well, COBRA's comparison against assembly-only and binning tools is useful, but it is hard to evaluate its significance since the tool is not compared against existing contig extension methods (e.g. one explicitly for viral metagenomes, published in 2021 – ContigExtender

<https://bmcbioinformatics.biomedcentral.com/articles/10.1186/s12859-021-04038-2>).

The approach is different, so it is not clear how these two tools compare.

Data and methodology

Your assessment of the validity of the approach, the quality of the data, and the quality of presentation. We ask reviewers to assess all data, including those provided as supplementary information. If any aspect of the data is outside the scope of your expertise, please note this in your report or in the comments to the editor. We may, on a case-by-case basis, ask reviewers to check code provided by the authors (see this Nature editorial for more information).

Reviewers have the right to view the data and code that underlie the work if it would help in the evaluation, even if these have not been provided with the submission (see this Nature editorial). If essential data are not available, please contact the editor to obtain them before submitting the report.

The approach is valid: the accuracy of the contigs was acquired by comparing the COBRA-extended short-read contigs to complete genomes assembled from the same dataset. These genomes were taken from a set of published, manually curated, complete freshwater genomes. The methodology is very well detailed in the main text and supplemental making it very clear how COBRA works and how the study was carried out. Multiple metrics are reported such as the N50, longest contig recovered, average contig length, and total number of contigs, and reported for before and after contigs are passed through COBRA to easily demonstrate the improvement. Statistical significances are shown. Multiple widely-used assemblers were also used and compared showing that COBRA's behavior is comparable across assembler programs. Supplementary tables provide highly detailed metadata on COBRA and assembler performance, as well as all published data used the main and supplementary figures. Supplementary Table 3 and 4 for example include all features (columns) necessary to reproduce the freshwater genome findings from this study. All freshwater genomes are available via figshare at a link provided in the Data Availability Section, and all other data used is cited with the study it came from.

The data tables are all easily human readable, and the ones used for computing the figures all appear to be easily parsed by computers.

Analytical approach

Your assessment of the strength of the analytical approach, including the validity and comprehensiveness of any statistical tests. If any aspect of the analytical approach is outside the scope of your expertise, please note this in your report or in the comments to the editor.

Analytical approach of the benchmarking and validation is very solid. They show a lot of data proving COBRA creates longer contigs, and is highly accurate compared to other viral binning tools with statistical significance tests. They compare COBRA using multiple widely used assemblers as well showing it can be used with different assemblers. Data presentation makes it easy to see how the data quality improves with COBRA.

Suggested improvements

Your suggestions for additional experiments or data that could help strengthen the

work and make it suitable for publication in the journal. Suggestions should be limited to the present scope of the manuscript; that is, they should only include what can be reasonably addressed in a revision and exclude what would significantly change the scope of the work. The editor will assess all the suggestions received and provide additional guidance to the authors.

The biggest needs are:

1) Compare COBRA vs ContigExtender

a. ContigExtender addresses chimeric sequences, but there is no addressal of chimeric sequences from assembly in COBRA as far as I can tell. I guess the validation strategy accounts for false positive extensions, but explicit language in the discussion addressing the chimera issue and how COBRA could extend chimeras, creating longer, but still chimeric sequences (and therefore incorrect) would be good to do. COBRA may not cause errors, but it can't fix existing errors from the assembly process.

2) Add in compute times / resources required for COBRA

3) In the discussion section, address any limitations of COBRA and where contig extension could be improved, or how it could be integrated into existing assembly algorithms. Only 50% of the contigs from the validation could be extended—I'd be curious to know if there were any patterns to why the other 50% could not be extended, especially since these were pulled directly from the complete viral genomes and have a correct pairing.

4) How low coverage can COBRA operate at? What are the authors guidelines for what coverage should be trusted vs not to make contig extensions?

5) Bring Supp. Fig. 6 into the main manuscript and summarize the text describing it.

6) Could the authors have a paragraph that pragmatically walks the reader through the relative efforts / costs of the hybrid method of Beaulaurier et al 2020 vs COBRA+short-read methods? From one perspective, COBRA got 241 complete genomes only as compared to 1864 for the hybrid method, but at what cost?

Clarity and context

Your view on the clarity and accessibility of the text, and whether the results have been provided with sufficient context and consideration of previous work. Note that we are not asking for you to comment on language issues such as spelling or grammatical mistakes.

The paper is well written. Beyond the suggestions above, I would only suggest:

1) There is perhaps too much detail on how the method works in the results section. It could be summarized and put in context with supplementary figure 6 if it's brought into the main figure. Right now it's just too much text and hard to follow each case.

2) Consider moving the "Application of COBRA on whole metagenomes and non-viral microbial genomes" to before the huge phage section. This section is more similar to the viromes validation section and may flow better.

References

Your view on whether the manuscript references previous literature appropriately.

As described above, context of an early few sentences or paragraph that speaks to the advances the viromics field has already made in virus identification, taxonomic classification, automating AMG identification, estimating completeness, etc. These each represent large efforts in the field, but all will be made better by the larger genomic context that COBRA provides. As well, the comparison against the 2021 publication of ContigExtender.

Your expertise

Please indicate any particular part of the manuscript, data or analyses that you feel is outside the scope of your expertise, or that you were unable to assess fully.

I was able to review the ms fully.

Response to referees' comments

Below are our point-by-point responses to the Reviewers' comments, with the comments in black, and our response in blue. The modifications are highlighted in the main text of the revised manuscript. The references we cited in our responses are listed at the end of the responses to each referee.

Reviewer #1 (Remarks to the Author):

In this paper, Chen and Banfield describe COBRA, an approach to extend scaffolds generated from metagenomic assemblies. Using simulated data and benchmarking with some ocean viromes, the authors show that COBRA can generate a greater number and quality of viral scaffolds and genomes from metagenomes. Next, the authors extend this approach to all metagenomes and microbial genomes from a few datasets. Finally, the authors use COBRA on freshwater lake metagenomes to recover new phages and describe their diversity and ecology. Overall, this is an interesting take on the problem of recovering viral genomes and issues with genome binning.

We thank the Reviewers for their careful reading of our manuscript, and the positive comments.

However, this study needs more careful consideration of the implementation and benchmarking before it can be used widely. While the viral ecology part of the manuscript is generally well done, there are several flaws in the COBRA's approach. In principle, COBRA's logic and workflow make sense but I have several reservations about the manner in which COBRA has been implemented and benchmarked. Additionally, I describe some specific flaws with the approach which could lead to outright wrong extensions. First, I think the authors are cherry-picking results and the benchmarking in my opinion is very poor and not up to the mark. Second, there are several outdated approaches as well as lack of sophistication that lead to several questions about the accuracy of this approach. Third, the description of viral diversity and ecology (especially taxonomy and other approaches) involve non-standardized approaches that are not easy to translate to other studies. I elaborate on these and other issues below.

We thank the Reviewers for their comments and suggestions, and we will respond to them individually below.

Major concerns

1. My primary concern with COBRA is that it has not been benchmarked for lacking bias. In the results the authors clearly mention that "We acknowledge that these joins may not be legitimate, but rather serve as hypotheses that can be evaluated using information discussed below."

This sentence describes the importance of steps that follow the initial step of identifying end overlaps. The subsequent steps that we outline involve testing based on coverage and contig linkage (via paired-end reads mapping). To avoid any misunderstanding, we have modified the sentence to read:

"We acknowledge that these initially identified potential joins may not be legitimate, but subsequent steps that make use of additional information (see below) identify and remove inaccurate joins."

There are two sources of potential bias that must be investigated: Is this something that will work in ocean viromes which are not very complex but not in other environments? This should be benchmarked in a few diverse environments (soil and human gut come to mind). Soil is especially interesting since there can be a number of microbes and viruses at similar coverage due to its complexity. How will COBRA fare in view of this complexity?

We agree with the Reviewers that the soil and human gut environments may have more complex viromes, given that they have higher viral diversity than the ocean (Dion et al., 2020). For benchmarking there are some very specific requirements: a natural sample (not a mock one), sequenced with both short Illumina reads and long Nanopore sequences, and with complete viral genomes obtained from Nanopore sequences. We selected the ocean virome dataset because it is the only one we could find that met these criteria.

We knew that there were some human gut-related studies published with complete prokaryotic genomes. We did not select them for our analyses due to one or more of the following reasons, (1) the number of genomes is small, thus not sufficient for reliable evaluation, (2) virus genomes were not reported, (3) there were no Illumina paired-end reads.

Even now, there are no other suitable datasets that can be used for benchmarking. To address this reviewer's request for additional benchmarking, we developed a mock soil viral community dataset and used this to further evaluate COBRA.

Specifically, we extracted all the 53,381 soil viral genomes (minimum length, 10 kb, and without Ns) from IMG/VR v3 (Roux et al., 2021), and clustered them into 34,303 clusters at 95% similarity (Subfigure a below). Each cluster was represented by a representative sequence.

We then random extracted,

(1) 500 representatives and added a direct terminal repeat sequence to each of them with a random length between 100-200 bp (category "with_DTRs");

(2) 500 representatives and randomly mutated the sequences at a 1% rate to get another 500 sequences (category "two_subpopulations_1p");

(3) 500 representatives and randomly mutated the sequences at a 3% rate to get another 500 sequences (category "two_subpopulations_3p");

(4) 500 representatives and randomly mutated the sequences at a 5% rate to get another 500 sequences (category "two_subpopulations_5p"); and

(5) 300 representatives and randomly mutated the sequences at a 3% rate, and 5% rate to get another 600 sequences (category "three_subpopulations").

We thus have a total of 4,400 simulated genomes. We then *in silico* sequenced these genomes using the Hiseq model (paired-end reads, 126 bp in length) with a random coverage of each genome (10X-100X).

Because this simulated dataset appeared to be of lower than relevant complexity, we combined the *in silico* reads (~20 Gb) with ~12 G of paired-end reads from a complex natural soil microbiome from California, USA (Crits-Christoph et al., 2018). This combined dataset was assembled using metaSPAdes with the k-mer set of "21,33,55,77,99" using 48 threads. The assembly took a total of 32.3 hours, resulting in a total of 2,924,241 scaffolds (2.1 Gb). In comparison, the assembly of the ocean virome sample took metaSPAdes 8.2 hours using 16 threads. These statistics clearly indicate that the final simulated dataset is far more complex than the ocean virome dataset originally used for benchmarking.

The obtained scaffolds with a minimum length of 2,500 bp were compared against the 4,400 simulated genomes using BLASTn. The 7,532 hit scaffolds with a minimum nucleotide similarity of 99% and with at least 97% of its length aligned to the genome, were retained as queries for subsequent COBRA analyses. These query scaffolds had an average length of 36.8 kbp (2.5-480 kbp; Subfigure b below), and an average

depth of 49X (5-284X; **Subfigure c below**). Among these scaffolds, 458 matching genomes of “with_DTRs”, 1,143 matching genomes of “two_subpopulations_1p”, 2,079 matching genomes of “two_subpopulations_3p”, 2,207 matching genomes of “two_subpopulations_5p”, and 1,645 matching genomes of “three_subpopulations” (**Subfigure d below**). After application of COBRA to this query dataset, the program identified “self_circular” or join scaffolds to circular or partial genomes for 22-75% of the query scaffolds matching each of the five category simulated genomes (**subfigure e below**).

Similar to the pattern we observed from the ocean virome dataset, the “orphan_end” query scaffolds had significantly lower coverage than those in other COBRA categories (Subfigure a below). As a whole, COBRA extended 2,413 queries to 910 unique partial genomes and extended 136 queries to 62 unique circular genomes. The extended_circular and extended_partial sequences had average nucleotide identity of 99.83% to the corresponding simulated genomes and alignment fractions of $\geq 98\%$ (AF_COBRA), which collectively indicate the joins were accurate (Subfigures b and c below). COBRA joined up to 19 scaffolds together (Subfigure d below). The length of scaffolds increased from 20 to 56 kbp (for extended_partial genomes) and from 35 to 78 (for extended_circular genomes) due to COBRA analyses (Subfigure e below).

To compare the performance of COBRA against the binning tools, we evaluate the above extended sequences with at least two query scaffolds joined, and checked (1) if these query scaffolds matched best to the same genome (the same original genome or the same mutated genome), i.e., good joins, (2) if some query scaffolds matched best to the original genome, and some matched best to the corresponding mutated genome, i.e., problematic joins, or (3) the query scaffolds matched best to different unrelated genomes, i.e., contaminated joins (subfigure a below).

To bin the same 7,532 scaffolds using the binning tools, we simulated another two sets of reads from the 4,400 simulated genomes so the binners have coverage profiles from at least three samples to have a better performance. The two sets of simulated reads were individually mapped to all the assembled scaffolds for coverage calculation. The binning of the 7,532 scaffolds using the three binners was performed with the coverage from the three simulated read sets. The bins were evaluated for (1) “good bins”, (2) “problematic bins”, and (3) “contaminated bins” (subfigure a below). The data were summarized and shown in the figure below.

For this dataset, we found that COBRA also has the highest accuracy among all the tools, with 93.7% good joins, 5.9% problematic joins and only 0.4% contaminated joins, compared against 15.4-79.7% good bins, 1.3-6.7% problematic bins and 13.6-83.3% contaminated bins (Subfigure b below). For the three binners, metaBAT2 has the best performance, in agreement with what we observed on the ocean virome dataset. The good COBRA joined sequences have a longer average length (62 kbp) than the length of the good bins (48-53 kbp), though metaBAT2 has the largest total length of good bins (subfigure d below).

Please see our response below to another comment using five human gut metagenomes for benchmarking.

2. I feel like there are some specific flaws in COBRA's workflow that the authors ignore. For starters some of the options presented are impractical and will certainly be used by the community in an inappropriate fashion.

- COBRA begins by processing all contigs/scaffolds from an assembly and looks for joins. This makes sense. But there seems to be an incomplete understanding and description of some of these processes. For example, it is posited that end A may only share a join with end B – but this seems to be impossible. As in why would the assembler break these scaffolds if there only one path forward and there clearly were reads that spanned this region (as presented by the join?). This is also shown in Supplementary Figure 6 and as presented seems to be an oversimplification.

Regarding the impossibility of case 1 of “one_path_end”, where end A only shares a join with end B, and end B only shares a join with end A, we were also initially surprised to see that this occurs occasionally but consistently. We note that we have manually curated numerous genomes taking advantage of this phenomenon and provided evidence that the resulting complete genomes are correct (Chen et al. 2020).

As there is no output from an assembler that explains why decisions were made, it is not possible to provide an overall explanation. Clearly, some reads in the metagenome dataset provide information that disrupts the assembly process. In a few examples that we investigated manually, we find there is always a

single SNP supported by multiple reads exactly outside the contig overlap region. We determined that metaSPAdes was less prone to this problem, which makes sense given that it downplays the importance of SNPs in the assembly process (Nurk et al. 2017).

Below we show two randomly chosen examples from metaSPAdes ((1) and (2)), IDBA_UD ((3) and (4)), and MEGAHIT ((5) and (6)) assembly of the ocean virome 250 m sample. As we observed in the 6 examples, there is a SNP site exactly outside of the end overlap region. For examples (1), (3), (4), and (6), the base at the SNP position is different from that of the consensus sequence. In summary, we conclude that (1) it is possible that end A only joins with end B, and end B only joins with end A as well, and (2) the examples presented above indicate what we showed in Supplementary Figure 6 is not an oversimplification but fully supported by real data.

We hope that our explanations and the corresponding examples shown here have resolved the concerns raised by the Reviewers.

(1) NODE_8138_length_5040_cov_6.092204_Lrc and NODE_54191_length_1441_cov_5.841705_L

(2) NODE_17454_length_3049_cov_2.872005_L and NODE_17425_length_3053_cov_2.887560_Lrc

(3) contig-140_14875_Lrc and contig-140_19412_L

(4) contig-140_31188_L and contig-140_30704_Lrc

(5) k141_188548_L and k141_627012_Lrc

(6) k141_373711_L and k141_77854_Lrc

- I would really implore the authors to consider the de bruijn graph itself rather than simply use scaffolds.

COBRA was developed on a method that we have already benchmarked and documented by manual curation. We understand that the *de bruijn* graph itself should provide the potential linkage information between contigs/scaffolds, but some assemblers (for example, IDBA_UD, MEGAHIT, to our best knowledge) do not output such files directly. We anticipate that future developments of this approach may leverage this strategy.

- A fundamental flaw in COBRA's workflow IMO is the provision for query contigs in step 2. Frankly I am confused by this. My understanding is that this query file will always be a subset of the entire scaffold file. If there are multiple paths to extend a contig, an easy but wrong solution is to provide a query dataset since an extension can be conducted but should not have been. I think COBRA should only extend all scaffolds for accuracy but then again I realize the caveats of this also being an issue for binning (A user may indeed choose to bin a subset of scaffolds instead of the entire dataset).

The Reviewers are correct that users may only be interested in extending some of their contigs/scaffolds, for example, (1) viral contigs/scaffolds, (2) contigs/scaffolds with a minimum length (e.g., 2500 bp), (3) contigs/scaffolds with a minimum sequencing coverage of 20x. Another reason not to extend all contigs/scaffolds is about the time this takes for computation.

In the original manuscript, we noticed that a given query may be extended in multiple paths, thus COBRA will evaluate all these paths to ensure (1) they are the same unique path, or (2) all shorter paths are subsets of the longest path. If the paths of a given query meet neither (1) nor (2), all the involved queries in these paths will be assigned as “extended_failed”.

3. I agree with the choice of MetaSpades and Megahit but why use IDBA-UD? To my knowledge, this assembler generates erroneous scaffolds with serious issues in making contig joins and has not been widely used in the field for a few years now. I would not recommend using these analyses. Please consider switching to MetaSPADES scaffolds for all comparisons in the manuscript.

We agree - and disagree - with this point. All assemblers have strengths and weaknesses. Historically, we have found IDBA_UD makes longer scaffolds than metaSPAdes and this is very helpful for both binning and achieving complete genomes (after removing the errors, in part using ra2.py; Brown et al., 2015).

In the analyses in our original manuscript we used IDBA_UD contigs, not scaffolds. Thus, we retain the results presented in the comparative analyses (Figures 2 and 3). However, we have followed this Reviewer’s advice and only used metaSPAdes scaffolds in the new benchmarking.

4. Contamination and accuracy: This entire benchmarking relies on a single dataset for which I could not find any information about the illumina reads. For starters, I am unable to determine if any amplification was involved that may throw off read coverage and binning (After all this is very common for illumina library prep such as with Nextera or TruSeq). The original manuscript focuses entirely on nanopore reads and only makes the illumina reads available but does not describe them. I would really suggest using additional datasets from other environments to get a sense of how good these numbers are, and specifically that these patterns of COBRA doing much better than binners is consistent.

Regarding the information on the Illumina reads of the ocean virome samples, we found it in the Supplementary Material document of the original paper, and below is the screenshot of the Methods the authors used for sequencing. According to their descriptions, the amplification step was not mentioned.

Preparation of short-read libraries and Illumina sequencing

A total of 60 ng of genomic DNA from each sample was sheared to an average size of 350 bp using a Covaris M220 Focused-ultrasonicator (Covaris, Woburn, MA) with Micro AFA fiber tubes (Covaris, #520166, Woburn, MA). Libraries were sequenced using a 150 bp paired-end NextSeq High Output V2 reagent kit (Illumina, FC-404-2004, San Diego, CA). Illumina short reads were assembled into contigs in two steps. First, low quality sequence was removed using `iu-filter-quality-minoche` from the `illumina-utils` package (Eren et al. 2013). Second, remaining reads were assembled into contigs using the “meta-sensitive” mode of MEGAHIT (Li et al. 2015).

Regarding further benchmarking, please see the analyses described in our response to **Comment 1** above, and also in the revised manuscript.

5. There is no benchmarking on metagenomes whatsoever and the dataset used is for viruses/viromes. Without any benchmarking on microbial genomes (which can be more complex in structure than viruses),

I do not agree with the recommendation to broadly use this on metagenomes as well as microbial genomes. And if anything, asking users to use this on a subset of scaffolds is even more problematic.

We thank the Reviewers for the suggestion. Recently, Jin et al. (Jin et al., 2023) sequenced 180 human gut metagenomes with both Illumina reads and Nanopore long sequences, and reported 802 closed genomes. In response to this comment, we used the five samples with the most complete genomes, i.e., B38 (14 genomes, SRR17827854), B111 (10 genomes, SRR17827942), C64 (10 genomes, SRR17827895), C162 (9 genomes, SRR17827819) and A171 (8 genomes, SRR15910069), for additional benchmarking.

The paired-end reads (150 bp in length) of each sample were downloaded, filtered to remove contaminations and low-quality bases and reads, and assembled using metaSPAdes with the k-mer set of "21,33,55,77,99,127". The assembled scaffolds with a minimum length of 2,500 bp were searched against the closed genomes from the corresponding samples. Subsequently, for each sample, the scaffolds with $\geq 99\%$ similarity and $\geq 97\%$ alignment fraction with the closed genomes were used as queries for COBRA analyses (subfigure a below). We retrieved 2-108 scaffolds that matched each of the 51 closed genomes from the five samples (subfigure b below), these scaffolds with an average length of 62 kbp (2.5-1586.5 kbp) (subfigure c below). COBRA was able to join 27%-42% of the scaffolds (i.e., 57-187) from each of the samples (subfigure d below).

We then analyzed the COBRA results by comparing the extended COBRA sequences against the corresponding closed genomes using ANI analyses (subfigure e below). We calculated the ANI of the COBRA sequences against the corresponding closed genomes and also the alignment fraction of the COBRA sequences (i.e., AF_COBRA), the higher the ANI and the AF_COBRA are, the higher the accuracy is. Given that all the closed genomes are complete and with only one sequence for each genome, we do not need to care about their alignment fractions. The COBRA output sequences had an average ANI of 99.9% and an average AF_COBRA of 99.4%, and 12 of them had an AF_COBRA < 97%. We manually checked all these 12 COBRA sequences, and found that they had a relatively lower AF_COBRA because COBRA joined the scaffolds representing the subpopulation with higher relative abundance while the corresponding closed genome represented the subpopulation with lower relative abundance (see subfigure f below for one example).

We hope that the Reviewers will find this new analysis supports the reliability of COBRA analyses on whole metagenomes.

The methodological descriptions are added in the Methods section, please see details of the results in the revised manuscript.

6. If the authors want to suggest using COBRA on metagenomes and microbial genomes, I think it would be important to benchmark their analyses against microbial binners: Metabat2, Maxbin2, VAMB, Dastool to name a few.

For viral analyses (the main focus of this paper), we recommend extending the scaffolds but not using binning (which we show produces relatively poor results). For metagenome analyses, we recommend using COBRA to extend the contigs before binning because the longer the scaffolds, the more accurate the binning signals. In this scenario, evaluation of the subsequent binning outcomes is not relevant to

evaluation of COBRA. To address this comment, we now include these suggestions in the revised manuscript.

In our original manuscript, we used COBRA on the bins from the three groundwater metagenomes. We found there was one issue that the current version of COBRA could not handle, that is, if one contig from bin A and one contig from bin B was joined by COBRA, it will be difficult to determine which bin the the joined COBRA should be assigned. We thus removed this part from our manuscript.

7. No stats are provided on accuracy, F1 score, precision etc. I think using these metrics are important for evaluating performance.

The values are provided in Supplementary Table 3.

8. While viral diversity and ecology analyses are generally well conducted, the taxonomy and some approaches used are very outdated. I would suggest using standardized ICTV taxonomy. For example, instead of terL trees, I would suggest using vContact2, gene networks, or at least a concatenated protein tree.

We thank the Reviewers for their positive comments on our viral analyses. As suggested, we used ICTV taxonomy in the revised manuscript, including (1) all the 7,334 genome clusters (Figure 5 in the revised manuscript), (2) the huge phages (Figure 6 in the revised manuscript), and (3) the phages encoding *cysC* and/or *cysH* (Figure 8 in the revised manuscript). The assignment to the new ICTV standardized taxonomy was performed using PhaGCN2 (Jiang et al. 2023) and geNomad (Camargo et al., 2023). We did not use vContact2 as the relevant reference database has not been updated with the new ICTV taxonomy yet, to our best knowledge.

9. For all taxonomy, I would suggest using ICTV's standardized taxonomy. The jumbo phage families described here are not recognized by ICTV to my knowledge.

As suggested in this comment and **Comment 8**, we rebuilt the tree using a concatenated sequence of several core structural proteins (including TerL, major Capsid protein, portal protein, and prohead protein). We added the most recent ICTV's taxonomy information if available, which was analyzed using PhaGCN2 (Jiang et al. 2023) and geNomad (Camargo et al., 2023) with the new ICTV's standardized taxonomy.

Minor concerns

Change quarry -> query

"quary" in the original Figure 1 has been corrected to "query" as suggested.

Reference cited in the responses

Brown, Christopher T., Laura A. Hug, Brian C. Thomas, Itai Sharon, Cindy J. Castelle, Andrea Singh, Michael J. Wilkins, Kelly C. Wrighton, Kenneth H. Williams, and Jillian F. Banfield. "Unusual biology across a group comprising more than 15% of domain Bacteria." *Nature* 523, no. 7559 (2015): 208-211.

Chen, Lin-Xing, Karthik Anantharaman, Alon Shaiber, A. Murat Eren, and Jillian F. Banfield. "Accurate and complete genomes from metagenomes." *Genome research* 30, no. 3 (2020): 315-333.

Crits-Christoph, Alexander, Spencer Diamond, Cristina N. Butterfield, Brian C. Thomas, and Jillian F. Banfield. "Novel soil bacteria possess diverse genes for secondary metabolite biosynthesis." *Nature* 558, no. 7710 (2018): 440-444.

Jin, Hao, Keyu Quan, Qiuwen He, Lai-Yu Kwok, Teng Ma, Yalin Li, Feiyan Zhao, Lijun You, Heping Zhang, and Zhihong Sun. "A high-quality genome compendium of the human gut microbiome of inner Mongolians." *Nature Microbiology* 8, no. 1 (2023): 150-161.

Roux, Simon, David Páez-Espino, I-Min A. Chen, Krishna Palaniappan, Anna Ratner, Ken Chu, T. B. K. Reddy et al. "IMG/VR v3: an integrated ecological and evolutionary framework for interrogating genomes of uncultivated viruses." *Nucleic acids research* 49, no. D1 (2021): D764-D775.

Camargo, Antonio Pedro, Simon Roux, Frederik Schulz, Michal Babinski, Yan Xu, Bin Hu, Patrick SG Chain, Stephen Nayfach, and Nikos C. Kyrpides. "You can move, but you can't hide: identification of mobile genetic elements with geNomad." *bioRxiv* (2023): 2023-03.

Jiang, Jing-Zhe, Wen-Guang Yuan, Jiayu Shang, Ying-Hui Shi, Li-Ling Yang, Min Liu, Peng Zhu, Tao Jin, Yanni Sun, and Li-Hong Yuan. "Virus classification for viral genomic fragments using PhaGCN2." *Briefings in Bioinformatics* 24, no. 1 (2023): bbac505.

Almeida, Alexandre, Stephen Nayfach, Miguel Boland, Francesco Strozzi, Martin Beracochea, Zhou Jason Shi, Katherine S. Pollard et al. "A unified catalog of 204,938 reference genomes from the human gut microbiome." *Nature Biotechnology* 39, no. 1 (2021): 105-114.

Dion, Moïra B., Frank Oechslin, and Sylvain Moineau. "Phage diversity, genomics, and phylogeny." *Nature Reviews Microbiology* 18, no. 3 (2020): 125-138.

Nurk, Sergey, Dmitry Meleshko, Anton Korobeynikov, and Pavel A. Pevzner. "metaSPAdes: a new versatile metagenomic assembler." *Genome research* 27, no. 5 (2017): 824-834.

Reviewer #2 (Remarks to the Author):

In "COBRA improves the quality of viral genomes assembled from metagenomes" the authors introduce and evaluate the COBRA tool, which improves metagenomic assemblies and allows better biological interpretation from sequencing data. Notably, the authors exemplify COBRA's potential by using it to retrieve a large amount of complete or near-complete viral genomes that were not originally assembled in a single contig. The algorithm underlying the tool is solid and represent an automatization of the manual genome curation process that the authors described in a previous work.

We thank the Reviewers for their careful reading and overall positive attitudes toward our manuscript.

Undoubtedly, COBRA is a highly useful tool with widespread applicability within the field. The manuscript provides compelling illustrations of COBRA's practical application in the investigation of environmental viruses, a field that is hampered by the fragmentation of assembled genomes. However, some technical details require clearer explanation, and the methodology and results of the novelty analysis need further clarification. Additionally, improving the tool's installation process and usability would greatly benefit users.

We thank the Reviewers for the suggestions, we will respond to them individually below.

Below, I list my main comments and suggestions.

- In my opinion, COBRA represents an important advancement to enable the precise discovery of metagenomic plasmids. At the present time, the research of environmental plasmids is largely precluded by the small number of circular sequences in metagenomes. Since plasmids present a gene content that is very similar to that of integrated elements, it is often difficult to distinguish between plasmid fragments and integrated genomic islands. COBRA enables researchers to increase the retrieval of circular sequences and improve the identification of plasmids.

Although the paper is focused on the application of COBRA for phage discovery, I wonder if it also allowed the retrieval of complete plasmids. I think it could be valuable to report the number of complete plasmids detected before and after processing the freshwater and groundwater metagenomes with COBRA.

We understand that some users would be interested in using COBRA for complete plasmid genomes, and we agree with the Reviewers that COBRA will be a useful tool in the field. To be honest, we have another independent manuscript (in preparation) that will use the freshwater metagenomes for the discovery of complete plasmid genomes. Also, given that we only used predicted viral contigs as queries for extension from these freshwater samples in the current study, it would be better if we did not report the number of complete plasmid genomes here.

Regarding the groundwater metagenomes, we obtained a total of 223 circular genomes (including "self_circular" and "extended_circular"). We used geNomad (<https://github.com/apcamargo/genomad>) and found that 4 of them were plasmid genomes (180 were viral genomes). We added this information to the corresponding position of the Results section.

- Have the authors performed some sort of analysis to estimate the false positive rate (i.e. the fraction of circularizations that are incorrect). I understand that this analysis might be hindered by the lack of a ground truth, but it could be feasible with simulated data. An estimate, based on the authors' manual curation, would also work. Another option would be to look at clearly chromosomal circular sequences that are way below the expected length for a chromosome. I believe it is important that the readers get an idea whether they should expect false circularizations and how common they are.

We thank the Reviewers for this suggestion.

With the ocean virome dataset, we obtained 31-79 “self_circular” and 69-87 “extended_circular” genomes from the three different assemblers. Compared with the polished genomes, we found all the circularizations are correct, this was reflected in **Figure 2b** of the manuscript.

As suggested, and in combination with **Comment 1** from **Reviewer #1**, we evaluated the performance of COBRA using simulated genomes based on sequences from the IMG/VR v3 database (Roux et al. 2021), please see our response to **Comment 1** of **Reviewer #1**. These simulated genomes were *in silico* sequenced and the reads were assembled. The scaffolds matching the simulated genomes with at least 99% similarity and at least 97% alignment fraction (of the scaffold length) via BLASTn analyses, were used as queries for COBRA analyses. As a result, COBRA identified 341 “self_circular” genomes and 62 unique “extended_circular” genomes. We compared the length of these genomes against the corresponding simulated genomes (based on previous BLASTn analyses), and found all the circularizations are correct.

Given that both the real dataset and the simulated dataset suggested the circularizations were correct, we believe COBRA has reliable and high accuracy.

Nonetheless, we agree that false circularization is possible where COBRA is applied to whole metagenomes and the query sequences are not pre-selected to be putative viral fragment.

- Regarding the novelty analysis presented in Figure 4, I think some clarification regarding the data and methodology are necessary. In panel E, the numbers of phage species from other studies (horizontal bars) seem strangely low. All these studies have a higher number of species than shown here. From the Methods, it seems that the authors performed a filter prior to the clustering, but this is not immediately clear from the figure or the legend. Also, the figure says that the authors required a 10 kb alignment between their viruses and the references in order to include a given reference in the clustering, however the methods section states that the minimum alignment length was 5 kb. Which cutoff was used?

We thank the reviewer for pointing out this discrepancy. We modified the figure text legend to make it clear (current Figure 5). The minimum alignment is 10 kbp. This has been corrected in the methods section of the revised manuscript.

- How was the taxonomic assignment of viral genomes performed (Figures 4 and 7)? This is not described in the methods. Additionally, the Myoviridae, Podoviridae, and Siphoviridae families are not recognized by ICTV anymore, as these groupings are not supported by phylogenetic analysis. Can these phages be assigned to any families, orders, or classes of the current ICTV taxonomy?

The original taxonomic assignment of viral genomes in the original manuscript was performed using VPF-Class (Pons et al., 2021).

Given that the *Myoviridae*, *Podoviridae*, and *Siphoviridae* families are no longer recognized by ICTV, we applied PhaGCN2 (Jiang et al., 2023) and geNomad (Camargo et al., 2023) for analyses in the revised manuscript. The results are shown in (1) all the 7,334 genome clusters (Figure 5 in the revised manuscript), (2) the huge phages (Figure 6 in the revised manuscript), and (3) the phages encoding *cysC* and/or *cysH* (Figure 8 in the revised manuscript). The corresponding results were modified accordingly, and the methods descriptions were added in the Methods section.

- “For example, paths may differ due to a strain variant, ‘contig 1-> contig 2a -> contig 3’ and ‘contig 1 -> contig 2b -> contig 3’, where 2a and 2b are sequence variants. In this case, the same contigs (contig 1 and 3) have more than a single possible placement so no joins are made (‘extended_failed’, category ii-c)”. Isn’t this the situation (ii-1) in Supplementary Figure 6C? From that figure, I’d assume that the query would be expanded using either 2a or 2b, depending on their coverage.

In the example we described in the main text, we mean that (1) when contig 1 is the query, the path is 'contig 1-> contig 2a -> contig 3', and (2) when contig 3 is the query, the path is 'contig 1-> contig 2b -> contig 3'. This is possible if contig 1 has more similar coverage with contig 2a, while contig 3 has more similar coverage with contig 2b. In this situation, to avoid any risky joins, COBRA will assign both contig 1 and contig 3 into the category of "extended_failed". COBRA is very careful in making any potential joins, we listed the example in the main text to illustrate such cases though it should be rare.

Regarding Supplementary Figure 6c (ii-1, shown below), which showed the general rules of joining at each working end. Here contig a is the query contig, and a_R (right end of contig a) is the working end and it is a "two_paths_end", a_R could join with b_L (left end of B) and b'_L. We call b_L and b'_L the "equal paths" if their right end (i.e., b_R and b'_R) could join with the same (and only) end (termed as c_L here). In this situation, if the coverage of b is more similar to that of a, COBRA will join the contigs as "contig a -> contig b -> contig c".

- In Figure 1, the representation of the second case as a "self-circular" contig may be misleading since the depicted scaffold does not exhibit DTRs (that is, $c1_L \neq c1_R$). Why is circularity inferred in that case?

We thank the Reviewer for pointing out this. We had an inaccurate description in the original figure (see screenshot below), "the contig has no end overlap $\geq \text{mink}$ " should be "the contig has $\text{minK} \leq \text{end_overlap} < \text{maxK}(-1)$ ", which should be the length of the overlap is at least the minimum kmer length used in the assembly but does not equal the largest kmer.

original figure:

revised figure:

COBRA identifies "self_circular" (category i) contigs with two possible cases (see figures below),

(1) If the two ends of a given contig/scaffold (termed as A) share their sequence with a length of maxK (metaSPAdes or MEGAHIT) or maxK-1 (IDBA_UD), i.e., $A_L = A_R$.

(2) Neither end (with length of maxK(-1)) of a given contig/scaffold (termed as B) shared the sequence with the end sequence of any contigs/scaffolds from the whole metagenome but has a shorter end overlap length that is $\geq \min K$ but $< \max K(-1)$.

We consider that case (2) is also a situation where circularization is appropriate. In our simulations, we found if both ends of a given contig/scaffold do not share its sequence (maxK or maxK-1 in length) with any other end in the metagenome, it is appropriate to accept a shorter end overlap to indicate a circular genome. We found that it is important to require an end overlap of at least the minK.

We have modified this part of the manuscript accordingly..

- In Figure 2, can the authors change the colors that were used for “extended” partial and “extended_failed”? I’m colorblind and I couldn’t distinguish those. A safe bet is to use the Okabe-Ito color scheme. In panel F, I can’t see the difference between “consistency” and “divergency”.

We modified the colors and anticipate that they are distinguishable now.

- “Of the 7,334 phage species genomes identified, 167 were classified as huge phages (Fig. 3f)”. Is the correct figure being referenced here? This panel does not show huge phages in any way.

We apologize for this error. The figure reference has been corrected in the revised manuscript (Fig. 5d).

- In Figure 7, non-phage sequences are referred to as “Bacteria”. However, no taxonomic assignment was performed and some genes could be encoded by other taxa (e.g. Archaea). It would be more precise to refer to those genes as “non-phage” or something similar.

We agree, and now use “non-phage” instead of “Bacteria” in the figure as suggested, and also in the Results and Methods sections.

- Could the authors include a benchmark of speed and memory usage to give readers an expectation of how fast it is?

Done as suggested. Please see the figure below for more information on the COBRA analyses of the ocean virome sample of 250 m using contigs from IDBA_UD, metaSPAdes, and MEGAHIT. We added one Supplementary Table to include the time (speed) and other key information for some COBRA analyses we performed in this manuscript. During the revised period we modified the script by changing some data structures. As a result, the time and memory needed for the same dataset have dropped considerably (see Supplementary Table 9).

Minor comments:

- It is important to acknowledge the existence of other tools that aim to improve the circularization of metagenomes (e.g. <https://github.com/lmlui/Jorg>).

Done as suggested.

- "(1) the length of the overlap is $\max K$ (for metaSPAdes and MEGAHIT) or $\max K - 1$ (for IDBA_UD), or (2) the length of the overlap is at least the minimum kmer length used in the assembly. Next, COBRA searches for potential joining paths for each end based on valid path pairs (either "one_path_end" and "two_paths_end")" Case (2) should include all instances of (1), since the minimum required DTR length is lower. Why are they listed separately?

Please see the response to the comment related to Figure 1 above.

- "Although this is equivalent to the reverse path for the case 2 of 'one_path_end', it is considered separately because the end under consideration for extension depends on the direction that COBRA is progressing through the contig set". Does this mean that you might get different results depending on the order the contigs are processed? For instance:

A_R → B_L
 A_R → C_L
 B_Rrc → A_Lrc
 C_Rrc → A_Lrc

Assuming all joins are valid (two_paths_end), if you start processing from A, you could get (A+B, C). If you start processing from C, you'd get (C+A, B). If that's indeed the case, please clarify in the text.

Our descriptions may have misled the Reviewers. The original sentence means that in the two cases, the ends under consideration are different. Also, this part is about the definition of "one_path_end" and "two_paths_end", not about an extension.

In case 2 of "one_path_end" (see figure below), end A of contig 1 is the end under consideration, then it has only one end for potential join (i.e., end B from contig 2), thus end A is an "one_path_end".

However, for the case of “two_paths_end” (see figure below), it is end B of contig 2 that is under consideration, it has two potential joins (i.e., end A of contig 1, and end C of contig 3), thus end B is a “two_paths_end”.

Though both cases are from the same breakpoint in the *de novo* assembly, the ends under consideration are different, thus they were considered separately.

To avoid any potential misunderstanding, we modified this sentence, and hope this statement is clear now. Due to a related comment raised by **Reviewer #3**, the descriptions have been moved to the text legend of Supplementary Figure 6 in the Supplementary Information.

Regarding the example that the Reviewers provided, no matter whether COBRA starts with A or C, if the coverage and spanned paired reads conditions are met, C+A+B will be obtained as the final sequence.

- In Figure 1 the acronym “DTR” is used, however nowhere in the text it is mentioned that DTR means direct-terminal repeat, which could confuse readers that are not familiar with the term.

Thanks for pointing this out. We understand that the end overlap of contigs from the metagenomic assembly is not the real “DTR” (direct-terminal repeat) of the viral genomes. Given that “DTR” is not widely used in the manuscript, we modified Figure 1 to exclude it. The descriptions in the figure now read: ‘Neither end in “All path pairs”, and the contig has no end overlap \geq *mink*’, which is the definition of “orphan_end” contigs.

- COBRA is written as “CORBA” multiple times in the manuscript.

Corrected.

- “The short Illumina reads of the ocean virome 250M sample”, “250M” should be replaced by “250 m”, since the “m” stands for meters.

Modified as suggested.

- “blastn” and “BLASTn” are being used interchangeably.

“BLASTn” is now used throughout the manuscript.

- In Figure 2, the acronyms are confusing. For example, “C” means “COBRA” in panel D and “extended_circular” in panel E.

We modified the acronyms in panel E as follows, “extended_circular” = “e_c” and “extended_partial” = “e_p”.

- In Figure 2, I recommend replacing “AF similarity” with “ANI” to keep consistency with the rest of the manuscript.

Done as suggested.

- In Figure 4 panel E, the “References” text on top of the box “7,334 species genomes” make it seem that the 7,334 species are from the references, when they are actually from this study.

Thanks for the suggestion. We removed “References” and added “this study” in the figure.

- Which version of IMG/VR was used in the novelty analysis? The citation is for version 2, but it is not clear in the text if that was indeed the version the authors used.

It is IMG/VR v3. The correct citation (IMG/VR v3: an integrated ecological and evolutionary framework for interrogating genomes of uncultivated viruses) is now cited in the revised manuscript.

- Is there a reason for some genes being italicized (e.g. *cysC*) and some not (e.g. *whiB*, *pmoC*, *TerL*)? We

modified all the gene names to be italicized in the revised manuscript.

- The “gap.check.py” script is not publicly available. Make sure to provide it before publication. The script is now publicly available on the COBRA GitHub page.

Considering that the paper aims to describe a tool that is publicly available, I have conducted a test on COBRA and noted down my primary recommendations below. I firmly believe that COBRA holds the potential to become a widely-used tool. However, its current implementation is not friendly to a broad audience and could be easily improved to enhance adoption by the community. The authors should feel free to not implement those suggestions, as these won’t affect the overall quality of the manuscript.

We thank the Reviewers for testing COBRA. We respond to the specific suggestions and comments below.

- The current step-by-step assumes a lot from the user and doesn’t make several important details clear, for instance: (1) to compute the coverage, the BAM/SAM files need to be sorted beforehand (an information that is not mentioned in the guide), so it could be useful to have an example showing how to produce the BAM/SAM and sort it (using Bowtie2 and samtools, for example); (2) the guide says that the user should process the coverage file to produce a two-column file, but it doesn’t explain how this can be achieved.

(1) The step-by-step pipelines to generate the sorted BAM/SAM files, and (2) the script (coverage.transfer.py) used to convert the coverage file produced by “jgi_summarize_bam_contig_depths” into a two-column file, are now available at the GitHub page of COBRA.

- Right now, COBRA is not distributed in any package manager (Conda and/or PyPI), which is expected of bioinformatics tools nowadays. The process to install COBRA is very manual (install the correct Python requirements, external bioinformatics tools, clone the repository, etc.) and prevents it from being easily incorporated into pipelines.

COBRA has been distributed via PyPI and thus could be installed via `pip install`. This has been updated on the GitHub page.

- As a consequence of the lack of distribution through a package manager, COBRA is not in the user PATH and needs to be executed as a script ("`python cobra.py`" instead of just "`cobra`"), preventing it from being available as a central installation in servers.

COBRA has now been distributed via PyPI.

- The execution log is not shown on the screen. Instead, the script prints the name of some of the input contigs without any context.

(1) The execution log details are saved in the log file in the output folder.

(2) The original version of COBRA the Reviewer tested had a line to print the names of some specific contigs/scaffolds for testing. This now has been fixed in the updated version.

- Is there a reason that the query contigs need to be provided as a FASTA file if their sequences are already present in the whole assembly? It would be more user-friendly if the query could be provided as a .txt with a list of accessions. For big assemblies, storing a redundant FASTA file would be a waste of space.

We really appreciate this suggestion, the updated version of COBRA could take a fasta format file or a list of contigs/scaffolds names in a text file as input, this has been mentioned in the revised manuscript and also the GitHub page.

- "`jgi_summarize_bam_contig_depths`" could be replaced with `pyCoverM` or `CoverM`. These tools are much faster and eliminate the need for processing the coverage file to generate the required two-column format, as they can directly produce the desired format. By utilizing `pyCoverM` or `CoverM`

We thank the Reviewers for suggesting `pyCoverM` and `CoverM`. We now have a more detailed procedure to guide users on how to obtain the mapping file all the way to the coverage file. We kept the way of using "`jgi_summarize_bam_contig_depths`" to get the coverage file along with the script needed to generate a two-column file, we also introduced `pyCoverM` and `CoverM` as alternative ways.

- Would it be possible to make the tool assembler-agnostic, as long as "`mink`" and "`maxk`" are provided? If the user uses an assembler other than MEGAHIT, IDBA_UD, or metaSPAdes, it is not immediately clear that they can use COBRA as long as they input the correct k-mer lengths and "`lie`" to the tool about the assembler they used.

Unfortunately, it is impossible to do so for now due to two primary reasons, (1) COBRA has only been tested for contigs/scaffolds assembled by MEGAHIT, IDBA_UD, and metaSPAdes, and (2) the expected overlap length is $\text{maxK}-1$ for contigs/scaffolds from IDBA_UD but exactly maxK for those from MEGAHIT and metaSPAdes.

To avoid any confusion, we mention on the GitHub page that currently COBRA should only be used for assembled contigs/scaffolds from these three assemblers. However, we tested that if the `maxK` provided is not the one used in the assembly, very very few contigs/scaffolds will have shared ends with the expected length.

Reference cited in the responses

Brown, Christopher T., Laura A. Hug, Brian C. Thomas, Itai Sharon, Cindy J. Castelle, Andrea Singh, Michael J. Wilkins, Kelly C. Wrighton, Kenneth H. Williams, and Jillian F. Banfield. "Unusual biology across a group comprising more than 15% of domain Bacteria." *Nature* 523, no. 7559 (2015): 208-211.

Chen, Lin-Xing, Karthik Anantharaman, Alon Shaiber, A. Murat Eren, and Jillian F. Banfield. "Accurate and complete genomes from metagenomes." *Genome research* 30, no. 3 (2020): 315-333.

Jiang, Jing-Zhe, Wen-Guang Yuan, Jiayu Shang, Ying-Hui Shi, Li-Ling Yang, Min Liu, Peng Zhu, Tao Jin, Yanni Sun, and Li-Hong Yuan. "Virus classification for viral genomic fragments using PhaGCN2." *Briefings in Bioinformatics* 24, no. 1 (2023): bbac505.

Pons, Joan Carles, David Paez-Espino, Gabriel Riera, Natalia Ivanova, Nikos C. Kyrpides, and Mercè Llabrés. "VPF-Class: taxonomic assignment and host prediction of uncultivated viruses based on viral protein families." *Bioinformatics* 37, no. 13 (2021): 1805-1813.

Roux, Simon, David Páez-Espino, I-Min A. Chen, Krishna Palaniappan, Anna Ratner, Ken Chu, T. B. K. Reddy et al. "IMG/VR v3: an integrated ecological and evolutionary framework for interrogating genomes of uncultivated viruses." *Nucleic acids research* 49, no. D1 (2021): D764-D775.

Camargo, Antonio Pedro, Simon Roux, Frederik Schulz, Michal Babinski, Yan Xu, Bin Hu, Patrick SG Chain, Stephen Nayfach, and Nikos C. Kyrpides. "You can move, but you can't hide: identification of mobile genetic elements with geNomad." *bioRxiv* (2023): 2023-03.

Reviewer #3 (Remarks to the Author):

Key results

Your overview of the key messages of the study, in your own words, highlighting what you find significant or notable. Usually, this can be summarized in a short paragraph.

The manuscript by Chen and Banfield describes COBRA, an informatics tool that extends contigs from metagenomic assemblies by leveraging contig overlap and sequencing coverage to resolve multiple suggested contig joins at de Bruijn graph-based assembler breakpoints. Contig end pairs with unique significant overlap and similar coverage are joined. Extending contigs is a beneficial step in a metagenomics pipeline because it enables identification of longer, more complete sequences, which in turn improves all downstream analyses such as comparative genomics and population genetics.

COBRA was optimized against simulated data (that included within genome repeats, regions shared across genomes, and within population sequence variation) and then benchmarked using an ocean virome dataset that had both short and long (nanopore) sequencing data available against modern assemblers and binning tools. Against the assemblers, COBRA was able to extend ~50% of the contigs and increased the average contig length from 20 to 32.5 kbp and increased the number of complete and high-quality viral genomes from 46 to 241 from the short-read data alone (the polished short- plus long-read datasets achieved 1,864 complete viral genomes). Against the binners, COBRA was much more accurate in combining sequences from the same viral genome, recovering 1.7 to 5.8 times more accurate viral bins, and each bin was 3-6 kbp longer using COBRA than other bins. COBRA also had extremely low bin contamination, with only 1 out of 400 of the bins being contaminated versus much worse numbers (111-261 contaminated bins) across the binning algorithms.

Once optimized and benchmarked, COBRA was applied to 122K (>10kb) contigs from 231 published freshwater metagenomic datasets and filtered the results for CheckV-complete genomes, which resulted in 12,118 high-quality or complete viral genomes. These data were mined for some high-level inferences including expanding genomic representation for huge phages, assessing marker genes for transcripts, and uncovering Actinobacteria phages and phage-encoded AMGs of interest. There are tens of papers that could be written about this treasure trove of new sequences, and it will be of great value for the freshwater microbial and virus ecology community to be able to mine such data with domain specific knowledge and starting from such high-quality genomes.

We thank the Reviewers for their very careful reading of our manuscript and the overall positive evaluation of our work.

Validity

Your evaluation of the validity and robustness of the data interpretation and conclusions. If you feel there are flaws that prohibit the manuscript's publication, please describe them in detail.

Overall, this study clearly demonstrates COBRA's ability to extend contigs and produce more complete viral genomes from virus-enriched (virome) datasets and whole metagenomes. Comparing COBRA constructed complete viral genomes to "polished" virus genomes (short- plus long-read hybrid assembled virus genomes) made it easy to see COBRA's accuracy. It was impressive to see how well it outperformed standard assembly-only and viral and non-viral binning tools.

However, several suggestions are as follows:

1) What is the run-time and computational power needed for COBRA on datasets of different sizes?

The run-time that COBRA needs is highly dependent on (1) the total number of contigs/scaffolds, (2) the number of queries, and (3) the size of the mapping file. We added a Supplementary Table (Supplementary Table 9) in the revised manuscript to show all this information (including running time and memory) of the datasets we tested in this study.

2) Is COBRA limited to reconstructing double-stranded DNA viruses, or can it be used for RNA viruses? ssDNA viruses?

We anticipate that COBRA could be applied to both RNA and ssDNA viruses if the samples were sequenced with paired-end short reads and assembled with one of the three assemblers that we have tested.

3) How does COBRA compare against contig extension papers already out, such as ContigExtender (<https://bmcbioinformatics.biomedcentral.com/articles/10.1186/s12859-021-04038-2>)? Are there others?

As suggested, we compared the efficiency and accuracy of ContigExtender using the same data that we used to compare COBRA and binning tools. Due to the relatively slow processing rate of ContigExtender, we were able to include the results of only 29 contigs in the revised manuscript. We included these results in Supplementary Table 4 and the corresponding descriptions in the revised manuscript.

4) Can the authors make clear, what are the current technical limitations of COBRA? How can contig extension be further improved?

We added one paragraph in the Discussion section as suggested.

Significance

Your view on the potential significance of the conclusions for the field and related fields. If you think that other findings in the published literature compromise the manuscript's significance, please provide relevant references.

COBRA appears extremely useful for recovering more high-quality and complete viral genomes from short-read datasets, and the tool focuses in an area that is under-developed in viromics. However, the authors largely ignore the virus ecogenomics field. Though virus contig extension is a large hole, and one COBRA seems to handle well, there is a lot of the remaining pipeline that is years in the making and solid. Consider an early few sentences or paragraph that speaks to the advances the field has made in areas such as virus identification, taxonomic classification, automating AMG identification, estimating completeness, etc. These each represent large efforts in the field, but all will be made better by the larger genomic context that COBRA provides at the front-end of these advances.

We agree with the Reviewers that all the mentioned advances in the field are very important. We added the information to the end of the first paragraph of the Introduction section.

As well, COBRA's comparison against assembly-only and binning tools is useful, but it is hard to evaluate its significance since the tool is not compared against existing contig extension methods (e.g. one explicitly for viral metagenomes, published in 2021 – ContigExtender <https://bmcbioinformatics.biomedcentral.com/articles/10.1186/s12859-021-04038-2>). The approach is different, so it is not clear how these two tools compare.

We added a separate section in the Results section to compare the performance of COBRA and ContigExtender (“Performance comparison of COBRA against ContigExtender”).

Data and methodology

Your assessment of the validity of the approach, the quality of the data, and the quality of presentation. We ask reviewers to assess all data, including those provided as supplementary information. If any aspect of the data is outside the scope of your expertise, please note this in your report or in the comments to the editor. We may, on a case-by-case basis, ask reviewers to check code provided by the authors (see this Nature editorial for more information).

Reviewers have the right to view the data and code that underlie the work if it would help in the evaluation, even if these have not been provided with the submission (see this Nature editorial). If essential data are not available, please contact the editor to obtain them before submitting the report. The approach is valid: the accuracy of the contigs was acquired by comparing the COBRA-extended short-read contigs to complete genomes assembled from the same dataset. These genomes were taken from a set of published, manually curated, complete freshwater genomes. The methodology is very well detailed in the main text and supplemental making it very clear how COBRA works and how the study was carried out. Multiple metrics are reported such as the N50, longest contig recovered, average contig length, and total number of contigs, and reported for before and after contigs are passed through COBRA to easily demonstrate the improvement. Statistical significances are shown. Multiple widely-used assemblers were also used and compared showing that COBRA's behavior is comparable across assembler programs.

Supplementary tables provide highly detailed metadata on COBRA and assembler performance, as well as all published data used the main and supplementary figures. Supplementary Table 3 and 4 for example include all features (columns) necessary to reproduce the freshwater genome findings from this study. All freshwater genomes are available via figshare at a link provided in the Data Availability Section, and all other data used is cited with the study it came from.

The data tables are all easily human readable, and the ones used for computing the figures all appear to be easily parsed by computers.

We thank the Reviewers for the positive comments on the “data and methodology” of our manuscript.

Analytical approach

Your assessment of the strength of the analytical approach, including the validity and comprehensiveness of any statistical tests. If any aspect of the analytical approach is outside the scope of your expertise, please note this in your report or in the comments to the editor.

Analytical approach of the benchmarking and validation is very solid. They show a lot of data proving COBRA creates longer contigs, and is highly accurate compared to other viral binning tools with statistical significance tests. They compare COBRA using multiple widely used assemblers as well showing it can be used with different assemblers. Data presentation makes it easy to see how the data quality improves with COBRA.

We thank the Reviewers for the positive comments on the “Analytical approach” of our manuscript.

Suggested improvements

Your suggestions for additional experiments or data that could help strengthen the work and make it suitable for publication in the journal. Suggestions should be limited to the present scope of the manuscript; that is, they should only include what can be reasonably addressed in a revision and exclude what would significantly change the scope of the work. The editor will assess all the suggestions received and provide additional guidance to the authors.

The biggest needs are:

1) Compare COBRA vs ContigExtender

a. ContigExtender addresses chimeric sequences, but there is no addressal of chimeric sequences from assembly in COBRA as far as I can tell. I guess the validation strategy accounts for false positive extensions, but explicit language in the discussion addressing the chimera issue and how COBRA could extend chimeras, creating longer, but still chimeric sequences (and therefore incorrect) would be good to do. COBRA may not cause errors, but it can't fix existing errors from the assembly process.

The Reviewers are correct that COBRA can't fix existing errors, like chimeras, from the assembly process. We concur that chimeras are a significant problem that will contribute to overall incorrect sequences. We now note this in the Discussion section of the revised manuscript.

2) Add in compute times / resources required for COBRA

We listed the compute times in **Supplementary Table 9**. We acknowledge that it is very difficult to predict the compute times from sample to sample, as the number of query contigs/scaffolds and the complexity of the corresponding microbiome are different.

3) In the discussion section, address any limitations of COBRA and where contig extension could be improved, or how it could be integrated into existing assembly algorithms. Only 50% of the contigs from the validation could be extended—I'd be curious to know if there were any patterns to why the other 50% could not be extended, especially since these were pulled directly from the complete viral genomes and have a correct pairing.

To be exact, for the queries from the three assemblies, 44-57% were assigned as “extended_failed” and “orphan_end”.

As for why “only 50% of the contigs from the validation could be extended”, we believe the main reason is the long-read sequencing captured the information in the regions that the short-read sequencing missed. This could be partly documented by what we showed in **Figure 2a**, i.e., the sequencing coverage of the contigs is the significant factor hindering their extension by COBRA. It is understandable that the assemblers could not extend a contig longer if there are no more reads available.

4) How low coverage can COBRA operate at? What are the authors guidelines for what coverage should be trusted vs not to make contig extensions?

COBRA could operate at coverage of any level, but may not be able to extend contigs with low coverage. As observed on the ocean virome dataset, once extended, extensions of contigs with coverage < 10X are still reliable (see Figure 2 for details).

5) Bring Supp. Fig. 6 into the main manuscript and summarize the text describing it.

We thank the Reviewers for this suggestion. However, given that we already had many figures in the main text, and that we agree with the Reviewers that this figure included too many details (see our response to another related comment below), we kept **Supplementary Figure 6** in the Supplementary Information. We also moved some of the detailed descriptions of how COBRA works from the Results section to the legend of **Supplementary Figure 6**, to make it more understandable.

6) Could the authors have a paragraph that pragmatically walks the reader through the relative efforts / costs of the hybrid method of Beaulaurier et al 2020 vs COBRA+short-read methods? From one perspective, COBRA got 241 complete genomes only as compared to 1864 for the hybrid method, but at what cost?

The Beaulaurier et al. 2020 study used only long reads and they obtained 1,205 “complete” genomes from the 250 m sample that we used in our benchmarking, and 1,864 genomes from all three samples they analyzed. In our experience, it is essential to use accurate short (Illumina) reads to fix the inevitable errors (e.g., single bp indels) in long read assemblies to achieve true completion. Given this, the cost increase for a hybrid assembly is substantial. We also point out the value of COBRA for improvement of analyses that utilize the massive number of existing short read only datasets and note that not all samples can be sequenced using long read methods due to DNA requirements. We added one paragraph to briefly address this.

Clarity and context

Your view on the clarity and accessibility of the text, and whether the results have been provided with sufficient context and consideration of previous work. Note that we are not asking for you to comment on language issues such as spelling or grammatical mistakes.

The paper is well written. Beyond the suggestions above, I would only suggest:

1) There is perhaps too much detail on how the method works in the results section. It could be summarized and put in context with supplementary figure 6 if it's brought into the main figure. Right now it's just too much text and hard to follow each case.

We thank the Reviewers for this suggestion, and we agree that we have stated how COBRA works in detail in the Results section, including all the four primary steps in COBRA analyses, and only steps 1 and 2 are related to the original **Supplementary Figure 6** in the original manuscript. To avoid too many details in the Results section of the main text, we kept **Supplementary Figure 6** in the **Supplementary Information** with some modifications to make it clearer, and some details were moved to its text legend. We hope that the Reviewers will be satisfied with our revisions here.

2) Consider moving the "Application of COBRA on whole metagenomes and non-viral microbial genomes" to before the huge phage section. This section is more similar to the viromes validation section and may flow better.

The section "Application of COBRA on whole metagenomes and non-viral microbial genomes" is moved to before "Application of COBRA to freshwater metagenomes to recover high quality viral genomes". With the Comments (5 and 6) from **Reviewer #1**, we also added a benchmarking of whole metagenomes (with complete bacterial genomes) in the section.

References

Your view on whether the manuscript references previous literature appropriately.

As described above, context of an early few sentences or paragraph that speaks to the advances the viromics field has already made in virus identification, taxonomic classification, automating AMG identification, estimating completeness, etc. These each represent large efforts in the field, but all will be made better by the larger genomic context that COBRA provides. As well, the comparison against the 2021 publication of ContigExtender.

We described the advances in the viromes field at the end of the first paragraph in the Introduction section of our revised manuscript.

The comparison of COBRA against ContigExtender was performed, and one section titled "Performance comparison of COBRA against ContigExtender", was added in the Results section of the revised manuscript.

Your expertise

Please indicate any particular part of the manuscript, data or analyses that you feel is outside the scope of your expertise, or that you were unable to assess fully.

I was able to review the ms fully.

We thank the Reviewers again for their comments on our manuscript.

Reference cited in the responses

Beaulaurier, John, Elaine Luo, John M. Eppley, Paul Den Uyl, Xiaoguang Dai, Andrew Burger, Daniel J. Turner et al. "Assembly-free single-molecule sequencing recovers complete virus genomes from natural microbial communities." *Genome Research* 30, no. 3 (2020): 437-446.

Decision Letter, first revision:

Message: 8th November 2023

Dear Jill,

Thank you for your patience while your manuscript "COBRA improves the quality of viral genomes assembled from metagenomes" was under peer-review at Nature Microbiology. It has now been seen by 3 referees, whose expertise and comments you will find at the of this email. You will see from their comments below that while they find your work of interest, some important points are raised. We are very interested in the possibility of publishing your study in Nature Microbiology, but would like to consider your response to these concerns in the form of a revised manuscript before we make a final decision on publication.

In particular, you will see that the referees ask to better discuss the limitations (only 5000 queries - why is this?), they have some technical suggestions (referee #1), and they ask to tone down broader conclusions e.g. application to all metagenomes, and to improve the presentation of Fig 5E. Please note that editorially, we will not require you to benchmark your method to additional tools at this point, but that we would ask you to instead better introduce and discuss the different approaches and then tone down conclusions around more general application of the tool. The rest of the referees' reports are clear and the remaining issues should be straightforward to address.

If you have not done so already please begin to revise your manuscript so that it conforms to our Article format instructions at <http://www.nature.com/nmicrobiol/info/final-submission/>

The usual length limit for a Nature Microbiology Article is six display items (figures or tables) and 3,000 words. We have some flexibility, and can allow a revised manuscript at 3,500 words, but please consider this a firm upper limit. There is a trade-off of ~250 words per display item, so if you need more space, you could move a Figure or Table to Supplementary Information.

Some reduction could be achieved by focusing any introductory material and moving it to the start of your opening 'bold' paragraph, whose function is to outline the background to your work, describe in a sentence your new observations, and explain your main conclusions. The discussion should also be limited. Methods should be described in a separate section following the discussion, we do not place a word limit on Methods.

Nature Microbiology titles should give a sense of the main new findings of a manuscript, and should not contain punctuation. Please keep in mind that we strongly discourage active verbs in titles, and that they should ideally fit within 90 characters each (including spaces).

Please include a data availability statement as a separate section after Methods but before references, under the heading "Data Availability". This section should inform readers about the availability of the data used to support the conclusions of your study. This information

includes accession codes to public repositories (data banks for protein, DNA or RNA sequences, microarray, proteomics data etc...), references to source data published alongside the paper, unique identifiers such as URLs to data repository entries, or data set DOIs, and any other statement about data availability. At a minimum, you should include the following statement: "The data that support the findings of this study are available from the corresponding author upon request", mentioning any restrictions on availability. If DOIs are provided, we also strongly encourage including these in the Reference list (authors, title, publisher (repository name), identifier, year). For more guidance on how to write this section please see:

<http://www.nature.com/authors/policies/data/data-availability-statements-data-citations.pdf>

To improve the accessibility of your paper to readers from other research areas, please pay particular attention to the wording of the paper's opening bold paragraph, which serves both as an introduction and as a brief, non-technical summary in about 150 words. The opening paragraph should not contain references. Because scientists from other sub-disciplines will be interested in your results and their implications, it is important to explain essential but specialised terms concisely. We suggest you show your summary paragraph to colleagues in other fields to uncover any problematic concepts.

If your paper is accepted for publication, we will edit your display items electronically so they conform to our house style and will reproduce clearly in print. If necessary, we will re-size figures to fit single or double column width. If your figures contain several parts, the parts should form a neat rectangle when assembled. Choosing the right electronic format at this stage will speed up the processing of your paper and give the best possible results in print. We would like the figures to be supplied as vector files - EPS, PDF, AI or postscript (PS) file formats (not raster or bitmap files), preferably generated with vector-graphics software (Adobe Illustrator for example). Please try to ensure that all figures are non-flattened and fully editable. All images should be at least 300 dpi resolution (when figures are scaled to approximately the size that they are to be printed at) and in RGB colour format. Please do not submit Jpeg or flattened TIFF files. Please see also 'Guidelines for Electronic Submission of Figures' at the end of this letter for further detail.

Figure legends must provide a brief description of the figure and the symbols used, within 350 words, including definitions of any error bars employed in the figures.

Please include a statement before the acknowledgements naming the author to whom correspondence and requests for materials should be addressed.

Finally, we require authors to include a statement of their individual contributions to the paper -- such as experimental work, project planning, data analysis, etc. -- immediately after the acknowledgements. The statement should be short, and refer to authors by their initials. For details please see the Authorship section of our joint Editorial policies at http://www.nature.com/authors/editorial_policies/authorship.html

- * include a point-by-point response to any editorial suggestions and to our referees.
- Please include your response to the editorial suggestions in your cover letter, and please

upload your response to the referees as a separate document.

* ensure it complies with our format requirements for Letters as set out in our guide to authors at www.nature.com/nmicrobiol/info/gta/

* state in a cover note the length of the text, methods and legends; the number of references; number and estimated final size of figures and tables

* resubmit electronically if possible using the link below to access your home page:

*This url links to your confidential homepage and associated information about manuscripts you may have submitted or be reviewing for us. If you wish to forward this e-mail to co-authors, please delete this link to your homepage first.

Please ensure that all correspondence is marked with your Nature Microbiology reference number in the subject line.

Nature Microbiology is committed to improving transparency in authorship. As part of our efforts in this direction, we are now requesting that all authors identified as 'corresponding author' on published papers create and link their Open Researcher and Contributor Identifier (ORCID) with their account on the Manuscript Tracking System (MTS), prior to acceptance. This applies to primary research papers only. ORCID helps the scientific community achieve unambiguous attribution of all scholarly contributions. You can create and link your ORCID from the home page of the MTS by clicking on 'Modify my Springer Nature account'. For more information please visit www.springernature.com/orcid.

We hope to receive your revised paper within three weeks. If you cannot send it within this time, please let us know.

Yours sincerely,

Reviewer Expertise:

Referee #1: Computational biology, bioinformatics, viral ecology

Referee #2: Viruses, bioinformatics

Referee #3: Metagenomics, viral ecology

Reviewers Comments:

Reviewer #1 (Remarks to the Author):

I have now had the opportunity to revisit the manuscript by Chen and Banfield. Overall, I really commend the authors for the changes made and for taking the time to answer my queries, patiently and thoroughly. The manuscript is much improved. I only have a few outstanding comments and suggestions.

1. I suggest the authors to include a conda install – this will make their own lives much easier down the road as installation issues can often mount without conda.

2. I am still not convinced that the benchmarking conducted to broadly use this approach for all metagenomic data is sufficient. I understand that the authors mention using COBRA before binning, however I still remain cautious about this for two reasons:

o First – the authors provide users the opportunity/suggest that some scaffolds or a part of the dataset may be used for extension in lieu of all scaffolds or the entire metagenome. I still fear that this may be used inappropriately, more so as COBRA is limited to 5000 queries. Time is mentioned as a possible issue, but time should be a barrier to accuracy. I think this may be addressed in the form of appropriate language/recommendations to

users which I think is too loose right now.

o Second – The human dataset used for benchmarking is a simple dataset overall. While the benchmarking done for viruses is now very robust, I do not think this is the case of metagenomes. I still wonder how COBRA will fare if a more complex dataset such as soil is used by a user. Will the low false positivity rate remain in case of the high strain variation observed in some natural environments? As an example, all manuscripts describing virus identification software typically examine multiple environments to evaluate performance and biases which can be environment specific.

Reviewer #2 (Remarks to the Author):

In this revision, the authors have effectively addressed significant concerns, resulting in an overall improvement in the manuscript's quality. I understand that the authors are preparing another manuscript describing complete plasmids assembled with COBRA. Simply providing the number of plasmids found in the groundwater metagenomes should suffice to showcase COBRA's ability to identify different types of circular MGEs. I deeply appreciate the authors' efforts to enhance the tool's usability and documentation and I believe COBRA has the potential to become an important software in the field. This study is thorough and of high quality, however, there remain certain aspects in the manuscript

Major comments:

- I agree with Reviewer #1 that it is essential to investigate why assemblers do not perform the joins that COBRA does and that assembly graphs should be explored for that. This would significantly improve the manuscript because it is built on the idea that assemblers often miss valid joins, but the study currently lacks an analysis of assembly graphs, which are the basis for assemblers' decisions. Such an analysis could provide insights into the general usefulness of tools like COBRA in improving assemblies. SPAdes outputs assembly graphs by default and the output of MEGAHIT can be processed to generate graphs using the instructions described here:
<https://github.com/voutcn/megahit/wiki/Visualizing-MEGAHIT's-contig-graph>

- I appreciate the inclusion of a section in which the authors compare COBRA to ContigExtender, as well as the citation of other tools that enhance assembly contiguity in the introduction. However, I believe there is room for improvement in the way other tools are introduced, as it is essential to acknowledge the differing approaches employed by COBRA and these other tools. Furthermore, I recommend that the authors consider incorporating additional tools into the benchmark, particularly those that leverage assembly graphs (e.g., phables) or reassembly (e.g., Jorg). This would offer an opportunity for the authors to emphasize that COBRA can be more generally applied, as it does not require assembly graphs or resource-intensive reassemblies.

- I thank the authors for addressing my concerns regarding the novelty analysis. However, I still believe the way the results are presented can be misleading. As I understand it, the authors (1) conducted a prefiltering step to select a subset of sequences which are similar to their identified viruses (with a minimum alignment length of 10 kb and a minimum ANI of 90%); (2) performed clustering using their genomes and the sequences selected in the prefiltering. While this prefiltering was necessary to avoid clustering unrelated sequences, Figure 5's current presentation may lead readers to misunderstand the dataset sizes. Additionally, this panel includes information that isn't directly relevant to the study (for instance, there's no reason to present the overlap between the external datasets since only a subset of their sequences are used for this comparison). I suggest simplifying the presentation of results by showing only the number of genomes that are exclusive to this study (novel species) and those shared with other datasets (known species). This would avoid misrepresentation of the external datasets.

Minor comments:

- How does COBRA deal with ends that contain ambiguous bases (i.e. Ns)?

- Did you evaluate if low-complexity sequences are the cause of problematic/contaminated joins?

- In Figure 8 the gene names are not italicized.
- Phage taxa in Figure 6, Figure 8, and elsewhere should be italicized.
- checkV → CheckV
- “The divergent performance of COBRA and the binning tools was also supported by the metrics”: I think this sentence is a bit confusing, the reader can’t tell what these metrics are measuring without opening the table.

I have additional suggestions concerning the software itself. These suggestions, as previously mentioned, will not impact my review of the manuscript and should be regarded as recommendations aimed at improving the installation and usability of COBRA.

- In my tests, I noticed that COBRA exhibited sluggish performance when handling large assemblies. Upon inspecting the code, I identified multiple steps that could be easily parallelized. Implementing these modifications would result in a substantial enhancement of COBRA’s processing speed, enabling it to efficiently handle datasets, including large ones. These optimizations require minor adjustments, such as storing data in memory rather than processing it sequentially as files are read.

- Having the software available for installation via Mamba/Conda would be greatly appreciated by the community, since these package managers are pretty much standard in bioinformatics nowadays. Since COBRA is available in PyPI already, it should be straightforward to write a recipe for Bioconda.

Reviewer #3 (Remarks to the Author):

The authors have done a great job of revising the manuscript to assuage my concerns and criticisms. IT is exciting that further benchmarking showed this tool to be so helpful.

One surprise was that the tool can only do 5,000 queries in several hours, the limitation being parallel processing. Is there something about COBRA that prevents parallel processing? It would make mining the vast short-read metagenomic space much easier and contribute to its broader use if it could at least implement parallel processing as a way to grow the number of queries that can be handled.

Author Rebuttal, first revision:

Response to referees’ comments

Below are our point-by-point responses to the Reviewers’ comments, with the comments in black, and our response in blue. The modifications are highlighted in the main text of the revised manuscript.

Reviewer #1 (Remarks to the Author):

I have now had the opportunity to revisit the manuscript by Chen and Banfield. Overall, I really commend the authors for the changes made and for taking the time to answer my queries, patiently and thoroughly. The manuscript is much improved. I only have a few outstanding comments and suggestions.

We thank the reviewer for all their helpful comments.

1. I suggest the authors to include a conda install – this will make their own lives much easier down the road as installation issues can often mount without conda.

We thank the reviewer for the suggestion. We have been working on it and will update the GitHub page once it is approved by the Bioconda team, which may take several days to several weeks.

2. I am still not convinced that the benchmarking conducted to broadly use this approach for all metagenomic data is sufficient. I understand that the authors mention using COBRA before binning, however I still remain cautious about this for two reasons:

o First – the authors provide users the opportunity/suggest that some scaffolds or a part of the dataset may be used for extension in lieu of all scaffolds or the entire metagenome. I still fear that this may be used inappropriately, more so as COBRA is limited to 5000 queries. Time is mentioned as a possible issue, but time should be a barrier to accuracy. I think this may be addressed in the form of appropriate language/recommendations to users which I think is too loose right now.

It is a misunderstanding that COBRA is limited to 5,000 queries. Actually, this is just an example number, to give the readers and users a quick evaluation of how long they may need to finish the analyses, and this is not a speed-limiting factor due to the improvement of the codes. However, if the community is complicated (indicated by large size of the whole assembled contigs/scaffolds), COBRA may need more time to process it, as it has to carefully evaluate all the potential joins.

We have modified the sentences and noted the users that COBRA may be a little slow if the community is complicated.

o Second – The human dataset used for benchmarking is a simple dataset overall. While the benchmarking done for viruses is now very robust, I do not think this is the case of metagenomes. I still wonder how COBRA will fare if a more complex dataset such as soil is used by a user. Will the low false positivity rate remain in case of the high strain variation observed in some natural environments? As an example, all manuscripts describing virus identification software typically examine multiple environments to evaluate performance and biases which can be environment specific.

We understand the reviewers' concerns about the performance of COBRA on more complex datasets such as soil. Given that, (1) we already showed the robustness and accuracy of COBRA in extending the soil viral sequences, (2) the lack of more complicated datasets (for example, soil) to perform whole metagenome benchmark, and (3) the suggestions from the editor, we thus do not perform additional benchmark on whole metagenomes. We now have essentially eliminated all reference to application to whole metagenomes from our revised manuscript.

Reviewer #2 (Remarks to the Author):

In this revision, the authors have effectively addressed significant concerns, resulting in an overall improvement in the manuscript's quality. I understand that the authors are preparing another manuscript describing complete plasmids assembled with COBRA. Simply providing the number of plasmids found in the groundwater metagenomes should suffice to showcase COBRA's ability to identify different types of circular MGEs. I deeply appreciate the authors' efforts to enhance the tool's usability and documentation and I believe COBRA has the potential to become an important software in the field. This study is thorough and of high quality, however, there remain certain aspects in the manuscript.

We thank the reviewer for their positive evaluation of our manuscript, and their understanding that we only showed the number of plasmid genomes that COBRA obtained without additional details in this manuscript.

Major comments:

- I agree with Reviewer #1 that it is essential to investigate why assemblers do not perform the joins that COBRA does and that assembly graphs should be explored for that. This would significantly improve the manuscript because it is built on the idea that assemblers often miss valid joins, but the study currently lacks an analysis of assembly graphs, which are the basis for assemblers' decisions. Such an analysis could provide insights into the general usefulness of tools like COBRA in improving assemblies.

SPAdes outputs assembly graphs by default and the output of MEGAHIT can be processed to generate graphs using the instructions described here:

<https://github.com/voutcn/megahit/wiki/Visualizing-MEGAHIT's-contig-graph>

We understand that the assembly graphs could provide some additional information that may help, but again we want to emphasize that COBRA was developed on a method that we have already benchmarked and documented by manual curation, using assembly graphs will foundationally change the general idea of COBRA.

- I appreciate the inclusion of a section in which the authors compare COBRA to ContigExtender, as well as the citation of other tools that enhance assembly contiguity in the introduction. However, I believe there is room for improvement in the way other tools are introduced, as it is essential to acknowledge the differing approaches employed by COBRA and these other tools. Furthermore, I recommend that the authors consider incorporating additional tools into the benchmark, particularly those that leverage assembly graphs (e.g., phables) or reassembly (e.g., Jorg). This would offer an opportunity for the authors to emphasize that COBRA can be more generally applied, as it does not require assembly graphs or resource-intensive reassemblies.

We thank the reviewers for the suggestions of introducing other tools like phables and Jorg, etc., we include them in the introduction section but do not benchmark them (as noted by the editor). And, the reviewer is correct that one advantage of COBRA is that it does not require assembly graphs (for example, phables) or resource-intensive reassemblies (for example, Jorg). We added this to the Discussion section accordingly.

- I thank the authors for addressing my concerns regarding the novelty analysis. However, I still believe the way the results are presented can be misleading. As I understand it, the authors (1) conducted a prefiltering step to select a subset of sequences which are similar to their identified viruses (with a minimum alignment length of 10 kb and a minimum ANI of 90%); (2) performed clustering using their genomes and the sequences selected in the prefiltering. While this prefiltering was necessary to avoid clustering unrelated sequences, Figure 5's current presentation may lead readers to misunderstand the

dataset sizes. Additionally, this panel includes information that isn't directly relevant to the study (for instance, there's no reason to present the overlap between the external datasets since only a subset of their sequences are used for this comparison). I suggest simplifying the presentation of results by showing only the number of genomes that are exclusive to this study (novel species) and those shared with other datasets (known species). This would avoid misrepresentation of the external datasets.

We thank the reviewers for the careful reading and the helpful suggestions, and we have modified the figure as suggested.

Minor comments:

- How does COBRA deal with ends that contain ambiguous bases (i.e. Ns)?

The contigs/scaffolds with ambiguous bases at their ends will not be used by COBRA, as no shared ends could be found.

- Did you evaluate if low-complexity sequences are the cause of problematic/contaminated joins?

We evaluated the 13 problematic joins and the 1 contaminated join from COBRA on the ocean virome 250 m sample.

For problematic joins, it is because the contigs from two related original genomes have similar sequencing coverage, COBRA joined them together based on the sequencing coverage ratio and spanned reads conditions.

For the only contaminated join from COBRA, it is because the two original genomes shared several small but identical regions and the assembly was broken, also the two genomes had similar sequencing coverage, then COBRA joined 4 contigs from one genome and 5 contigs from the other genome together.

The inspection showed that these joins are not random joins, but still obey the mechanisms of COBRA.

- In Figure 8 the gene names are not italicized.

Modified as suggested.

- Phage taxa in Figure 6, Figure 8, and elsewhere should be italicized.

Modified as suggested.

- checkV → CheckV

Done as suggested.

- "The divergent performance of COBRA and the binning tools was also supported by the metrics": I think this sentence is a bit confusing, the reader can't tell what these metrics are measuring without opening the table.

We modified the sentence to make it more clear by adding numbers of the metrics, which is now read "Compared against the binning tools, COBRA had the highest values of the metrics including precision (0.98 vs 0.06-0.47), recall (0.83 vs 0.61-0.71), F1-score (0.90 vs 0.11-0.56), specificity (0.98 vs 0.07-0.61) and accuracy (0.91 vs 0.12-0.64) (Supplementary Table 3)".

nature portfolio

I have additional suggestions concerning the software itself. These suggestions, as previously mentioned, will not impact my review of the manuscript and should be regarded as recommendations aimed at improving the installation and usability of COBRA.

- In my tests, I noticed that COBRA exhibited sluggish performance when handling large assemblies. Upon inspecting the code, I identified multiple steps that could be easily parallelized. Implementing these modifications would result in a substantial enhancement of COBRA's processing speed, enabling it to efficiently handle datasets, including large ones. These optimizations require minor adjustments, such as storing data in memory rather than processing it sequentially as files are read.

We thank the reviewers for testing COBRA and thus giving these suggestions. We noticed that the reviewers have tested some huge assemblies, which usually indicate complicated communities. As shown by our tested data, such samples usually took relatively longer as there should be more potential joins for COBRA to evaluate, which thus led to the sluggish performance they experienced. Unfortunately this step could not be parallelized for now, as COBRA has to evaluate all potential joins at the same time. This has been added to the Discussion of the revised manuscript.

We inspected our codes and tried our best to make it faster by optimizing several processing steps. The new version of COBRA has been updated on the GitHub page accordingly.

- Having the software available for installation via Mamba/Conda would be greatly appreciated by the community, since these package managers are pretty much standard in bioinformatics nowadays. Since COBRA is available in PyPI already, it should be straightforward to write a recipe for Bioconda.

We thank the reviewers for the suggestion. We are working on it, and it may take several days to several weeks to be approved by the Bioconda team. We will update the GitHub page accordingly once it is approved.

Reviewer #3 (Remarks to the Author):

The authors have done a great job of revising the manuscript to assuage my concerns and criticisms. It is exciting that further benchmarking showed this tool to be so helpful.

We thank the reviewers for the helpful comments and positive evaluation of our revised manuscript.

One surprise was that the tool can only do 5,000 queries in several hours, the limitation being parallel processing. Is there something about COBRA that prevents parallel processing? It would make mining the vast short-read metagenomic space much easier and contribute to its broader use if it could at least implement parallel processing as a way to grow the number of queries that can be handled.

Our original sentence of “We suggest trying to extend no more than 5,000 queries to complete the analysis in several hours” may mislead the reviewers to believe that COBRA could only analyze 5,000 queries. Actually, this is just an example number, to give readers and users a quick evaluation of how long they may need to finish the analyses.

Actually, due to the coding optimization during the last revision, the high number of the queries is not a speed-limiting factor for COBRA (Supplementary Table 9). As noticed by Reviewer #2 that COBRA may take a longer time to proceed if the whole assembled dataset is huge (see our response above).

We have modified the corresponding sentence in the Discuss section to avoid any misunderstanding.

Decision Letter, second revision:

Message: Our ref: NMICROBIOL-23061360B

24th November 2023

Dear Jill,

Thank you for your patience as we've prepared the guidelines for final submission of your Nature Microbiology manuscript, "COBRA improves the quality of viral genomes assembled from metagenomes" (NMICROBIOL-23061360B). Please carefully follow the step-by-step instructions provided in the attached file, and add a response in each row of the table to indicate the changes that you have made. Ensuring that each point is addressed will help to ensure that your revised manuscript can be swiftly handed over to our production team.

In recognition of the time and expertise our reviewers provide to Nature Microbiology's editorial process, we would like to formally acknowledge their contribution to the external peer review of your manuscript entitled "COBRA improves the quality of viral genomes assembled from metagenomes". For those reviewers who give their assent, we will be publishing their names alongside the published article.

Nature Microbiology offers a Transparent Peer Review option for new original research manuscripts submitted after December 1st, 2019. As part of this initiative, we encourage our authors to support increased transparency into the peer review process by agreeing to have the reviewer comments, author rebuttal letters, and editorial decision letters published as a Supplementary item. When you submit your final files please clearly state in your cover letter whether or not you would like to participate in this initiative. Please note that failure to state your preference will result in delays in accepting your manuscript for publication.

Cover suggestions

COVER ARTWORK: We welcome submissions of artwork for consideration for our cover. For more information, please see our [a href=https://www.nature.com/documents/Nature_covers_author_guide.pdf target="new">guide for cover artwork](https://www.nature.com/documents/Nature_covers_author_guide.pdf).

Nature Microbiology has now transitioned to a unified Rights Collection system which will allow our Author Services team to quickly and easily collect the rights and permissions required to publish your work. Approximately 10 days after your paper is formally accepted, you will receive an email in providing you with a link to complete the grant of rights. If your paper is eligible for Open Access, our Author Services team will also be in touch regarding any additional information that may be required to arrange payment for your article.

Please note that *Nature Microbiology* is a Transformative Journal (TJ). Authors may publish their research with us through the traditional subscription access route or make their paper immediately open access through payment of an article-processing charge (APC). Authors will not be required to make a final decision about access to their article until it has been accepted. [Find out more about Transformative Journals](https://www.springernature.com/gp/open-research/transformative-journals)

Authors may need to take specific actions to achieve [compliance](https://www.springernature.com/gp/open-research/funding/policy-compliance-faqs) with funder and institutional open access mandates. If your research is supported by a funder that requires immediate open access (e.g. according to [Plan S principles](https://www.springernature.com/gp/open-research/plan-s-compliance)) then you should select the gold OA route, and we will direct you to the compliant route where possible. For authors selecting the subscription publication route, the journal's standard licensing terms will need to be accepted, including [self-archiving policies](https://www.nature.com/nature-portfolio/editorial-policies/self-archiving-and-license-to-publish). Those licensing terms will supersede any other terms that the author or any third party may assert apply to any version of the manuscript.

Best regards,

Final Decision Letter:

Message 19th December 2023

:

Dear Jill,

I am pleased to accept your Article "COBRA improves the completeness and contiguity of viral genomes assembled from metagenomes" for publication in Nature Microbiology. Thank you for having chosen to submit your work to us and many congratulations.

You may wish to make your media relations office aware of your accepted publication, in case they consider it appropriate to organize some internal or external publicity. Once your paper has been scheduled you will receive an email confirming the publication details. This is normally 3-4 working days in advance of publication. If you need additional notice of the date and time of publication, please let the production team know when you receive the proof of your article to ensure there is sufficient time to coordinate. Further information on our embargo policies can be found here:

<https://www.nature.com/authors/policies/embargo.html>

Please note that *Nature Microbiology* is a Transformative Journal (TJ). Authors may publish their research with us through the traditional subscription access route or make their paper immediately open access through payment of an article-processing charge (APC). Authors will not be required to make a final decision about access to their article until it has been accepted. [Find out more about Transformative Journals](https://www.springernature.com/gp/open-research/transformative-journals)

Authors may need to take specific actions to achieve [compliance](https://www.springernature.com/gp/open-research/funding/policy-compliance-faqs) with funder and institutional open access mandates. If your research is supported by a funder that requires immediate open access (e.g. according to [Plan S principles](https://www.springernature.com/gp/open-research/plan-s-compliance)) then you should select the gold OA route, and we will direct you to the compliant route where possible. For authors selecting the subscription publication route, the journal's standard licensing terms will need to be accepted, including [self-archiving policies](https://www.nature.com/nature-portfolio/editorial-policies/self-archiving-and-license-to-publish). Those licensing terms will supersede any

nature portfolio

other terms that the author or any third party may assert apply to any version of the manuscript.

Congratulations once again and I look forward to seeing the article published.

With kind regards,